# The master energy homeostasis regulator PGC-1α exhibits an mRNA nuclear export function

Simeon R. Mihaylov [1,2,10], Lydia M. Castelli[1,10], Ya-Hui Lin [1,10], Aytac Gül [1], Nikita Soni [1], Christopher Hastings[1], Helen R. Flynn [3], Oana Păun[4], Mark J. Dickman [5,6], Ambrosius P. Snijders[3,9], Robert Goldstone[7], Oliver Bandmann [1,6,8], Tatyana A. Shelkovnikova[1,6], Heather Mortiboys [1,6,8], Sila K. Ultanir [2] & Guillaume M. Hautbergue [1,6,8] ✉

PGC-1α plays a central role in maintaining mitochondrial and energy metabolism homeostasis, linking external stimuli to transcriptional co-activation of genes involved in adaptive and age-related pathways. The carboxyl-terminus encodes a serine/arginine-rich (RS) region and an RNA recognition motif, however the RNA-processing function(s) were poorly investigated over the past 20 years. Here, we show that the RS domain of human PGC-1α directly interacts with RNA and the nuclear RNA export receptor NXF1. Inducible depletion of PGC-1α and expression of RNAi-resistant RS-deleted PGC-1α further demonstrate that its RNA/NXF1-binding activity is required for the nuclear export of some canonical mitochondrial-related mRNAs and mitochondrial homeostasis. Genome-wide investigations reveal that the nuclear export function is not strictly linked to promoter-binding, identifying in turn novel regulatory targets of PGC-1α in non-homologous end-joining and nucleocytoplasmic transport. These findings provide new directions to further elucidate the roles of PGC-1α in gene expression, metabolic disorders, aging and neurodegeneration.

The peroxisome proliferator-activated receptor gamma (PPARγ) coactivator 1 alpha (PGC-1α, encoded by the *PPARGC1A* gene) exerts a central regulatory role in sensing signal transduction pathways and maintaining the energy metabolism homeostasis in response to physiological processes during adaptive thermogenesis, exercise and fasting[1–4]. It is predominantly expressed in tissues with high energy demands such as heart, skeletal muscle, brown adipose tissue, liver, kidney and brain[1,5]. Conversely, the expression level/activity of PGC-1α and PGC-1α-dependent metabolic pathways are down-regulated in aging and age-related diseases (muscle wasting, metabolic, neurodegenerative and neurodevelopmental disorders)[6,7], highlighting the key importance of this master homeostasis regulator in human physiology.

Through its N-terminal domain, PGC-1α co-activates the transcription of genes involved in mitochondrial biogenesis and

[1]Sheffield Institute for Translational Neuroscience (SITraN), Department of Neuroscience, University of Sheffield, 385 Glossop Road, Sheffield S10 2HQ, UK. [2]Kinases and Brain Development Laboratory, The Francis Crick Institute, 1 Midland Road, London NW1 1AT, UK. [3]Proteomics Science Technology Platform, The Francis Crick Institute, 1 Midland Road, London NW1 1AT, UK. [4]Neural Stem Cell Biology Laboratory, The Francis Crick Institute, 1 Midland Road, London NW1 1AT, UK. [5]Department of Chemical and Biological Engineering, Sir Robert Hadfield Building, University of Sheffield, Mappin Street, Sheffield S1 3JD, UK. [6]Neuroscience Institute, University of Sheffield, Western Bank, Sheffield S10 2TN, UK. [7]Bioinformatics and Biostatistics Science and Technology Platform, The Francis Crick Institute, 1 Midland Road, London NW1 1AT, UK. [8]Healthy Lifespan Institute (HELSI), University of Sheffield, Western Bank, Sheffield S10 2TN, UK. [9]Present address: Life Science Mass Spectrometry, Bruker Daltonics, Banner Lane, Coventry CV4 9GH, UK. [10]These authors contributed equally: Simeon R. Mihaylov, Lydia M. Castelli, Ya-Hui Lin. ✉e-mail: g.hautbergue@sheffield.ac.uk

maintenance and in varied metabolic pathways through binding and activation of transcriptional activators, both nuclear receptors and other transcription factors, which co-activates the transcription of tissue-specific metabolic programmes[8]. PGC-1α was discovered in brown adipose tissue as a mediator of non-shivering thermogenesis upon exposure to low temperatures[1]. There, it binds the nuclear receptors PPARα/γ[1,9] and retinoid X receptor-alpha (RXRα)[10,11] to co-activate the expression of uncoupling protein-1 (UCP-1) in mitochondria which uncouples the electron transport chain (ETC) to produce heat. In the liver and skeletal muscle, interactions of PGC-1α with PPARs and RXRs lead to activation of the mitochondrial fatty acid oxidation while the co-activation of hepatocyte nuclear factor 4α (HNF4α) controls gluconeogenesis in the liver in an insulin-dependent manner[1–4,12]. Binding of PGC-1α to ERRα (estrogen-related receptor α) also regulates glucose and fatty acid metabolism in the striated muscle[13,14]. On the other hand, PGC-1α is highly upregulated in skeletal muscles after cold exposure or exercise which leads to the increase of mitochondrial biogenesis and maintenance to meet energy demands[5]. In skeletal muscle and brown adipose tissue, PGC-1α directly induces the transcription of nuclear respiratory factors-1/2 (NRF-1/2). It also directly interacts and co-activates these transcription factors to initiate in turn the transcription of nuclear-encoded mitochondrial transcription factor A (TFAM/mtTFA)[5] and other programmes of gene expression including electron transport chain and genes involved in oxidative phosphorylation (OXPHOS). Translocation of TFAM into the mitochondria subsequently stimulates replication and transcription of the mitochondrial genome promoting mitochondrial homeostasis and oxidative phosphorylation[15].

Promoter recruitment of PGC-1α to nuclear receptors and transcriptional factors also stimulates interactions with chromatin remodelling factors[16] and RNA polymerase II[17] through interactions with (i) Med1 in the Mediator/p300 complex via its C-terminal Serine/Arginine (RS)-rich region[18], (ii) the general transcription initiation factor TFIIH[19] and (iii) the cyclin-dependent kinase complex CDK9/P-TEFb[17], which overcome promoter-proximal pausing[20], to further promote the transcriptional activation of target genes. The C-terminal domain also interacts with RNA for the assembly of protein complexes regulating transcription and the recruitment of PGC-1α to transcriptionally-active chromatin condensates[21]. The expression and activity of PGC-1α is tightly regulated via methylation of its promoter[22,23], post-translational modifications[24–27] and protein degradation[28–30]/stabilisation[30,31].

The C-terminal domain of PGC-1α has been poorly investigated in comparison to its other domains. It harbours an RS-rich region flanked by a putative RNA Recognition Motif (RRM), reminiscent of SR-rich splicing factors, and a short motif which interacts with the CAP-binding protein 80 (CBP80) to promote the expression of a subset of transcripts[32]. On the other hand, the co-transcriptional recruitment of SR-rich splicing factors SRSF1,3,7[33–37] and Aly/REF[38–41] links deposition during pre-mRNA splicing and mRNA nuclear export through direct interactions with nuclear RNA export factor 1 (NXF1). This subsequently leads to the remodelling of NXF1 into a high RNA-affinity mode[42,43], which mediates transport through the channel of nuclear pores via transient interactions with protruding phenylalanine-glycine (FG) repeats of nucleoporins[44,45]. Interestingly, PGC-1α is involved in the co-transcriptional splicing of a co-activated reporter transcript[17]. Using enhanced ultraviolet (UV) cross-linking and immunoprecipitation followed by sequencing (eCLIP), the C-terminal domain of PGC-1α was also recently shown to interact with a specific subset of transcripts in response to glucagon, a central fasting axis in mouse hepatocytes[46]. Although its potential role(s) in the splicing or processing of cellular transcripts remain unknown, this study highlighted for the first time that the RNA-binding activity of PGC-1α is physiologically regulated.

Here, we show that the RS domain of PGC-1α directly interacts with RNA and NXF1 in assays with purified recombinant proteins and RNA oligonucleotides as well as with poly-adenylated mRNAs in human

embryonic kidney (HEK) 293T cells. Using a stable inducible isogenic complementation system which allows for depletion of endogenous PGC-1α and expression of an RNAi-resistant form of PGC-1α lacking the RS domain, we further demonstrate that the RNA-binding activity of PGC-1α is essential to the nuclear export of some known co-activated transcripts involved in mitochondrial metabolism as well as in mitochondrial respiratory chain functionality and cell proliferation. Genome-wide investigations, which included chromatin immunoprecipitation followed by sequencing (ChIP-seq), RNA-seq and quantitative proteomics, further confirmed in our model system the canonical roles of PGC-1α in binding >2200 promoters of genes involved in oxidative phosphorylation, thermogenesis, various metabolic pathways and the regulation of longevity, while also revealing that PGC-1α displays an additional function in the nuclear export of >1200 cellular mRNAs linked to or independent of its binding to promoters. The mRNA nuclear export analysis confirmed a role of PGC-1α in the glucagon-signalling pathway independently of the canonical transcriptional co-activation of OXPHOS/gluconeogenic promoters in agreement with the published RNA-binding targets[46]. It also identified novel PGC-1α targets involved in non-canonical pathways directly relevant to the age-related regulatory functions of PGC-1α, including non-homologous end-joining (NHEJ) double-strand break DNA repair and the nucleocytoplasmic transport. Taken together, we discovered that PGC-1α, a major physiological regulator extensively studied over the past 20 years (>4500 publications in *PubMed*), exhibits a novel mRNA nuclear export activity, essential to its function, at the heart of the energy metabolism homeostasis.

## Results

### PGC-1α binds RNA via the RS domain

The carboxyl-terminal domain of PGC-1α harbours a serine-arginine-rich (RS) region flanked by a putative RNA recognition motif (RRM) and a CBP80-binding motif (CBM). To characterize the RNA-binding domain of PGC-1α, we performed ultra-violet (UV) irradiation mediated covalent cross-linking of interacting RNA probes with purified recombinant proteins. PGC-1α full length could not be expressed in *E. coli* despite fusion to various solubility and expression tags including protein G domain B1 (GB1) and glutathione-S-transferase (GST). We further engineered a series of plasmids expressing GB1-hexa-histidine-tagged (GB1-6His) transcriptional co-activation (D1, amino acids aa1–234), suppression/regulatory (D2, aa254–564), putative RNA-binding (D3, aa565–798), RS (aa565–633) and RRM + CBM (aa634–798) domains of human PGC-1α (Fig. 1a). Recombinant proteins were successfully expressed in *E. coli* prior to ion metal affinity chromatography purification from soluble and insoluble fractions (Supplementary Fig. 1). Most domains, except D1 and CBM, were produced in inclusion bodies and were therefore solubilised using urea prior to large batch affinity purification in buffers with high salt concentrations to further disrupt potential contaminant interactions with bacterial RNA and proteins. Rapid dilution of denatured protein domains in the final binding assay buffers further allowed bypassing the intermediate concentrations of denaturing urea that typically lead to formation of intermolecular aggregation and misfolded proteins[47,48] (Supplementary Method 1). Purified human recombinant GB1-6His-tagged PGC-1α D3 or Aly/REF (a known mRNA nuclear export adaptor which binds RNA without sequence consensus[38,49] and 6His-MAGOH (a control protein which does not bind RNA[50,51]) were then incubated in the same buffer conditions with [32]P-radiolabelled synthetic AU-rich (AAAAUUx5) or GC-rich (GGGGCCx5) RNA probes prior to irradiation with UV where indicated (+). PhosphoImages showed that Aly/REF and PGC-1α D3 directly interact in a non-sequence specific manner with both AU- and GC-rich RNAs, with UV-cross-linked covalently-bound RNA molecules associated to the proteins visualized on the Coomassie-stained panels. The protein:RNA interactions were specific since no binding of RNA was detected in the absence of UV-irradiation

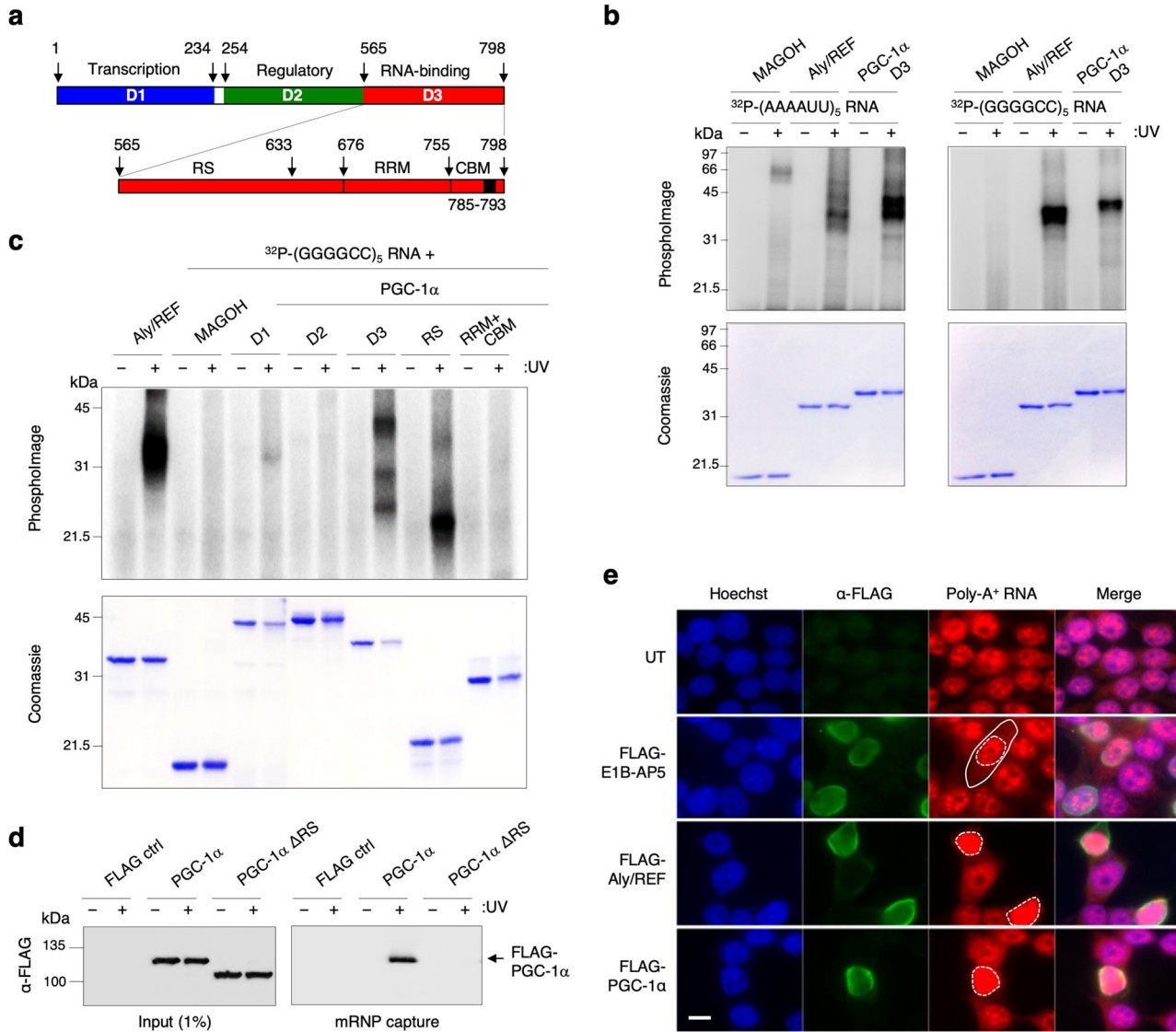

**Fig. 1 | The RS domain of PGC-1α interacts with RNA. a** Schematic representation of the primary structure of PGC-1α including the N-terminal transcriptional co-activation domain (blue), suppression/regulatory domain (green) and the C-terminal RNA-binding domain (red), which is further expanded below to highlight the RS, RRM and CBM domains. **b** Protein:RNA UV-cross-linking assays. Recombinant PGC-1α D3 (aa565–798) was incubated with AU- or GC-rich $^{32}$P-labelled RNA probes prior to UV irradiation (+) or not (−). Coomassie-stained gel shows recombinant proteins and phosphoimage shows radiolabelled RNA. Aly/REF and MAGOH are included as positive and negative controls, respectively. **c** Protein:RNA UV- Recombinant PGC-1α domains were incubated with GC-rich $^{32}$P-labelled RNA probe prior to UV irradiation (+) or not (−). Coomassie-stained gel shows recombinant proteins and phosphoimage shows RNA. Aly/REF and MAGOH are included as positive and negative controls, respectively. **d** mRNP capture assay. Human HEK293T cells were transfected for 48 h with either FLAG control, FLAG-tagged PGC-1α wild type or FLAG-tagged PGC-1α ΔRS. Cells were subjected to UV irradiation (+) or not (−). Protein extracts were subjected to denaturing mRNP capture assays using oligo-dT beads. Eluted proteins were analysed by western blotting probed with α-FLAG antibody. **e** Bulk poly(A)+ cellular distribution upon overexpression of PGC-1α. HEK293T cells were transfected with E1B-AP5/hnRNPUL1 (negative control), Aly/REF (positive mRNA export adaptor control) and PGC-1α for 48 h followed by a 2 h treatment with 5 μg/ml Actinomycin D. Cells were stained with anti-FLAG antibody (green) to visualise transfected cells and Cy3-oligo(dT) (red) that anneal to the poly(A) tails of RNA molecules. The solid white line delineates the cytoplasm while the dotted lines label nuclei. Cells with overexpression of FLAG-tagged Aly/REF and PGC-1α show a block in the bulk nuclear export of mRNAs with saturated accumulation of Poly-A + RNA nuclear staining and absence of detectable cytoplasmic signal. Scale bar: 10 μm. For panels (**b**–**e**), similar results were obtained in at least three independent experimental settings.

or with the negative control protein MAGOH (Fig. 1b). A similar investigation of D3 subdomains demonstrated that the RS domain is required and sufficient for direct binding to RNA (Fig. 1c). It is noteworthy that the addition of urea during the purification is not expected to affect potential folding of the RS domain as these arginine-serine-rich regions are intrinsically-disordered[52] (Supplementary Fig. 2).

To evaluate the in vivo relevance of these findings, we performed UV-mediated protein:RNA crosslinks on live cells prior to denaturing messenger-ribonucleoprotein (mRNP) capture assays using oligo-dT beads. These showed that FLAG-tagged PGC-1α, but not the full-length

protein lacking the RS domain (ΔRS), directly interacts with the fraction of poly(A)$^+$-enriched mRNAs in human embryonic kidney (HEK293T) cells (Fig. 1d, Supplementary Fig. 3). We have previously shown that overexpression of the conserved mRNA nuclear export adaptor Aly/REF blocks the bulk NXF1-dependent nuclear export of poly(A)$^+$ RNAs[53,54], likely due to excessive adaptor:NXF1 interactions which stall the high RNA-affinity remodelling of NXF1 in absence of recruitment to spliced transcripts. Similarly, Cy3-oligo-dT Fluorescence In Situ Hybridisation (FISH) experiments showed that FLAG-tagged Aly/REF or PGC-1α trigger nuclear accumulation of poly(A) +

RNA in anti-FLAG-stained cells highly overexpressing these proteins, while the control NXF1-binding ribonucleoprotein E1B-AP5/hnRNPUL1[54] did not (Fig. 1e, Supplementary Fig. 4).

Taken together, our data demonstrate that the RS domain of PGC-1α directly interacts with RNA oligonucleotides as well as poly-adenylated RNA in human cells in agreement with a recent genome-wide eCLIP study reporting binding of PGC-1α to RNA in absence of strict consensus elements[46]. Similarly, we have previously reported that arginine-rich regions adjacent to RRM domains constitute broad-specificity RNA-binding sites for the nuclear mRNA export adaptors SRSF1,3,7 and Aly/REF[35,36,42,55,56].

## The RS domain of PGC-1α interacts with NXF1

The hallmark of mRNA nuclear export adaptors is their direct over-lapping interactions with RNA and NXF1 through unstructured arginine-rich sites flanked by proline residues[35,36,42,55,56]. Since the RS domain of PGC-1α directly binds RNA (Fig. 1), we sought to test whether it would also interact with NXF1 (Fig. 2a). Interestingly, short arginine-rich sequences flanked by prolines are observed in the RNA-binding domain of PGC-1α (Fig. 2b). To assess the potential interaction of PGC-1α with NXF1, we performed pull-down assays using bacterially-expressed GST-NXF1 and $^{35}$S-radiolabelled PGC-1α proteins synthe-sised in rabbit reticulocytes in the presence of RNase A to obliterate indirect interactions bridged by RNA molecules[35,36,42,53]. GST-NXF1 was co-expressed with p15/NXT1, a co-factor which improves its stability through heterodimerization with the NTF2-like domain (aa371–551) of NXF1[57,58]. GST pull-down assays showed that PGC-1α full length (WT) and domain D3 encompassing the RS region interact with NXF1 in an RNA-independent manner (Fig. 2c). Detecting the cellular interaction of NXF1 with nuclear export adaptors is challenging due to the tran-sient nature of the NXF1-driven nuclear export process. Therefore, as in our previous reports[35,36,42,53], we co-transfected HEK293T cells with FLAG-control, FLAG-PGC-1α wild type (WT) or FLAG-PGC-1α lacking the RS domain (ΔRS) and Myc-NXF1 plasmids. Moreover, FLAG-tagged PGC-1α constructs are reported to have moderate overexpression because PGC-1α is substrate of the ubiquitin-proteasome pathway[17,28]. Co-immunoprecipitation assays in HEK cells confirmed that PGC-1α interacts with NXF1 in an RNase-independent manner through the RS domain (Fig. 2d). RNase-treated GST pull down assays using $^{35}$S-radiolabelled PGC-1α protein domains synthesised in rabbit reti-culocytes further showed in vitro that the RS domains specifically interacts with NXF1 (Fig. 2e). Similarly, the RS domain is required and sufficient for the direct binding of NXF1 using stringently-purified proteins expressed in *E. coli* (Fig. 2f, Supplementary Method 1). Inter-estingly, D3 interacts with NXF1 when synthesised in a mammalian transcription/translation-coupled system which allows for post-translational modifications (Fig. 2e) whereas it does not when expressed in bacteria (Fig. 2f), suggesting that eukaryotic post-translational modifications play a role in the NXF1-dependent bind-ing of PGC-1α. As expected, deletion of the RS domain in the full-length PGC-1α protein expressed in rabbit reticulocytes considerably atte-nuated the interaction with NXF1 (Supplementary Fig. 5). Taken toge-ther, we show that the RS domain of PGC-1α is required for interactions with NXF1 and RNA both in vitro and in vivo in human cells.

## PGC-1α interacts with NXF1 and the nuclear pore complex out-side of nuclear splicing speckles

To evaluate the cellular distribution of PGC-1α, we performed immu-nocytochemistry coupled with *MALAT1* and *NEAT1* RNA-FISH in HeLa cells, which are better suited for imaging than smaller and less flat HEK cells, to detect splicing speckles and paraspeckles, respectively. Similar to an earlier report using an overexpression system[17], endogenous PGC-1α was found to predominantly localize in the nucleus with dis-tinct foci, however, no enrichment was observed in nuclear speckles or paraspeckles (Fig. 3a). This agrees with a recent study highlighting that

a large proportion of PGC-1α foci corresponds to chromatin con-densates rather than nuclear speckles[21]. Consistent with the pulldown results, proximity ligation assay (PLA) confirmed that PGC-1α and NXF1 interact in a large number of co-localisation events specifically occur-ring in the nucleus (Fig. 3b, top panels) outside of nuclear splicing speckles which were visualised with transiently expressed SRSF1-GFP protein[59] (Fig. 3b, bottom panels). Moreover, PLA revealed that PGC-1α is in close proximity with NUP107, a component of the nuclear outer ring at the channel entrance of the nuclear pore complex (Fig. 3c), identifying thus sites of interaction in congruence with a role of PGC-1α in the nuclear export of mRNAs.

## Engineering of isogenic inducible cell lines to investigate the potential mRNA nuclear export activity of PGC-1α

To assess the functional importance of the RS domain, we sought to engineer a human cell complementation system which allows for depletion of endogenous PGC-1α and its replacement with either a control FLAG-tagged PGC-1α wild-type protein or the mutant lacking the RS domain. For this purpose, stable isogenic HEK293T-FlpIn cell lines were generated (Supplementary Method 2). They allow doxycycline-inducible expression of the following transgenes inte-grated at the single genomic FRT site (Flp Recognition Target): (i) Control-miRNA (Ctrl-RNAi) which does not target human transcripts; (ii) PGC-1α miRNA cells (PGC-1α-RNAi) expressing 2 chained miRNAs recognising nucleotide sequences in D1 and D2 of PGC-1α; (iii) RNAi-resistant FLAG-tagged full-length PGC-1α cells co-expressing the PGC-1α-RNAi cassette (WT-res); (iv) RNAi-resistant FLAG-tagged RS-deleted PGC-1α cells co-expressing the PGC-1α-RNAi cassette (ΔRS-res). Con-trol cells with recombination of an empty plasmid at the FRT site also serve as a control healthy HEK293T cell line (Sham) (Fig. 4a). In con-trast to Ctrl-RNAi cells, doxycycline-induced expression of the PGC-1α-RNAi cassette leads to robust and specific depletion of endogenous PGC-1α transcripts and protein quantified by qRT-PCR (Fig. 4b) and western blot using an anti-PGC-1α antibody validated in previous studies[46,60,61] (Fig. 4c, d). Both the depletion of endogenous PGC-1α and expression of FLAG-tagged PGC-1α wild-type protein or mutant lacking the RS domain were confirmed in the WT-res and ΔRS-res cell lines (Fig. 4e, f). These data indicate that these cell lines allow inducible depletion of PGC-1α and/or co-expression of RNAi-resistant PGC-1α and PGC-1α-ΔRS proteins.

## PGC-1α drives the nuclear export of mRNAs transcribed from canonical gene targets encoding mitochondrial-related proteins

PGC-1α is reported to co-activate the nuclear transcription of *TFAM*[5], *NRF-1*[5] and some transcripts encoding the cytochrome c oxidase (COX) subunits of the terminal mitochondrial respiratory chain complex IV, including COX assembly homolog 10 (*COX10*)[62], C1orf31/ COX Assembly factor 6 (*COA6*) and COX subunit 5A (*COX5A*)[63]. Both UV-C and formaldehyde cross-linked RNA immunoprecipitation (RIP) assays show that transcripts from the above co-activated genes specifically interact with FLAG-tagged PGC-1α while the ΔRS mutant exhibits sig-nificantly less binding to RNA (Fig. 5a, Supplementary Fig. 6a). On the other hand, catalase (*CAT*) transcripts which are expressed from a PGC-1α-independent promoter[62,63] do not interact with PGC-1α (Fig. 5a). To functionally characterise the potential role of PGC-1α in the nuclear export of mRNAs, these transcripts were further quantified by qRT-PCR in total, nuclear and cytoplasmic fractions isolated from the doxycycline-induced cell lines. The quality of subcellular fractionation was validated using western blots probed by the nuclear chromatin remodelling factor SSRP1 and the cytoplasmic β-tubulin marker TUJ1, showing absence of nuclear contamination in the cytoplasmic frac-tions (Fig. 5b, c). Doxycycline-induced depletion of PGC-1α, quantified using primers annealing in the coding sequence (CDS) or in the 3′-untranslated region (3′-UTR), leads to nuclear accumulation and con-comitant cytoplasmic decrease of *TFAM*, *NRF1*, *COX10*, *COX5A* and

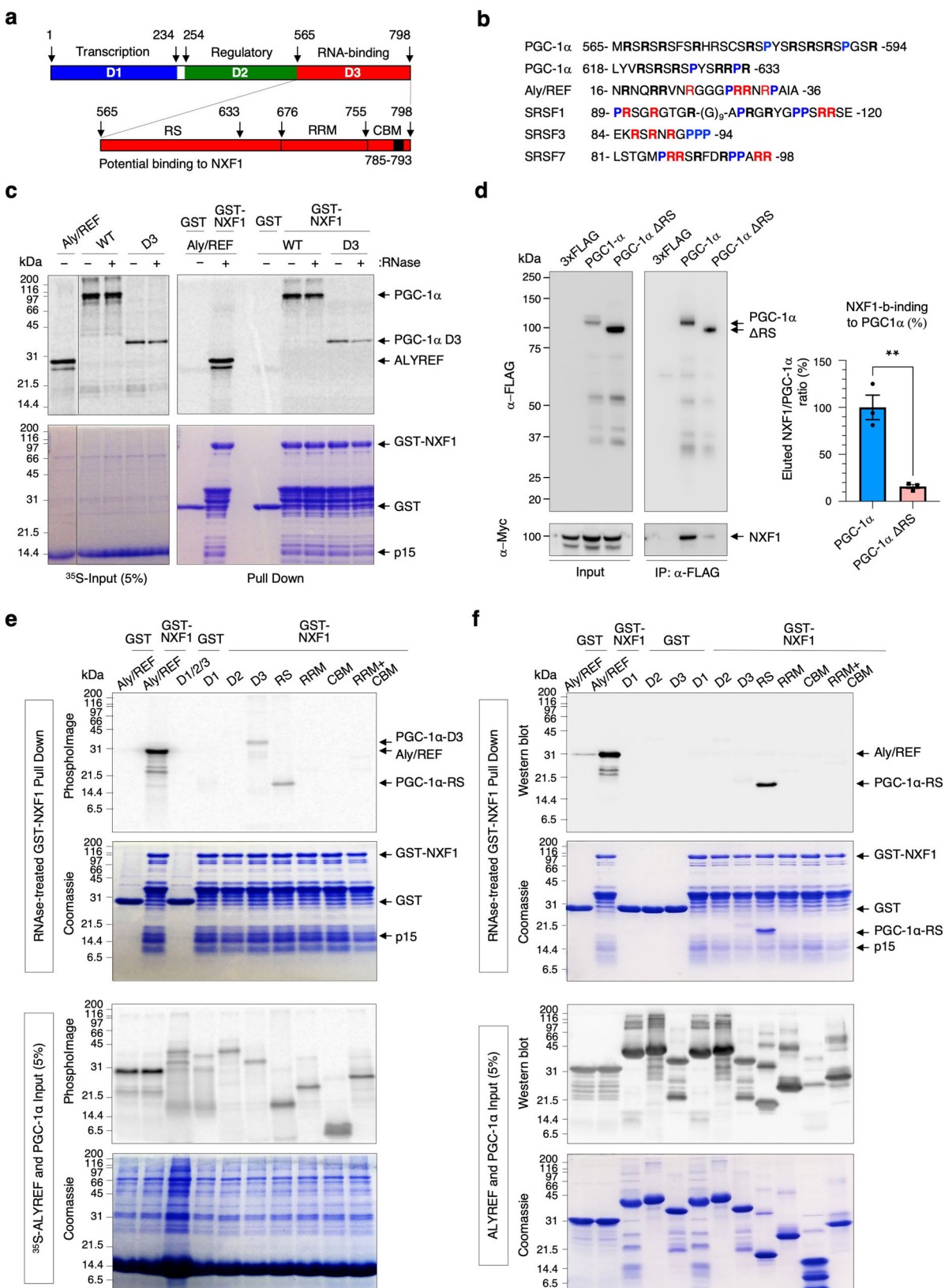

*COA6* in standard medium containing glucose (Fig. 5d), or with a 24-h galactose induction (Supplementary Fig. 6b, c) to stimulate mitochondrial cellular metabolism through oxidative phosphorylation instead of glycolysis[64]. The total expression levels of these transcripts did not significantly change upon ~70% depletion of PGC-1α mRNA, suggesting that the promoters of these canonical gene targets remain

transcriptionally activated by the remaining fraction of the PGC-1α protein and/or by other transcriptional co-activators which act in concert with PGC-1α on PGC-1α co-activated promoters. In contrast, the expression levels of *CAT* transcripts which are not transcriptionally regulated[62,63] or bound (Fig. 5a) by PGC-1α are not altered upon depletion of PGC-1α (Fig. 5d, Supplementary Fig. 6c). Similarly,

**Fig. 2 | The RS domain of PGC-1α interacts with the nuclear export receptor NXF1. a** Schematic representation of PGC-1α including the N-terminal transcriptional activation (D1, blue), regulatory (D2, green) and C-terminal (D3, red) domains. D3 is further expanded to highlight the RS, RRM and CBM domains. **b** Schematic showing arginine residues (in bold) flanked by prolines (in blue) in short unstructured peptide sequences of the arginine-rich RNA-binding domains of PGC-1α, Aly/REF, SRSF1, SRSF3 and SRSF7. Arginine residues labelled in red were previously shown to interact with RNA. **c** GST and GST-NXF1:p15 pull down assays with ³⁵S-methionine radiolabelled PGC-1α full length (WT) and D3. Aly/REF was used as a known mRNA nuclear export adaptor control. Coomassie-stained gels (bottom) show input and pull-down lanes. PhosphoImages (top) show expression of Aly/REF and PGC-1α (input) and co-purified proteins (pull down). **d** Co-immunoprecipitation assays. HEK293T cells were co-transfected with plasmids expressing FLAG control, FLAG-tagged PGC-1α WT or PGC-1α ΔRS and c-Myc-tagged NXF1. Whole-cell lysates were subjected to anti-FLAG immunoprecipitation in the presence of RNase. Western blots were carried out with antibodies against FLAG and c-Myc. The PGC-1α:NXF1 interactions were quantified in three independent experiments (Mean ± SEM; unpaired two-tailed Student's *t*-test; **$p = 0.0032$; $N = 3$). Source data with details of the statistical test is provided as a Source data file. **e** RNase-treated GST and GST-NXF1:p15 pull down assays with ³⁵S-methionine radiolabelled PGC-1α protein domains (top Coomassie-stained gels and PhosphoImages panels). Bottom two Coomassie-stained gels and PhosphoImages show input lanes. **f** RNase-treated GST and GST-NXF1:p15 pull down assays with bacterially-expressed and stringently-purified recombinant PGC-1α proteins (top Coomassie-stained gels and anti-6His western blots panels). Bottom two Coomassie-stained gels and anti-6His western blots show input lanes. For panels **c–f**, similar results were obtained in at least three independent experimental settings.

doxycycline-induced depletion of endogenous PGC-1α quantified with the 3'UTR primers and expression of RNAi-resistant FLAG-tagged PGC-1α WT/ΔRS measured by the CDS-specific primers led to nuclear accumulation and cytoplasmic decrease of *TFAM, NRF1, COX10, COX5A* and *COA6* mRNAs in the ΔRS-res cell line, showing that expression of the mutant PGC-1α lacking the RNA/NXF1-binding domain is not able to rescue the nuclear export of these transcripts, while restoring the expression of the wild type protein rescues the defects (Fig. 5e). In order to evaluate the functional relevance of the PGC-1α-dependent mRNA nuclear export function, we examined the expression levels of the TFAM protein and found that it has reduced expression levels in the induced PGC-1α-RNAi and ΔRS-res cell lines, but not in control and rescue conditions (Fig. 5f; quantification in Fig. 5g, h). The western blot quantification also confirmed the ~70% depletion of PGC-1α at protein level. Taken together, our data demonstrate that the NXF1/RNA-binding domain of PGC-1α exhibits a novel mRNA nuclear export function.

### The mRNA nuclear export activity of PGC-1α is essential to its cellular function and mitochondrial homeostasis

*TFAM* and *NRF1* encode factors involved in stimulating the expression of genes involved in mitochondrial biogenesis, while *COX10* and *COX5A* transcripts encode subunits of the respiratory chain complex IV which are of critical importance for the assembly of the complex[65]. Western blots probed with OXPHOS antibodies, a cocktail of 5 antibodies for simultaneous detection of a representative protein in each of the mitochondrial complexes I to V, indicated that the expression level of mitochondrial-encoded complex IV COX2 is down-regulated upon depletion of PGC-1α (Fig. 6a) or expression of the RNA/NXF1-binding mutant (Fig. 6b). We further confirmed the expected galactose-mediated induction of the COX2 protein and the significant alteration of its expression levels in PGC-1α-RNAi or ΔRS-res cell lines in galactose media (Fig. 6c, d) in agreement with the role of PGC-1α in the nuclear export of transcripts encoding proteins involved in mitochondrial biogenesis and complex IV assembly. On the other hand, the expression of proteins involved in complex I, III and V was not affected and there was a trend for an upregulation of the SDHB protein composing complex II (Supplementary Fig. 7a, b).

We next sought to use more sensitive assays to measure the enzymatic activity of respiratory chain complexes I, II and IV in the induced WT-res and ΔRS-res cell lines. These confirmed that depletion of PGC-1α and specific alteration of its RNA/NXF1-binding function leads to increased activity of complex II (Fig. 6e) and inhibition of complex IV activity (Fig. 6f), providing functional validation of protein changes observed above. On the other hand, the activity of complex I is not statistically altered compared to the Sham control line (Fig. 6g). We also sought to test if the altered nuclear export of *TFAM* and *NRF1* mRNAs affects mitochondrial genome replication in the induced PGC-1α-RNAi and ΔRS-res cell lines using qPCR analysis using an established protocol[66–68]. Expression of the unique and rarely deleted mitochondrial *MT-ND1* gene fragment is reduced upon depletion of PGC-1α and

expression of the ΔRS mutant (Fig. 6h), consistent with a role of PGC-1α in the nuclear export of transcripts which encode proteins stimulating the biogenesis of mitochondria. Finally, to evaluate the physiological relevance of the RNA/NXF1-binding function of PGC-1α, we measured cell proliferation using MTT assays in Sham, Ctrl-RNAi, PGC-1α-RNAi, WT-res and ΔRS-res cell lines. Both PGC-1α-RNAi and ΔRS-res cell lines showed reduced cell proliferation with a more pronounced defect in galactose condition (Fig. 6i, j), a result consistent with dependence on mitochondrial respiration and increased requirement of PGC-1α activity. Taken together, our data demonstrate that the NXF1-dependent mRNA nuclear export function of PGC-1α is essential to mitochondrial metabolism homeostasis and optimal cellular growth.

### Genome-wide promoter-binding, transcriptome and proteome dependence of the RS-mediated function of PGC-1α

To evaluate the global contribution of the Med1/NXF1/RNA-binding functions of PGC-1α in our complementation cell model, we performed ChIP-seq, RNA-seq and Tandem Mass Tag (TMT) quantitative mass spectrometry in the Sham, PGC-1α WT-res and ΔRS-res cell lines, grown in biological triplicate in same standard glucose conditions leading to expression of RNAi-resistant PGC-1α or PGC-1α-ΔRS and to ~70% depletion of endogenous PGC-1α (Fig. 5d–h).

For ChIP-seq, cells were subjected to cross-linking with di(N-succinimidyl)-glutarate (DSG) and formaldehyde prior to shearing chromatin into 300–400 bp fragments[69] and anti-FLAG-tagged PGC-1α ChIP (Supplementary Fig. 8a, b). Input and eluted PGC-1α-bound chromatin samples were sequenced on Illumina platform prior to bioinformatics analysis (Supplementary Methods 3–4). Using the Sham cell line as a background control for the anti-FLAG immunoprecipitation, we identified that binding of PGC-1α is enriched onto 2774 annotated promoters which include 2241 protein-coding promoters (81%) with a promoter standardly defined by the DNA region encompassing −5 kb and +5 kb from the transcription start site (TSS) (Fig. 7a, Supplementary Data 1 tabs 1–2). Representative snapshots of ChIP-seq peaks are provided for the known PGC-1α-bound *COX5A, COX10* and *COA6* promoters while the *SLC35F3* promoter, upstream of *COA6*, is a non-target example (Supplementary Fig. 9a). We also determined that PGC-1α-ΔRS interactions are enriched on 5992 annotated promoters of which 5025 are initiating the transcription of protein-coding genes (Supplementary Data 1 tab 3). 97% of PGC-1α-bound promoters overlap with those bound by the ΔRS mutant, indicating that recruitment to promoters is mostly independent of the RS domain (Fig. 7b). This finding is supported by biochemical data which showed that PGC-1α interacts with other transcriptional co-activators through zinc fingers located in the N-terminal domain[1,70,71]. *NRF1* and *TFAM* were also found in the list of promoters bound by PGC-1α-ΔRS. As anticipated, a KEGG pathway analysis indicated that PGC-1α-binding is significantly enriched on the promoters of genes involved in thermogenesis, respiration via oxidative phosphorylation, tricarboxylic

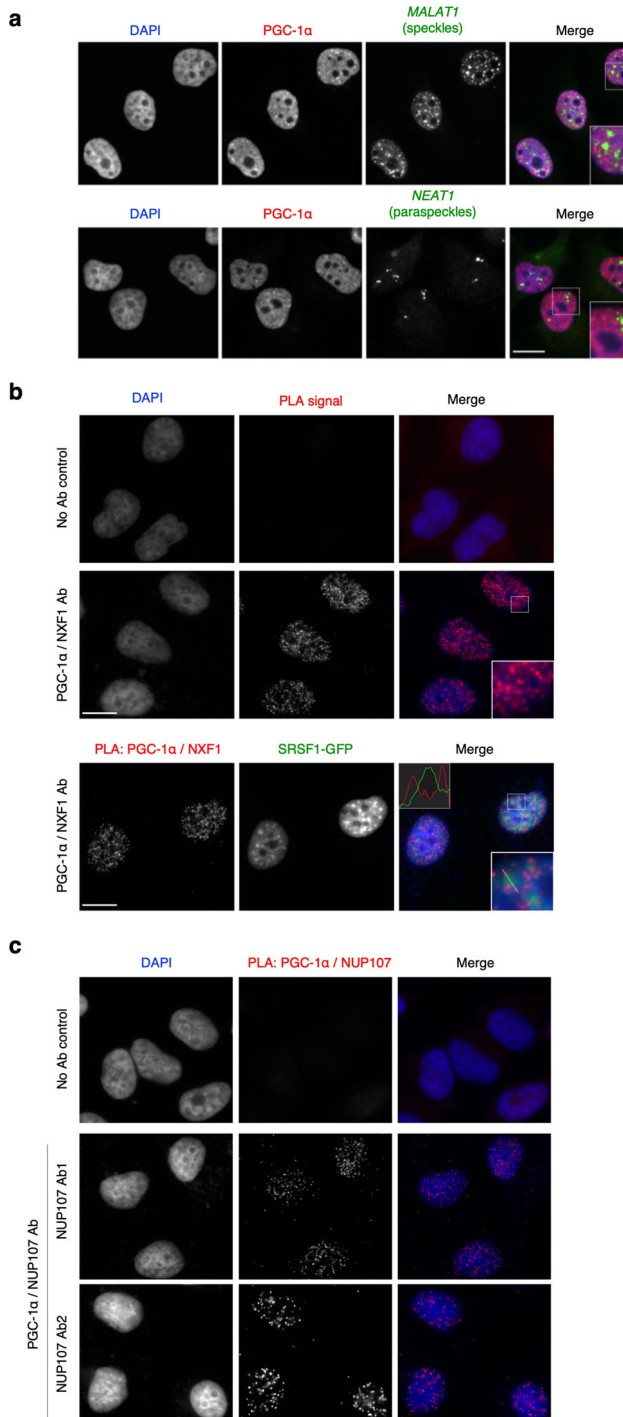

**Fig. 3 | PGC-1α interacts with NXF1 and NUP107 in the nucleoplasm outside of splicing speckles. a** Lack of PGC-1α enrichment in speckles and paraspeckles in HeLa cells. Anti-PGC-1α immunostaining was coupled with RNA-FISH using either Quasar570-labelled *MALAT1* or *NEAT1* probes to visualise speckles or paraspeckles, respectively. **b** Proximity ligation assays (PLA) were performed between endogenous PGC-1α and NFX1 in HeLa cells. SRSF1-GFP was transiently expressed to visualise splicing speckles (bottom panel). In control samples, the primary antibody step was omitted. Representative images are shown. **c** Endogenous PGC-1α extensively interacts with NUP107, a component of the nuclear outer ring of the nuclear pore complex. Representative PLA images for HeLa cells using two different anti-NUP107 antibodies are shown. In the control sample, the primary antibody step was omitted. NUP107 ab1: Proteintech 19217-1-AP; NUP107 ab2: Invitrogen PA5-30774; Scale bar: 10 μm in all panels. For panels **a–c**, similar results were obtained in at least three independent experimental settings.

acid (TCA)/Krebs cycle, carbon metabolism, AMPK/ insulin as well as other signalling pathways incuding the regulation of longevity (Fig. 7c). Promoter-binding and transcriptional co-activation by PGC-1α are regulated upon varied metabolic needs and cell types. Accordingly, we observed a poor overlap between our ChIP-seq in HEK cells and two other published datasets in mouse skeletal muscle[21] and human hepatic cells[63] (Supplementary Data 1 tabs 4–5, Supplementary Fig. 9b). However, as expected, gene ontology indicated that known regulated processes such as oxidative phosphorylation and metabolic pathways are commonly found in all ChIP-seq experiments, while the datasets of similar sizes in HEK and skeletal muscle cells exhibit multiple other pathways in common (Supplementary Fig. 9c, d).

To analyse the genome-wide RNA-processing functions of PGC-1α, we next generated whole-cell and cytoplasmic transcriptomes to identify RNA changes between the induced PGC-1α WT-res and ΔRS-res cell lines at expression, splicing and nuclear export levels. rRNA-depleted RNA sequencing libraries were subjected to high-depth RNA sequencing to investigate differential expression and splicing with high confidence (averaging 151.7 million reads per sample, 101-fold transcriptome coverage, Supplementary Data 2, Supplementary Methods 5–6). The lists of over 120,000 quantified transcript isoforms are presented in Supplementary Data 3 under 2 tabs for the whole-cell (WCT) and cytoplasmic (CyT) transcriptomes. Annotated transcripts from >19,000 genes, including >15,000 protein-coding genes (75% over ~20,000 protein-coding genes in the human genome), were commonly sequenced across all conditions, reflecting the generation of datasets with high transcriptome coverage without notable sequencing bias (Fig. 7d). Differentially-expressed transcript isoforms were filtered for fold change log2FC > 0.5 and p-value $p < 0.05$. Lists of total and cytoplasmic differentially-expressed RNA isoforms and genes (DEGs) are provided in Supplementary Data 4 tabs 1–2 for annotated and protein-coding transcripts in WCT and CyT transcriptomes, respectively. Expression levels of 968 transcripts (corresponding to 829 DEGs; <2% of the transcriptome comprising transcripts from ~42,500 coding and non-coding genes) are altered upon loss of the RS domain − 616 and 352 transcripts (corresponding to 465 and 300 mRNAs) being, respectively, down- and up-regulated (Fig. 7e left panel). On the other hand, we identified 3131 transcript changes in the cytoplasm (2625 DEGs) while 1835 and 1296 transcripts (corresponding to 1358 and 973 mRNAs) are, respectively, down- and up-regulated (Fig. 7e right panel). Consistent with the known PGC-1α-dependent role in the co-activation of protein-coding genes, >90% of RNA levels changed upon depletion of endogenous PGC-1α and expression of PGC-1α-ΔRS correspond to protein-coding transcripts (Fig. 7f). As aforementioned, the expression levels of a small number of mRNAs, 15% being transcribed from PGC-1α-bound promoters, were found to be down-regulated in the ΔRS-res cell line, reminiscent of the qRT-PCR analysis showing no alteration of total expression levels of canonical targets of PGC-1α such as *NRF1*, *TFAM*, *COX5A*, *COX10* or *COA6* (Fig. 5e). Down-regulated expression of mRNAs may be linked to alteration of the transcriptional co-activation which involves binding of the RS domain to the Mediator/P-TEFb complex on some promoters[18,20]. The KEGG analysis of down-regulated mRNAs highlighted few relevant pathways including carbon or signalling metabolisms (Supplementary Fig. 10a). A gene ontology based on biological processes provided more details indicating that an enriched fraction down-regulated mRNAs encodes 140 and 30 proteins involved in transcriptional regulation and RNA splicing, respectively (Supplementary Fig. 10b). Strikingly, more mRNAs are down-regulated in the cytoplasm of the ΔRS-res line (1358), consistent with the NXF1/RNA-binding domain of PGC-1α playing a role in the mRNA nuclear export. The KEGG analysis further highlighted that down-regulated cytoplasmic mRNAs are enriched in PGC-1α-related pathways including AMPK, insulin and

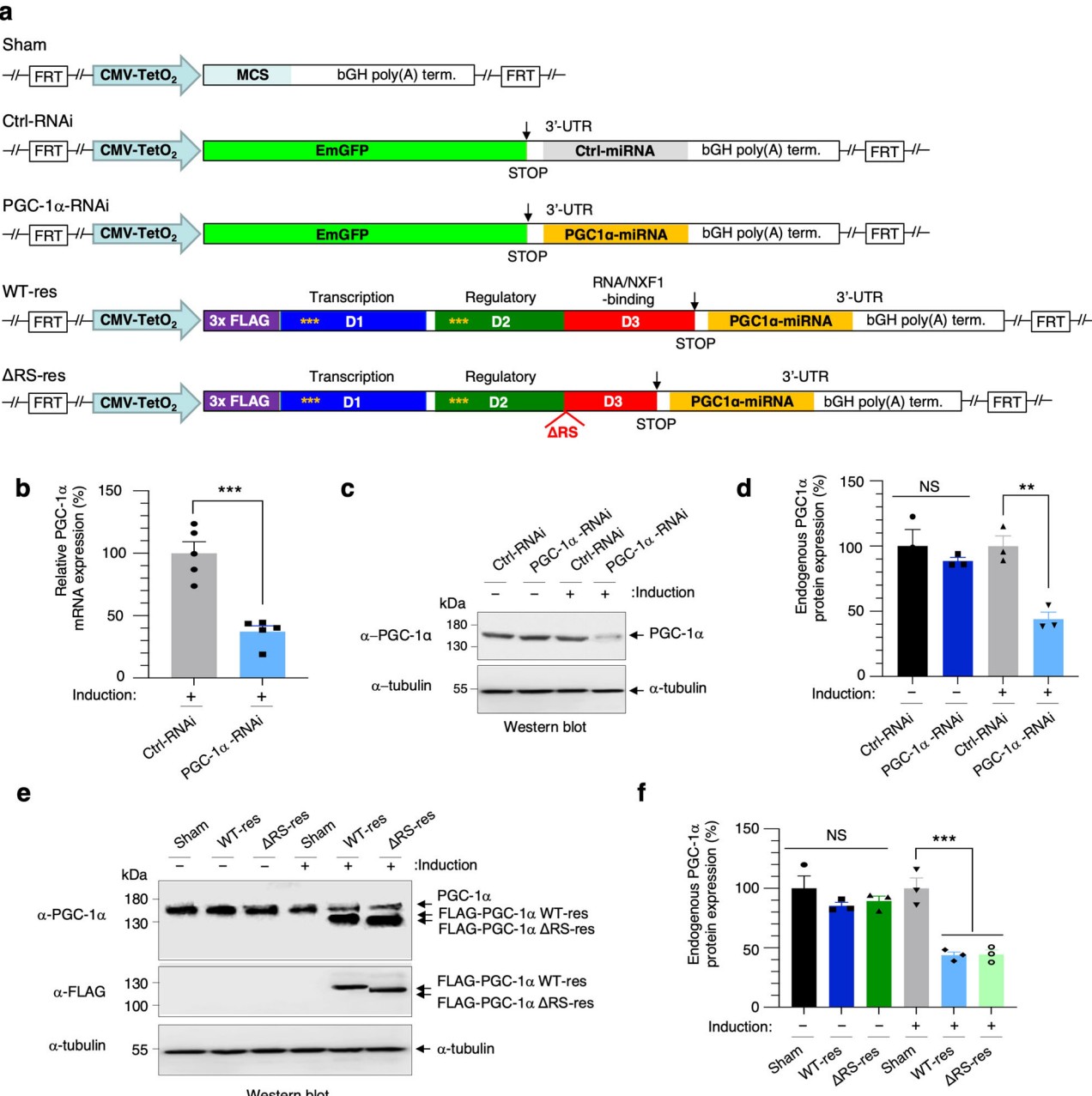

**Fig. 4 | Engineering and testing the functionality of human isogenic stable cell lines. a** Schematic representation of HEK293 Flp-In™ T-REx™ stable inducible cell line generated. Sham (Flp-In control) and either Ctrl-RNAi (scrambled non-targeting sequence) or PGC-1α RNAi stable cell lines were generated along with miRNA resistant N-terminally FLAG-tagged full length (WT) and PGC-1α-ΔRS cell lines which harbour the PGC-1α miRNA cassette in the 3'UTR. Yellow * highlight the position of the silent nucleotide changes which confer resistance to the miRNAs targeting endogenous PGC-1α transcripts. FRT indicates the plasmidic Flp Recognition Target sites which allows integration of the various transgenes at the single HEK293 Flp-In T-Rex genomic FRT site. **b** Endogenous PGC-1α transcript levels from of Ctrl-RNAi and PGC-1α RNAi cell lines induced with doxycycline for 72 h were quantified by qRT-PCR analysis following normalisation to U1 snRNA levels in five independent experiments (Mean ± SEM; Unpaired two-tailed Student's *t*-test;

***$p = 0.0003$, $N = 5$; $N = 5$). **c** Western blots from Ctrl-RNAi and PGC-1α RNAi cell lines either with or without doxycycline induction for 72 h. Blots were probed for PGC-1α and α-Tubulin. **d** Western blots shown in (**c**) were quantified in three independent experiments (Mean ± SEM; one-way ANOVA with Tukey's correction for multiple comparisons; NS: not significant, **$p = 0.0050$; $N = 3$). **e** Western blots from Sham, PGC-1α WT-res and PGC-1α PGC-1α ΔRS-res cell lines either with or without doxycycline induction for 72 h. Blots were probed for PGC-1α, FLAG, which detects the induced miRNA-resistant PGC-1α and α-Tubulin. **f** Western blots shown in (**e**) were quantified in three independent experiments (Mean ± SEM; one-way ANOVA with Tukey's correction for multiple comparisons; NS: not significant, ***$p = 0.0040$; $N = 3$). Source data with details of statistical tests are provided as a Source data file.

other signalling and metabolic pathways as well as regulation of longevity (Fig. 7g). Interestingly, axon guidance, nervous system development and neuronal-related processes were also highlighted in both the ChIP-seq and reduced total/cytoplasmic mRNAs lists. These pathways are relevant to the known involvement of altered PGC-1α

level/function in neurodegeneration, particularly in Parkinson's and Huntington's diseases.

To investigate the functional relevance of the RS domain of PGC-1α at proteome level, we identified the proteomes of WT-res and ΔRS-res cell lines using quantitative Tandem Mass Tag (TMT) mass

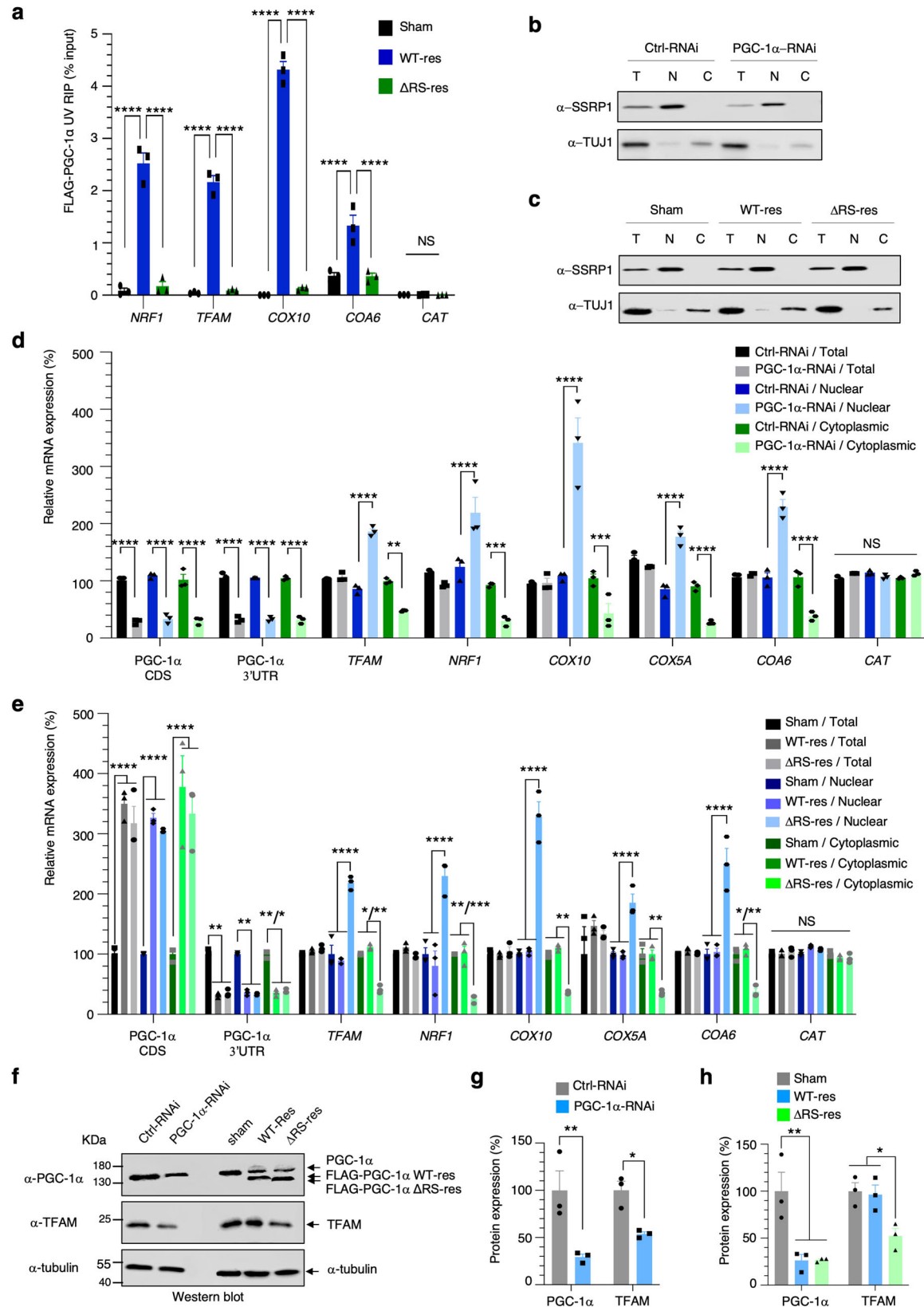

spectrometry (Supplementary Method 7). 7266 proteins were identified out of 75,777 in the database (~10% coverage). Expression levels of 128 and 86 proteins were significantly down- and up-regulated in the ΔRS-res cells, respectively (Fig. 7h, Supplementary Data 5). We further performed a KEGG pathway analysis on the list of down-regulated proteins to test how it compares with the largest lists of PGC-1α-bound

promoters or of mRNA down-regulated in the cytoplasm. Carbon and metabolic pathways were commonly highlighted while, interestingly, the non-homologous end-joining (NHEJ) double-strand break DNA repair pathway was common to reduced total/cytoplasmic mRNA (Supplementary Fig. 10b, Fig. 7g) and protein (Fig. 7i, Supplementary Fig. 10c) lists, including the NHEJ X-ray repair cross-complementing

**Fig. 5 | The RS domain of PGC-1α drives the mRNA nuclear export of some canonical gene targets encoding mitochondrial-related proteins. a** UV-C RNA immunoprecipitation (RIP) assays from Sham, FLAG-tagged RNAi-resistant PGC-1α WT (WT-res) and FLAG-tagged PGC-1α-ΔRS (ΔRS-res) human stable cell lines induced for 6 days with doxycycline and subjected to anti-FLAG immunoprecipitation. Purified RNA was analysed by qRT-PCR and expressed as a percentage of the input. RIPs were performed in three independent experiments (Mean ± SEM; two-way ANOVA with Turkey's correction for multiple comparisons; NS: not significant, ****$p < 0.0001$; $N = 3$). **b, c** Western blots of total (T), nuclear (N) and cytoplasmic (C) fractions from 6-day doxycycline-induced Ctrl-RNAi and PGC-1α-RNAi (**b**) or Sham, WT-res and ΔRS-res (**c**) cell lines. The chromatin remodelling factor SSRP1 was used to check for potential nuclear contamination in cytoplasmic fractions. Depletion of TUJ1 (beta-Tubulin III) in nuclear fractions was used to check for quality of the nuclear fractions. **d, e** Total, nuclear and cytoplasmic levels of fractionated RNA transcripts from (**b**) and (**c**) were quantified in three independent experiments by qRT-PCR following normalisation to U1 snRNA levels and to 100% in the Ctrl-RNAi (**d**) or Sham (**e**) cell lines (Mean ± SEM; two-way ANOVA with Tukey's correction for multiple comparisons, NS: not significant, **$p < 0.01$, ***$p < 0.001$, ****$p < 0.0001$; $N = 3$). The depletion of endogenous PGC-1α and expression of RNAi-resistant FLAG-tagged PGC-1α WT or ΔRS was validated with primers annealing in the 3′UTR of the PGC-1α gene or in the coding region (CDS) of the transgene, respectively. **f** Western blots from Ctrl-RNAi, PGC-1α-RNAi, Sham, WT-res and ΔRS-res cell lines induced with doxycycline for 6 days. Blots were probed for PGC-1α, TFAM and α-Tubulin antibodies. **g, h** Western blots shown in (**f**) were quantified in three independent experiments (Mean ± SEM; one-way ANOVA with correction for multiple comparisons; *$p < 0.05$, **$p < 0.01$; $N = 3$). The recommended Tukey's and Sidak's correction for multiple comparisons were, respectively, applied in (**g**) and (**h**). For panels **a, d, e, g, h**, source data are provided with details of statistical tests and exact *p*-values as a Source data file.

proteins 5 and 6 (XRCC5, XRCC6) and non-homologous end-joining factor 1 (NHEJ1). Interestingly, these non-canonical targets, involved in processes not previously known to be potentially regulated by PGC-1α, are linked to age-related telomere maintenance[72–74], pre-mature ageing and alteration of telomerase gene expression[75]. Down-regulation of DnaJ Heat Shock Protein Family (Hsp40) Member B6 (DNAJB6) was also linked to Huntington's disease[76], a neurodegenerative disorder known to exhibit reduced PGC-1α function in the substantia nigra of the brain. Down-regulation of the kidney-type mitochondrial glutaminase encoded by *GLS* provides another potential new link with mitochondrial/amino-acid metabolisms and neurotransmitter release as well as with cancer and neurological conditions. On the other hand, subunits of the respiratory chain complex II, SDHA, SDHB and SDHAF2/4 were upregulated in agreement with OXPHOS western blots and complex II activity (Fig. 6 and Supplementary Fig. 7). This respiratory complex has a dual function in both the oxidative phosphorylation and the tricarboxylic acid cycle which were both identified in the ChIP-seq data.

A Venn diagram comparing total mRNAs, cytoplasmic mRNAs and proteins down-regulated in the ΔRS-res cells highlights the different dynamic ranges of sensitivity between RNA-seq and proteomics, which covered ~75% of the coding transcriptome and ~10% of the proteome, respectively (Fig. 7j). However, importantly, 56% of the downregulated proteome changes (14 + 58 over 128) correlated with reduced cytoplasmic mRNA levels while only 15% (5 + 14 over 128) were linked to reduced mRNA expression (Fig. 7i), in line with genome-wide studies indicating that the abundance of mammalian proteins is mostly not attributed to total mRNA concentrations but rather on synthesis and stability[77–79].

**Genome-wide relevance of the mRNA nuclear export and potential splicing functions of PGC-1α**

Transcripts with unchanged total levels and reduced expression in the cytoplasmic fraction of ΔRS-res cells lacking the RNA-binding functions of PGC-1α were identified as PGC-1α-dependent RNA nuclear export (NE) targets (Supplementary Data 4 tab 3). A gene ontology analysis of the 1274 mRNAs NE targets of PGC-1α based on KEGG and biological processes highlighted the canonical AMPK-, adipose-, small GTPase- and longevity-related signalling pathways (Fig. 8a, b). It also confirmed the reported non-canonical RNA-binding role of PGC-1α in the glucagon-signalling pathway[46]. As previously revealed in the proteome and reduced expression of mRNAs, it further validated a role of the mRNA nuclear export function in the novel PGC-1α targets involved in the non-homologous end-joining dsDNA break DNA repair (*XRCC5*, *NHEJ1*) which is associated with age-related processes[72–75,80]. *XRCC6* mRNA showed a non-significant up-regulation in the total RNA samples and was thus not identified as a direct nuclear export target, however, its nuclear export is

predicted to be altered in the ΔRS-res cells since it is significantly down-regulated in the cytoplasm. A Venn diagram comparing the PGC-1α NE targets with the list of differentially expressed cytoplasmic mRNAs (up- and down-regulated) upon loss-of-function of the RS-domain indicates that the nuclear export function of PGC-1α controls the cytoplasmic abundance of 60% of its regulated mRNAs (Fig. 8c), while a comparison with the list of down-regulated cytoplasmic mRNAs shows that 94% of mRNAs with reduced cytoplasmic expression depends on the nuclear export function of PGC-1α (Fig. 8c). The PGC-1α RS domain-dependent genome-wide effects were also investigated at the alternative splicing level using the Whippet pipeline[81]. A total of 897 alternative splicing events were identified on 848 transcripts at the gene level including 691 alternatively spliced pre-mRNAs (Supplementary Data 4 tab 4). Tandem alternative polyadenylation transcription start sites were the most common alterations (Fig. 8d). The KEGG analysis highlights principally distinct pathways which are not enriched for the canonical metabolic functions, glucagon signalling or NHEJ dsDNA repair (Fig. 8e). It is noteworthy that 30 mRNAs encoding splicing factors showed reduced total expression levels (Supplementary Fig. 9e, Supplementary Data 4) which may account for the identified splicing dysregulation. There is indeed little overlap between the alternative splicing and mRNA nuclear export targets (Fig. 8f), suggesting that the nuclear export function of PGC-1α is not linked to the recruitment of PGC-1α during splicing in contrast to other mammalian mRNA nuclear export adaptors. In addition, a comparison with the promoter target list indicated that mRNAs transcribed from 18% of the promoters bound by PGC-1α are linked to the nuclear export function while 82% of PGC-1α mRNA export targets are not dependent on the canonical genes co-activated by PGC-1α, in agreement with findings from the RNA-binding target study[46]. Finally, we compared lists of promoters bound by PGC-1α with its nuclear export targets and the functional relationship at proteome level. This analysis showed that 52% of the down-regulated proteome changes are linked to the mRNA nuclear export function of PGC-1α while 17% of the proteome changes correlate with both PGC-1α-dependent binding to promoter and mRNA nuclear export (Fig. 8g). A more detailed gene ontology analysis was performed for all the aforementioned lists using the combined KEGG and GOTERMs CC (Cellular Component), BP (Biological Process) and MF (Molecular Function) functional annotation clustering of DAVID6.8[82]. Results are provided with gene names, enrichment scores and statistics in Supplementary Data 6. This investigation corroborated all previous findings with major pathways summarised in a heat map, including canonical promoter-binding targets and non-canonical nuclear export targets involved in the NHEJ dsDNA repair linked to the age-related maintenance of telomere and glucagon-

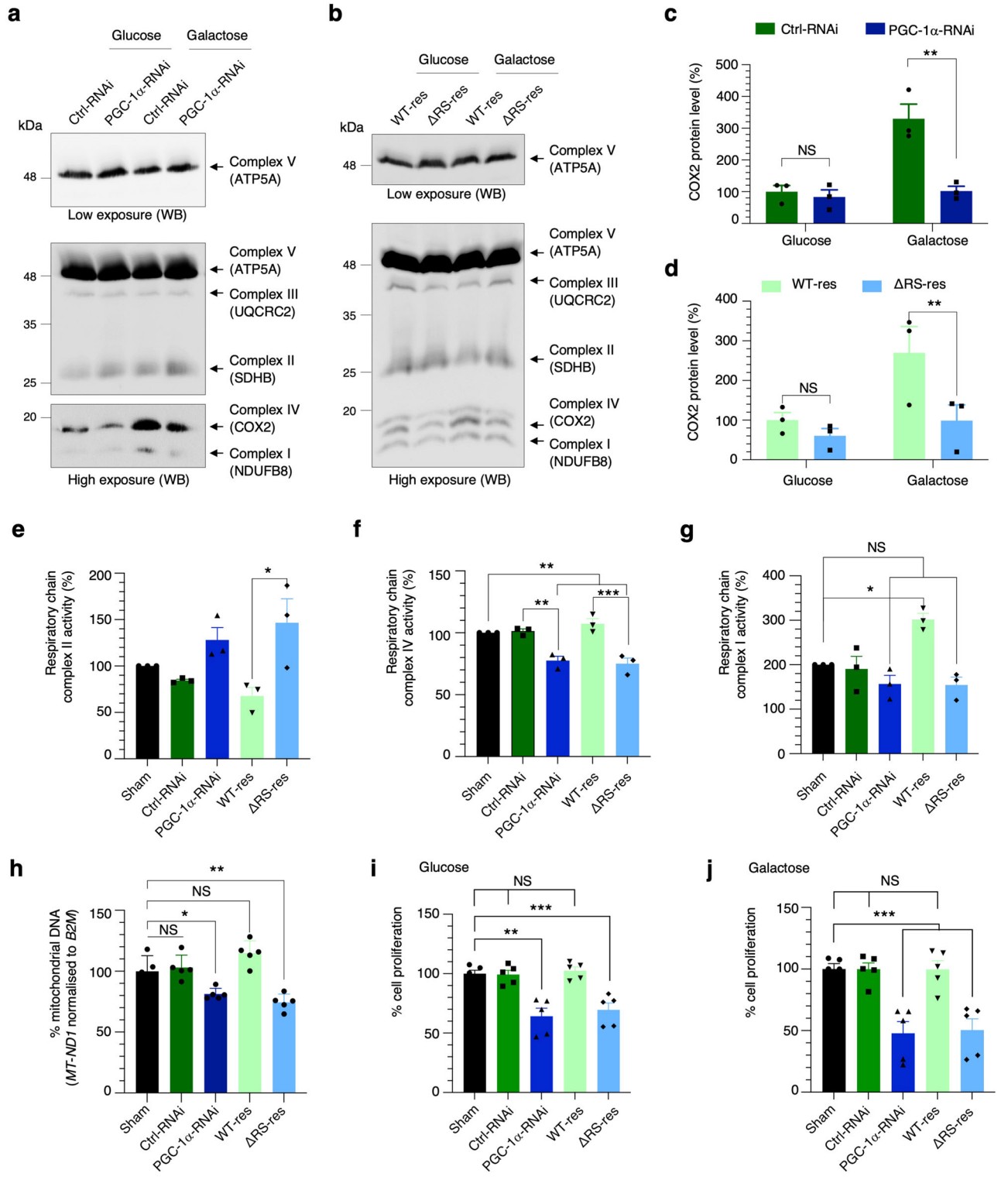

signalling (Fig. 8h). In addition, multiple processes are reassuringly shared across the various lists except for the reduced protein hits, a small list with less statistical power compared to the DNA/RNA-seq targets. This analysis also allowed detailing novel mRNA targets potentially regulated by PGC-1α, including nucleoporins *NUP58*, *NUP62*, *NUP98*, DEAD-box RNA helicase *DDX19B*, exportin *XPO1* and karyopherin *KPNA2* in the nucleocytoplasmic transport/mRNA export itself as well as some eukaryotic initiation factors and ribosomal subunits in the regulation of cytoplasmic translation. Interestingly, this finding provides another link to the age-related function of PGC-1α since alteration of the nucleocytoplasmic transport has also been reported as a hallmark of neuronal aging[80].

## PGC-1α regulates the nuclear export of non-canonical transcripts involved in age-related NHEJ maintenance of telomere and nucleocytoplasmic transport

The genome-wide investigation highlighted that PGC-1α regulates the mRNA nuclear export of transcripts expressed from canonical gene targets as well as from non-canonical and new targets. We have

**Fig. 6 | The mRNA nuclear export function of PGC-1α is essential to its cellular function and mitochondrial homeostasis. a–d** Cell lines were induced with doxycycline for 72 h and cultured in glucose prior to switching or not in galactose media for the last 24 h. OXPHOS western blots of complex I, II, III & IV protein levels of Ctrl-RNAi and PGC-1α RNAi (**a**) or of PGC-1α WT and PGC-1α ΔRS (**b**) cell lines. OXPHOS western blots shown in panels (**a**) and (**b**) were performed in three independent experiments and changes in COX2 levels quantified and normalised against the total mitochondrial protein level (the sum of all 5 complexes) (Mean ± SEM; two-way ANOVA with Sidak's correction for multiple comparisons; NS: not significant, $**p = 0.0013$ (**c**), $**p = 0.0024$ (**d**); $N = 3$). **e–j** Functional mitochondrial assays. Sham, Ctrl-RNAi, PGC-1α-RNAi, WT-res and ΔRS-res cell lines were induced with doxycycline for 6 days and cultured in glucose prior to galactose switch for the last 24 h. **e–g** Mitochondrial respiratory chain complex II (**e**), IV (**f**) and I (**g**) enzymatic activity were performed in three independent experiments (Mean ± SEM; one-way ANOVA with Turkey's correction for multiple comparisons; NS: not significant $*p < 0.05$, $**p < 0.01$, $***p < 0.001$; $N = 3$). **h** Genomic mitochondrial *MT-ND1* DNA levels were performed in five independent experiments and analysed by qPCR following normalisation to the nuclear *B2M* gene and to 100% of Sham cell line (Mean ± SEM; one-way ANOVA with Tukey's correction for multiple comparisons; NS: not significant, $*p = 0.0334$, $**p = 0.0026$; $N = 5$). **i, j** MTT cell proliferation assays were performed in glucose or after a 24 h galactose induction in five independent experiments (Mean ± SEM; one-way ANOVA with Tukey's correction for multiple comparisons; NS: not significant, $**p < 0.02$, $***p < 0.001$; $N = 5$). For panels **c–j**, source data are provided with details of statistical tests and exact *p*-values as a Source data file.

functionally validated some canonical mitochondrial-related targets in Fig. 5. We next aimed to validate some of the novel predicted bioinformatics findings regarding nuclear export of mRNAs implicated in age-related maintenance of telomere, the nucleocytoplasmic transport and the mitochondrial glutaminase. RNA immunoprecipitation assays conducted in induced Sham, WT-res and ΔRS-res cells further showed that PGC-1α specifically binds *GLS*, *XRCC5*, *XRCC6*, *NHEJ1*, *DDX19B*, *KPNA2*, *NUP58*, *NUP62*, *NUP98* and *XPO1* mRNAs via the RS domain (Fig. 9a, b). Moreover, while total mRNA expression levels are unchanged, the depletion of endogenous PGC-1α and expression of PGC-1α-ΔRS led to concomitant nuclear accumulation and cytoplasmic decrease (Fig. 9c), demonstrating that the NXF1/RNA-binding domain of PGC-1α is involved in the nuclear export of these transcripts. The functional relevance of the PGC-1α-dependent mRNA nuclear export findings were next examined by western blots at protein expression levels for the GLS and NHEJ-related proteins. Accordingly, GLS, XRCC5, XRCC6 and NHEJ1 exhibit reduced expression levels in the induced ΔRS-res cell line but not in Sham and WT-res control conditions (Fig. 9d; quantification in Fig. 9e). Taken together, these results show that the mRNA nuclear export activity of PGC-1α controls the gene expression of novel metabolic and age-related PGC-1α-dependent target genes that were identified in the genome-wide investigation. They provide new mechanistic insights to previous studies reporting that increased and decreased level/activity of PGC-1α are associated with longevity and pre-mature ageing, respectively[83–85], as well as with cell survival during stress[86], linking thus mitochondrial function and lifespan to stress-resistance.

## Discussion

In this study, we characterised the RNA-binding function of the human master energy homeostasis regulator PGC-1α, showing that its RS domain regulates the mRNA nuclear export pathway and exhibits the hallmarks of mRNA nuclear export adaptors by directly interacting with RNA and the nuclear export receptor NXF1. Functional assays further demonstrated that the novel mRNA nuclear export activity is essential for PGC-1α's known function in mitochondrial metabolism homeostasis and cell growth. A genome-wide investigation identified >1200 mRNA nuclear export targets encoding proteins involved in both known and novel pathways. Canonical processes such as mitochondrial, metabolic, AMPK and insulin-related pathways as well as regulation of RNA polymerase II transcription and longevity, involves both the promoter-binding and mRNA nuclear export functions of PGC-1α, while thermogenesis and oxidative phosphorylation appear to be restricted to its binding to promoters. On the other hand, the cytoplasmic abundance of non-canonical mRNAs involved in glucagon signalling, translation initiation, nucleocytoplasmic transport and NHEJ-dependent maintenance of telomere is specifically regulated by the nuclear export function of PGC-1α. A model summarising the recognised and novel functions of PGC-1α is presented in Fig. 10. Consistently, the glucagon-signalling pathway was shown to be enriched in the study which identified non-canonical RNA-binding targets of PGC-1α[46] while proteins interacting with the C-terminal domain of PGC-1α were recently showed to depend on RNA-binding and be enriched in mRNA processing and mRNA transport pathways[21]. In addition, we also showed that the mRNA nuclear export function of PGC-1α plays a significant contribution at genome-wide level by regulating 94% and 52% of the down-regulated cytoplasmic transcriptome and proteome changes, respectively (Fig. 8c, g).

Our genome-wide investigation also found 691 mRNAs with predominantly altered tandem alternative polyadenylation and transcription start sites (Fig. 8d). However, it remains unclear whether the RNA-binding function of PGC-1α plays a direct role in pre-mRNA splicing since the expression levels of mRNAs encoding 30 splicing factors were also identified to be downregulated upon loss of the RS domain, which also promotes transcriptional co-activation from some promoters via interaction with the MED1 subunit of the RNA polymerase II-associated Mediator/p300 complex[18]. Interestingly, the binding of PGC-1α is greatly increased when the RS domain is deleted (Fig. 7b), suggesting that the RS domain play a regulatory role in the binding of PGC-1α to promoters, potentially via its interactions with other proteins and/or RNA. Reciprocally, other proteins involved in binding the RS domain of PGC-1α may also contribute to the recruitment of PGC-1α to transcripts undergoing processing. The total expression levels of only 465 mRNAs were found to be reduced in the ΔRS-res cells (Supplementary Data 4 tab 1), which may be accounted by the partial rather than full depletion of endogenous PGC-1α (-70% at mRNA and protein level, Fig. 5e, h) and by HEK293T cells not necessarily relying on mitochondrial respiration for growth. It, however, validates the key role of the nuclear export function of PGC-1α which cannot be compensated in this model whilst the co-transcriptional activation function still takes place, presumably through interaction of other canonical co-activators to PGC-1α-regulated promoters.

PGC-1α interacts with both intronic and exonic RNA sequences[21,46], thus placing PGC-1α in an ideal position to further promote gene expression through co-transcriptional recruitment and coupling to mRNA nuclear export[87,88]. In humans, the bulk nuclear export of mRNAs is primarily linked to splicing-dependent recruitment of the Transcription-Export (TREX) complex[39–41]. On the other hand, SR-rich splicing factors SRSF1,3,7, which share homology with PGC-1α carboxyl-terminal domain, also promote the nuclear export of a subset of transcripts[33–37,42] independently of the TREX complex[89]. We found that the nuclear export function of PGC-1α is either linked or independent of its binding to promoters, suggesting that it could be recruited to nascent transcripts via at least 2 different mechanisms which remain to be identified in future studies. Interestingly, the yeast TREX complex, which predominantly couples transcription elongation to nuclear mRNA export[90,91], recruits SRSF-like Gbp2 and Hbr1 proteins during transcriptional elongation[92], while the general mRNA nuclear export adaptor Aly/REF, a subunit of TREX, was initially characterised as ALY (Ally of AML-1 and LEF-1) and BEF (bZIP enhancing

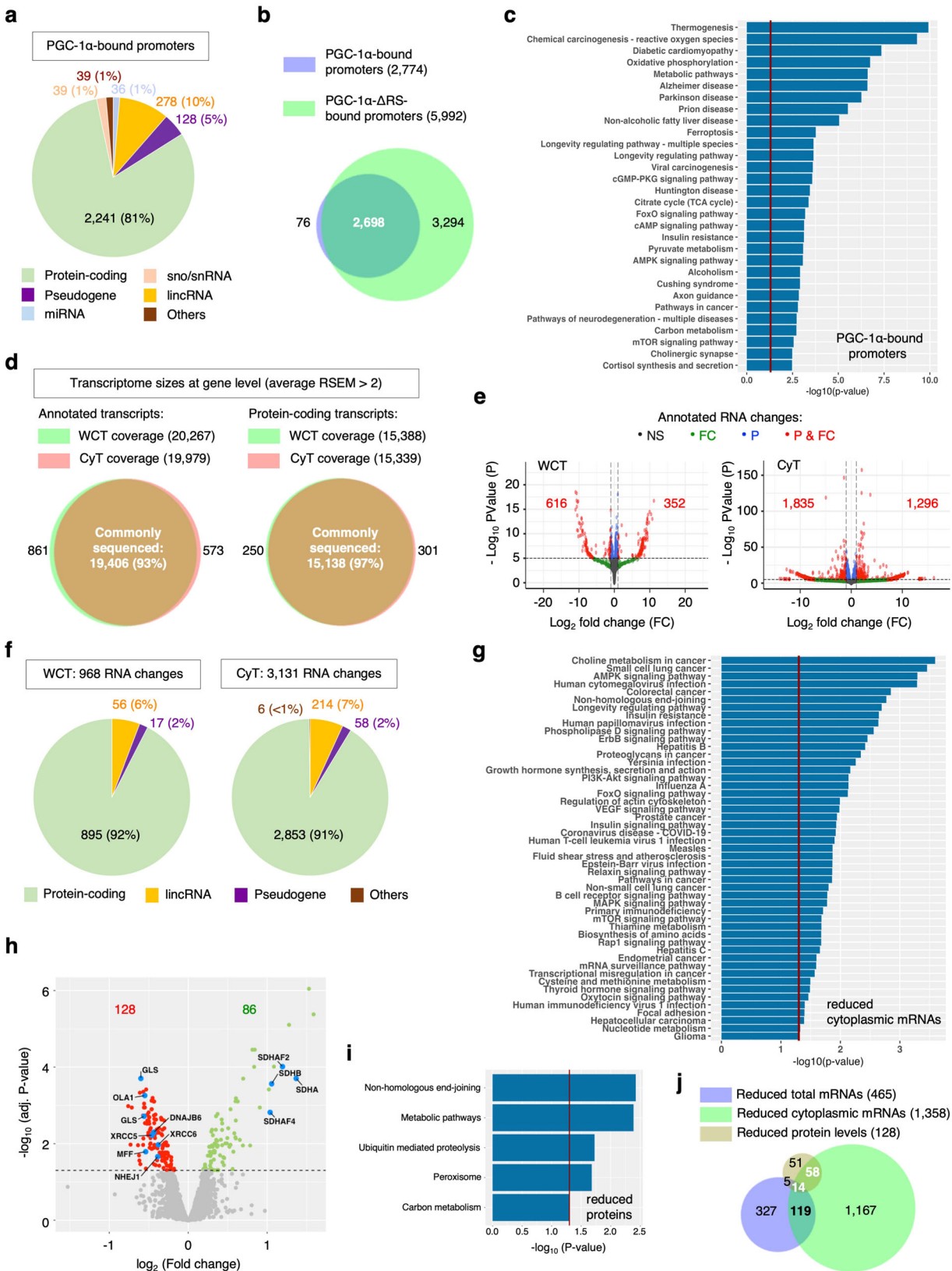

factor) in studies reporting that it stimulates transcriptional activation by enhancing the binding of some transcription factors to DNA[93,94]. The discovery of this novel biological function for a key cellular homeostasis regulator is anticipated to provide a stepping stone to progress PGC-1α research into fields covering gene expression, metabolic disorders, ageing and neurodegenerative diseases.

## Methods

### Plasmids

All plasmids used in this study are described in Supplementary Table 1. RNA interference plasmids were built using annealed DNA oligonucleotides (Supplementary Note 1) in the pcDNA6.2-GW/EmGFP plasmid (Block-iT™ miRNA expression kit from Invitrogen). Specific

**Fig. 7 | Genome-wide contribution of RS-mediated functions of PGC-1α at promoter-binding, transcriptome and proteome levels. a** Pie chart representing the relative distribution and diversity of promoters bound by PGC-1α. **b** Venn diagram representing promoters bound by PGC-1α and PGC-1α-ΔRS. **c** Bar charts representing KEGG pathways enriched in the promoters bound by PGC-1α. The statistical analysis uses a modified Fisher's Exact test within the DAVID package[82]. **d** Combined sizes of whole-cell transcriptomes (WCT) and cytoplasmic transcriptomes (CyT) for quantified annotated and protein-coding transcripts at gene level. **e** Volcano plots representing the genome-wide distribution of differentially-expressed transcripts according to p-values and fold changes upon loss-of-function of the RS domain (ΔRS-res versus WT-res). NS (black): non-significant; FC (green): fold changes >1.41, p-values > 0.05; P (blue): p-values < 0.05, fold changes <1.41; P & FC (red): significant p-values (<0.05) & fold changes (>1.41). Red labels indicate the numbers of significantly down- or up-regulated annotated transcripts in the whole cell and cytoplasmic transcriptomes. The DESeq2 package was used to statistically estimate the differential expression (Supplementary Method 6). **f** Pie chart representing annotated transcripts differentially expressed upon loss of the RS domain of PGC-1α in the whole cell and cytoplasmic transcriptomes. **g** Bar charts representing KEGG pathways enriched cytoplasmic mRNAs down-regulated upon loss-of-function of the RS domain. The statistical analysis uses a modified Fisher's Exact test within the DAVID package[82]. **h** Volcano plot representing differentially expressed proteins with adjusted p-values < 0.05. The LIMMA package was used to statistically estimate the differential expression (Supplementary Method 7). **i** Bar chart representing KEGG pathways enriched in proteins down-regulated upon loss-of-function of the RS domain. The statistical analysis uses a modified Fisher's Exact test within the DAVID ontology package[82]. **j** Venn diagram comparing the lists of down-regulated total/cytoplasmic mRNAs and proteins upon loss-of-function of the RS domain.

primers for pre-miRNA were designed using Life Technologies' online tool (www.lifetechnologies.com/rnai). Single miRNAs were generated following manufacturer's instructions (Block-iT; LifeTechnologies). For the chaining, a single miRNA was removed from one of the plasmids by double restriction digest and inserted into the other plasmid containing another miRNA. miRNAs were initially chained in pcDNA6.2-GW/EmGFP and the miRNA cassettes were subsequently amplified and cloned into the 3'UTR of an engineered pcDNA5-FRT/3xFLAG-PGC-1α plasmid.

## Cell culture and transfection

HEK293T and Hela cell lines were seeded in 10 cm dishes ($2 \times 10^6$ cells) or 24-well plates (50,000 cells/well) in media (DMEM High Glucose, 10% FBS (Sigma), 1% PenStrep). 24 h post plating, cells were transfected with 3.5 μg polyethylenimine (PEI)/ml media. 15 μg or 500–700 ng total plasmid DNA were, respectively, transfected in 10 cm or in each well of a 24-well plate. HEK293T-FlpIn cell lines were cultured in media (DMEM High Glucose, 10% Tet-Free FBS (Sigma), 1% PenStrep) in T175 flasks ($5 \times 10^6$ cells), 10 cm dishes ($2 \times 10^6$ cells), 6-well plates (200,000 cells/well) or 24-well plates (50,000 cells/well). For galactose experiments, cells were changed into galactose media 24 h prior to experiment using DMEM containing no glucose (Gibco™) supplemented with 10% Tet-Free FBS, 1% PenStrep and 5 mM galactose.

## RNA-protein UV cross-linking assay

1.5 μg RNA probes (5xAAAAUU or 5xGGGGCC) (Dharmacon) were end-radiolabelled with γ[$^{32}$P]-ATP (PerkinElmer) using T4 polynucleotide kinase (PNK, NEB). Reactions were set up by following the manufacturer's instructions. For the protein-RNA-binding assay, 600 ng purified recombinant proteins were incubated in 1x RNA Binding Buffer (15 mM HEPES pH 7.9, 100 mM NaCl, 5 mM MgCl$_2$, 0.05% v/v Tween 20, 10% v/v glycerol) with ~20 nmol radiolabelled RNA for 20 min at room temperature followed by 20 min on ice prior to UV-irradiation for 10 min at 1.5 J/cm$^2$. The reaction was terminated with 4x Laemmli Sample Buffer and heating at 95 °C for 5 min. The RNA-protein binding complex was resolved on SDS–PAGE prior to analysis by Coomassie staining and PhosphoImaging (Typhoon FLA 7000, GE Healthcare).

## mRNP capture assay

HEK293T cells were seeded in 10 cm dishes (2 per condition) and transfected with 3xFLAG-PGC-1α WT or ΔRS. Forty-eight hours post transfection, 1 dish for each condition was UV-cross-linked on ice at 0.3 J/cm$^2$ in 1 ml DEPC-treated PBS. All plates were then washed with ice cold DEPC PBS and lysed in 500 μl lysis buffer (50 mM Tris-HCl pH 7.5, 100 mM NaCl, 2 mM MgCl$_2$, 1 mM EDTA, 0.5% v/v Igepal C9-630, 0.5% w/v Na-deoxycholate) supplemented with fresh Complete protease inhibitor cocktail (Roche) and 20U Ribosafe (Bioline) on ice for 5 min. Samples were centrifuged for 10 min at $17,000 \times g$ at 4 °C. Cleared lysates were diluted to 2 mg/ml with lysis buffer, input samples

kept, and diluted with equal volume 2x Binding Buffer (20 mM Tris-HCl pH 7.5, 1 M NaCl, 1% SDS, 0.2 mM EDTA). Lysates were incubated with 25 μg oligo-dT beads rotating end-over-end at room temperature for 1 h. Beads were washed 3 times with 1x Binding Buffer and mRNP complexes were eluted in 50 μl Elution Buffer (10 mM Tris-HCl pH 7.5, 1 mM EDTA) supplemented with Complete protease inhibitor and 10 μg RNase A (Sigma) at 37 °C for 30 min with gentle agitation. Inputs and eluates were subjected to western blotting.

## RNA immunoprecipitation (RIP) assays

HEK FlpIn cell lines Sham, PGC-1α WT-res and PGC-1α ΔRS-res were cultured in either $2 \times 10$ cm dishes (UV) or T175 flasks (formaldehyde). Cells were either UV-cross-linked on ice with UV-C at 0.3 J/cm$^2$ in 1 ml DEPC-treated PBS or 1% formaldehyde was added to the medium of live cells for 10 min and subsequently quenched with 250 mM Glycine for 5 min at room temperature. Cells were washed with DEPC-treated PBS prior to being scraped into ice-cold RNase-free IP150 lysis buffer (DEPC-treated water containing 50 mM HEPES pH 7.5, 150 mM NaCl, 10% glycerol, 0.5% Triton X-100, 1 mM EDTA, 1 mM DTT, 1 μl RNase inhibitor, protease inhibitors). Cells were passed through a 21 G needle 10 times and lysed on ice for 10 min, followed by centrifugation at $17,000 \times g$ at 4 °C for 5 min and quantification using Bradford Reagent. 5 mg (UV) or 2 mg (formaldehyde) of total protein at a 1 mg/ml concentration was incubated with 40 μl pre-blocked (1% BSA and 5 μl/ml ssDNA) anti-FLAG-M2 affinity resin (Merck) for either 2 h (UV) or overnight (formaldehyde) at 4 °C rotating. Beads were washed 5 times with RNase-free lysis buffer followed by elution with 50 μl RNase-free lysis buffer supplemented with 100 μg/ml 3xFLAG peptide (Sigma #F4799) for 30 min at 4 °C rotating. The formaldehyde crosslinks were reversed by heating the samples for 1 h at 70 °C and RNA was extracted using PureZOL™ (BioRAD) as described in the RNA extraction section. Extracted RNA samples were re-suspended in 25 μl (input) or 15 μl (eluate) RNase-free water.

## GST-NXF1:p15 pull-down assays

BL21-RP cells co-transformed with pGEX4T1/GST-NXF1 and pET9a/p15 were induced overnight at 18 °C with 0.4 mM isopropyl-β-D-thiogalactoside (IPTG, Calbiochem) in Terrific Broth. Bacterial pellets (0.25 g) containing IPTG induced GST-tagged NXF1-p15 protein complex were lysed in RB-100 (25 mM HEPES pH 7.5, 100 mM potassium acetate, 10 mM MgCl$_2$, 1 mM DTT, 10% Glycerol and 0.5% Triton x-100), the soluble proteins were collected after a 5 min centrifuge at $17,000 \times g$ at 4 °C and immobilised on Glutathione-coated sepharose beads (GE Healthcare). For pull down with $^{35}$S-radiolabelled proteins, PGC-1α full-length and protein domain were translated in vitro in the T7 transcription and translation coupled reticulocyte system (Promega) supplemented with $^{35}$S-Methionine. $^{35}$S-radiolabled proteins were incubated with the immobilized GST-NXF1-p15 protein complex in RB-100 buffer with or without RNase A (20 ng/μl) at 4 °C for 1 h. GST-NXF1:p15 bound protein complexes were eluted in GSH elution

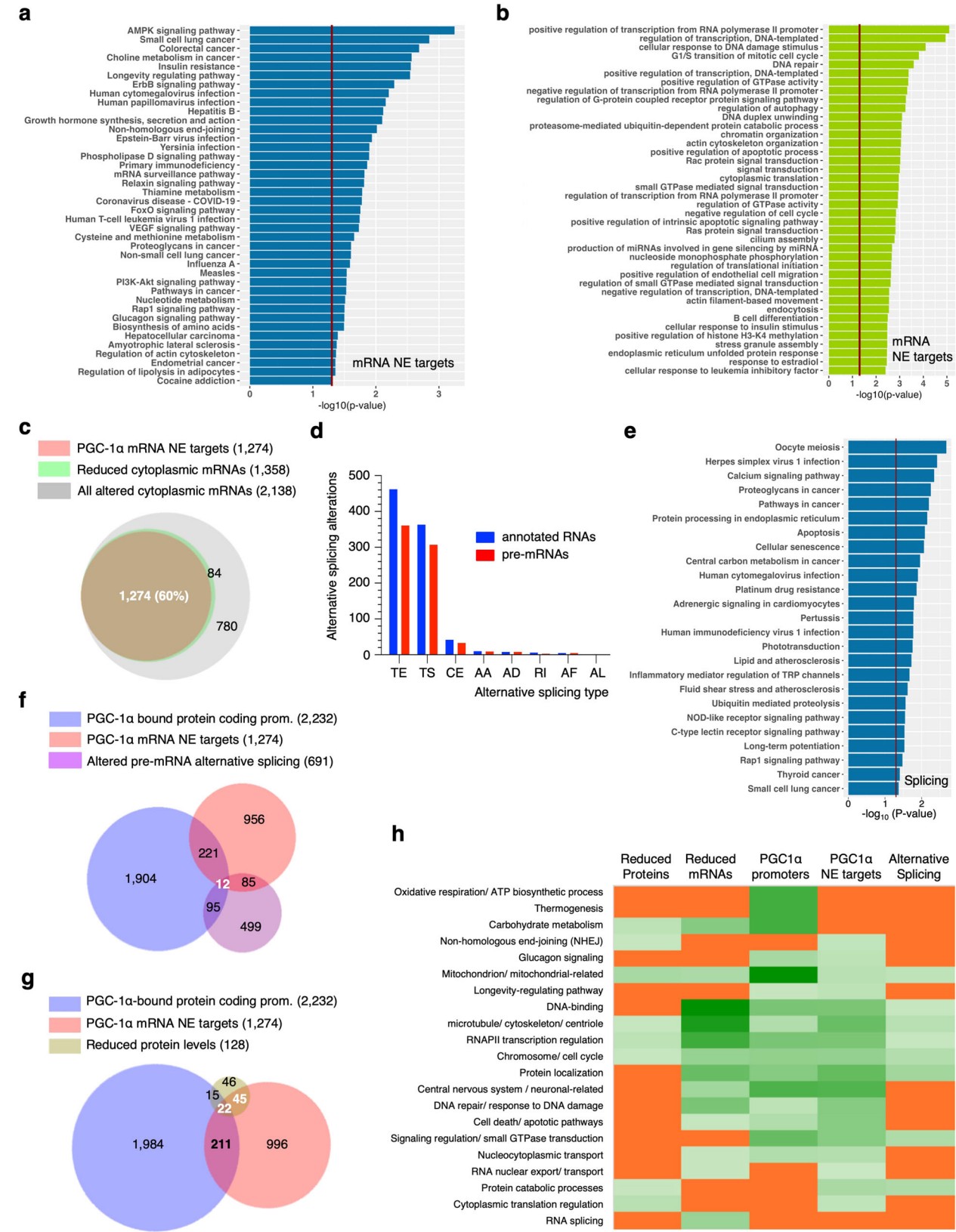

buffer (50 mM Tris, 100 mM NaCl, 40 mM reduced glutathione, pH 7.5) after 5x wash with binding buffer. The protein binding complex was separated on SDS–PAGE prior to analysis by Coomassie staining and Phosphoimaging (Typhoon FLA 7000, GE Healthcare). For pull down with purified PGC-1α protein domains, 25 µg purified recombinant proteins were added to the GST-NXF1:p15 binding reactions prior to incubation, wash and elution as highlighted for the pull down with the $^{35}$S-radiolabelled proteins. Co-purification of 6His-tagged PGC-1α protein domains was detected by western blot using 6xHis antibody.

**Fig. 8 | Genome-wide relevance of the splicing and mRNA nuclear export function of PGC-1α. a** Bar charts representing KEGG pathways enriched in the list of PGC-1α-dependent mRNA nuclear export (NE) targets. The statistical analysis uses a modified Fisher's Exact test within the DAVID ontology package[82]. **b** Bar charts representing Biological Processes (BP Direct) enriched in the list of PGC-1α-dependent mRNA nuclear export targets. The statistical analysis uses a modified Fisher's Exact test within the DAVID ontology package[82]. **c** Venn diagram comparing the lists of PGC-1α ΔRS-dependent down-regulation of cytoplasmic mRNAs, up- and down-regulation of cytoplasmic mRNAs (all altered) and PGC-1α mRNA nuclear export targets. **d** Bar chart representing the number of alternative spliced events for annotated transcripts and pre-mRNAs. The following alternative splicing types were identified: TE Tandem alternative polyadenylation site, TS Tandem transcription start site, CE Cassette exon, AA Alternative Acceptor splice site, AD Alternative Donor splice site, RI Retained intron, AF Alternative First exon, AL Alternative Last exon. DESeq2 was used within the Whippet package[81] to statistically estimate the differential expression (Supplementary Method 6). **e** Bar charts representing KEGG pathways enriched in the list of pre-mRNAs with altered alternative splicing upon loss-of-function of the RS domain. The statistical analysis uses a modified Fisher's Exact test within the DAVID ontology package[82]. **f** Venn diagram comparing the lists of PGC-1α-bound protein-coding promoters, PGC-1α-dependent mRNA nuclear export targets and RS domain-mediated alteration of pre-mRNA splicing. **g** Venn diagram comparing the lists of PGC-1α-bound protein-coding promoters, PGC-1α-dependent mRNA nuclear export targets and RS domain-dependent down-regulation of proteins. **h** Heat map comparing the enrichment scores of major pathways for the RS-mediated down-regulation of proteins/mRNAs levels and altered splicing with the PGC-1α-dependent mRNA nuclear export targets and PGC-1α-bound protein-coding promoters. A scale of light to dark green depicts pathways with increased enrichment scores while orange highlights those not identified in the gene ontology analysis (Supplementary Data 6).

## Co-immunoprecipitation assays

Two million HEK293T cells were seeded in 10 cm dishes ($2 \times 10^6$ cells/dish; 4 dishes per condition) and transfected with 3 μg of 13xMyc-tagged NXF1 and either 3 μg 3xFLAG or 3 μg 3xFLAG-PGC-1α plasmids using 18 μl XtremeGene 9 (Sigma). Forty-eight hours post transfection, cells were washed with ice-cold PBS and lysed in 500 μl IP150 lysis buffer (50 mM HEPES pH 7.5, 150 mM NaCl, 1 mM EDTA, 0.5 mM DTT, 0.5% v/v Triton X-100) supplemented with protease inhibitor cocktail (cOmplete, Roche) and 0.5 mM PMSF. Lysates were sonicated gently on ice and incubated rotating at 4 °C for 10 min. Samples were centrifuged for 2 min at $21,000 \times g$ at 4 °C, supernatant was transferred to a new tube and further centrifuged for 5 min. Cleared lysates were diluted to 2 mg/ml and 3 ml was pre-cleared with 200 μl slurry mouse IgG-Agarose in the presence of 10 μg/ml RNase A for 1 h at 4 °C rotating. Pre-cleared supernatant was then incubated with 30 μl slurry pre-blocked with 2% BSA overnight for 1.5 h at 4 °C rotating. Beads were collected and transferred to a new tube, washed three times with lysis buffer before being transferred to a new tube and washed one last time (four washes in total). Co-immunoprecipitated proteins were eluted with 60 μl of 300 μg/ml 3xFLAG peptide (Sigma) rotating for 1 h at room temperature. Fifty microlitres were then transferred to a new tube and incubated with 4X sample buffer (with DTT) at 95 °C for 5 min. Co-immunoprecipitated protein complexes and input samples were subjected to SDS–PAGE using 15-well, 1.5 mm, 4–12% Bis-Tris precast gels (ThermoFisher). Proteins were transferred onto PVDF membrane using BioRad wet transfer system, incubated in 5% milk in TBST for 1 h at room temperature followed by primary antibody incubation overnight at 4 °C. Membranes were washed with TBST, incubated with secondary antibody, washed again and imaged using ImageQuant 800 (Cytiva).

## Immunocytochemistry, RNA-FISH and proximity ligation assay (PLA)

ICC, RNA-FISH and PLA were performed as described[95] with small modifications. Briefly, HeLa cells were fixed on coverslips with 4% paraformaldehyde and soaked in 75% ethanol overnight. For ICC, coverslips were incubated in a mixture of anti-PGC-1α (mouse monoclonal clone 4C1.3, *Merck* ST1202) and anti-NXF1 (rabbit polyclonal clone D5X4G, Cell Signaling #12735) antibodies (both 1:1000) in blocking solution overnight at 4 °C, followed by secondary Alexa488- or Alexa546-conjugated antibody incubation for 1 h at RT. For RNA-FISH, commercially available Biosearch Technologies Stellaris® Quasar570-labelled probes were used (*NEAT1*, SMF-2037-1 and *MALAT1*, SMF-2035-1). After washing off the probe with PBS, cells were incubated in anti-PGC-1α primary antibody in PBS for 1 h and in Alexa488-conjugated antibody for 30 min. PLA was performed using Duolink® In Situ Orange Starter Kit Mouse/Rabbit (DUO92102, Sigma) using the same PGC-1α/NXF1 antibody combination as for ICC or using the same anti-PGC-1α antibody in combination with anti-NUP107

antibody (1:1000, rabbit polyclonal Proteintech 19217-1-AP or Invitrogen PA5-30774). The specificity of all antibodies used for PLA was verified by ICC prior to these experiments. Images were taken with a 100× objective on Olympus BX57 fluorescent microscope equipped with ORCA-Flash 4.0 camera (Hamamatsu) and cellSens Dimension software (Olympus). Panels were prepared using Image J and Photoshop CS3.

## Oligo-dT fluorescent in situ hybridization (FISH)

HEK293T cells were seeded in 24-well plates and transfected with 3xFLAG-e1BAP5, 3xFLAG-Aly/REF or 3xFLAG-PGC-1α. Forty-eight hours post transfection, cells were treated with 5 μg/ml Actinomycin D for 2 h followed by 3 washes with PBS and incubation in fix solution (4% PFA, 0.2% Triton X-100) for 30 min at room temperature. Cells were then washed 3 times with PBS and coverslips were transferred onto a Whatman paper in a petri dish. Each coverslip was incubated with 80 μl hybridization mix (20% Formamide, 2X SSC, 10% Dextran sulphate, 1% BSA, 4 μl ssDNA f, 1 μg Cy3-oligo-dT probe) for 2 h in the dark at 37 °C. Coverslips were then washed 3 times with PBS and blocked with 2% BSA in PBS for 20 min at room temperature. Cells were stained using anti-3xFLAG primary, AlexaFluor 488 secondary antibody and Hoechst stain. Mounted coverslips were imaged using Nikon LV100ND microscope with DS Ri1 Eclipse camera and images were processed using ImageJ software package.

## MTT cell proliferation assay

MTT cell proliferation assay. HEK FlpIn cell lines (Sham, Control-RNAi, PGC-1α-RNAi, PGC-1α WT-res and PGC-1α ΔRS-res) were cultured in 24-well plates. Each plate contained 4 wells with only media to serve as a blank and 4 wells/cell line. For MTT assay, 250 mg Thiazolyl Blue Tetrazolum Bromide reagent (MTT) was added to each well and incubated in the dark at 37 °C for 1 h. Cells were subsequently lysed with equal volume MTT lysis buffer (20% SDS, 50% Dimethylformamide (DMF)) and incubated, shaking, at room temperature for 1 h. Absorbance at 595 nm was assessed with a PHERAstar FS (BMG Labtech) and data retrieved using PHERAstar MARS (BMG Labtech). Percentage cell proliferation was calculated for each cell line relative to Sham and data analysed using GraphPad Prism (Version 9.0.2.161).

## Nuclear & cytoplasmic fractionation

HEK FlpIn cell lines (Sham, Control-RNAi, PGC-1α-RNAi, PGC-1α WT-res and PGC-1α ΔRS-res) were cultured in 6-well plates. Cytoplasmic fractionation was performed as described[51]. Briefly, cells from 6 wells were collected in DEPC PBS and pelleted by centrifugation at $400 \times g$ for 5 min. Cell pellets were quickly washed with hypotonic lysis buffer (10 mM HEPES pH 7.9, 1.5 mM MgCl$_2$, 10 mM KCl, 0.5 mM DTT) and lysed for 10 min on ice in hypotonic lysis buffer containing 0.16 U/μl RNase inhibitor and protease inhibitors. All lysates underwent

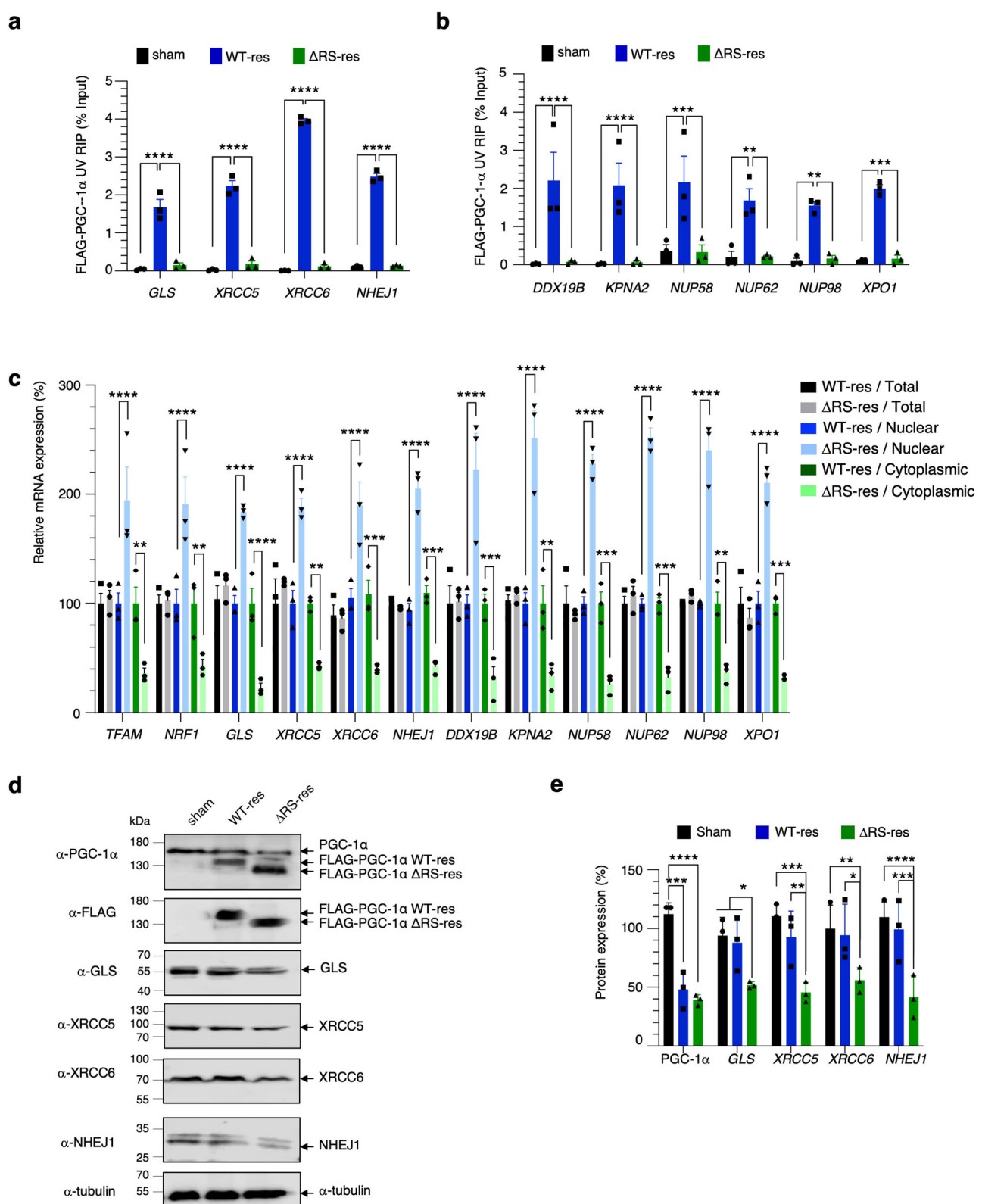

differential centrifugation at $1500 \times g$ (3 min), $3500 \times g$ (8 min), and $17,000 \times g$ (1 min) at 4 °C and the supernatants transferred to fresh tubes after each centrifugation. Nuclear pellets obtained after centrifugation at $1500 \times g$ for 3 min were washed 4 times with hypotonic lysis buffer prior to lysis in Reporter lysis buffer (Promega), passed through a 21G needle, incubated on ice for 10 min and centrifuged at $17,000 \times g$, 5 min, 4 °C. Total fractions were collected from 3 wells of a

6-well plate in Reporter lysis buffer containing 16 U μl⁻¹ RNase inhibitor protease inhibitors and passed through a 21G needle prior to lysis for 10 min on ice before centrifugation at $17,000 \times g$, 5 min, 4 °C. Equal volumes of total, nuclear and cytoplasmic lysates were subjected to western immunoblotting using SSRP1 and class III beta-tubulin antibodies. Total and fractionated extracts were added to PureZOL™ to extract the RNA.

**Fig. 9 | PGC-1α regulates the mRNA nuclear export of non-canonical PGC-1α target transcripts.** Sham, FLAG-tagged PGC-1α WT (WT-res) and FLAG-tagged PGC-1α ΔRS (ΔRS-res) cell lines were induced with doxycycline for 6 days. **a, b** RNA immunoprecipitation (RIP) assays of transcripts predicted to be novel mRNA nuclear export targets of PGC-1α. Cells were subjected to UV-C cross-link prior to anti-FLAG immunoprecipitation. Purified RNA was analysed by qRT-PCR and expressed as % of the input. RIPs were performed in three independent experiments (Mean ± SEM; two-way ANOVA with Turkey's correction for multiple comparisons; **$p < 0.01$, ***$p < 0.001$, ****$p < 0.0001$; $N = 3$). **c** Total, nuclear and cytoplasmic levels of fractionated RNA transcripts were quantified in three

independent experiments by qRT−PCR following normalization to U1 snRNA levels and to 100% in Sham cell line (Mean ± SEM; two-way ANOVA with Tukey's correction for multiple comparisons, NS: not significant, *$p < 0.05$, **$p < 0.01$, ***$p < 0.001$, ****$p < 0.0001$; $N = 3$). **d** Western blots from Sham, WT-res and ΔRS-res cell lines induced with doxycycline for 6 days. Blots were probed for PGC-1α, FLAG, GLS, XRCC5, XRCC6, NHEJ1 and α-tubulin antibodies. **e** Western blots in (**d**) were quantified in three independent experiments (Mean ± SEM; one-way ANOVA with Tukey's correction for multiple comparisons; *$p < 0.05$, **$p < 0.01$, ***$p < 0.001$; $N = 3$). For panels **a**−**c** and **e**, source data are provided with details of statistical tests and exact $p$-values as a Source data file.

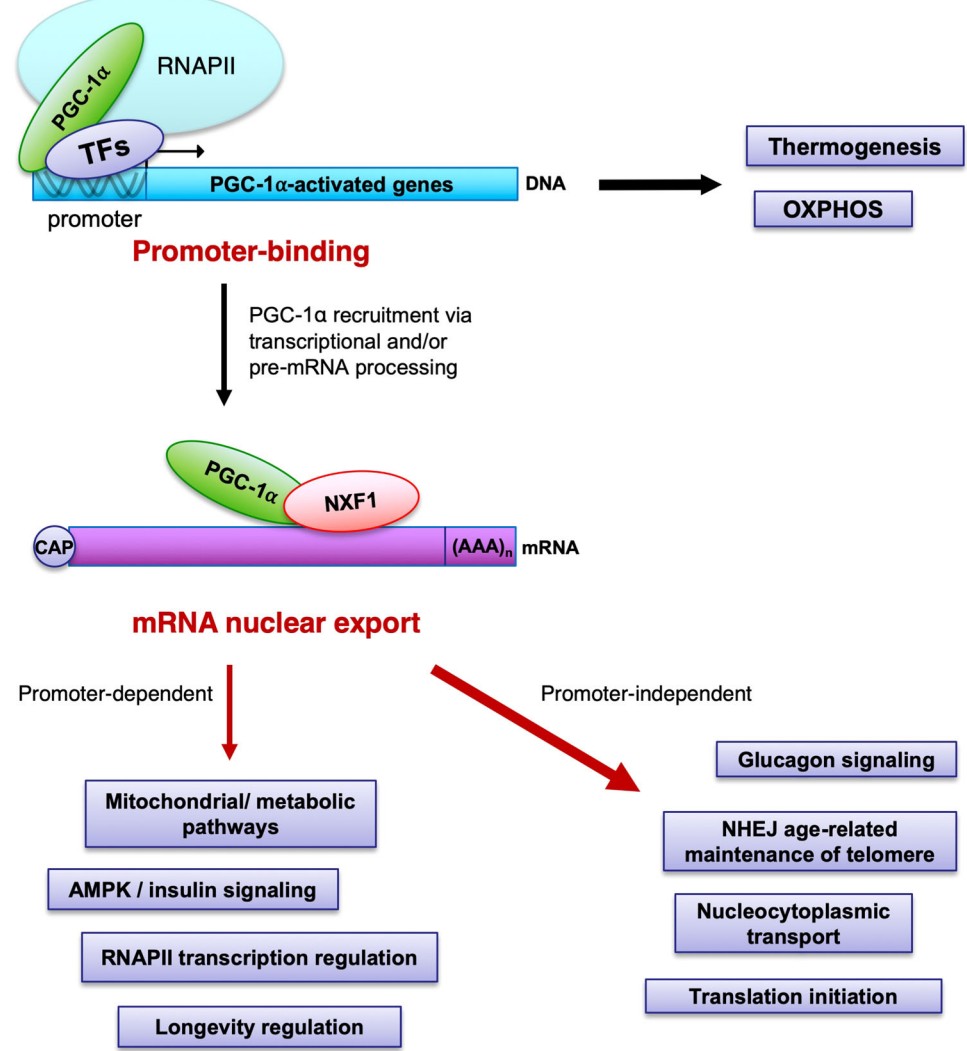

**Fig. 10 | Model integrating the promoter-binding and novel mRNA nuclear export functions of PGC-1α.** The master regulator PGC-1α is well characterised for its role in binding and co-activating several transcription factors (TFs) to stimulate the RNA polymerase II (RNAPII) dependent transcription of target genes known to regulate the mitochondrial and energy metabolism homeostasis. PGC-1α also regulates the NXF1-dependent nuclear export of canonical and non-canonical mRNAs into the cytoplasm in a process which is linked to or independent of its binding to promoters. Pathways regulated by the different promoter-binding and mRNA nuclear export functions of PGC-1α are indicated in purple-labelled boxes.

## SDS−PAGE and western blotting

HEK FlpIn cell lines (Sham, Control-RNAi, PGC-1α-RNAi, PGC-1α WT-res and PGC-1α ΔRS-res) were cultured in 6-well plates. Cells were washed in ice-cold PBS, scraped into ice-cold IP150 lysis buffer and lysed on ice for 10 min followed by centrifugation at $17,000 \times g$ at 4 °C for 5 min. Protein extracts were quantified using Bradford Reagent (BioRAD), resolved by SDS−PAGE, electroblotted onto nitrocellulose membrane and probed using the relevant primary antibody. All primary and secondary antibodies used in this study are detailed in Supplementary

Table 2. Images of full-size western blots are provided as a Source data file.

## RNA extraction and qRT-PCR

PureZOL™ was added to lysates as per manufacturers recommendations and cleared by centrifugation for 10 min at $12,000 \times g$, 4 °C. For RNA-seq, RNA was extracted from total and cytoplasmic fraction using the Direct-zol RNA miniprep kit as per the manufacturer's protocol, including the optional on-column DNA digestion step. For qRT-PCR,

samples were incubated at room temperature for 10 min followed by the addition of 1/5th volume of chloroform and vigorous shaking for 15 s. After a further 10 min incubation at room temperature, samples were centrifuged at $12,000 \times g$ for 10 min (4 °C) and the upper phase collected. RNA was precipitated overnight at −20 °C with equal volume isopropanol, 1/10th 3 M NaOAc, pH 5.2 and 1 µl Glycogen (5 µg/µl, Ambion) and subsequently pelleted at $12,000 \times g$ for 20 min (4 °C). Pellets were washed with 70% DEPC ethanol and re-suspended in DEPC water. All extracted RNA samples were treated with DNaseI (Roche) and quantified using a Nanodrop (NanoDrop Technologies). Following quantification, 2 µg RNA (fractionation) or 11 µl RNA (RNA immunoprecipitation) was converted to cDNA using BioScript Reverse Transcriptase (Bioline). Primers used in this study are provided in Supplementary Table 3. qPCR reactions were performed in duplicate using the Brilliant III Ultra-Fast SYBR Green QPCR Master Mix (Agilent Technologies) on a C1000 Touch™ thermal cycler using the CFX96™ Real-Time System (BioRAD) using an initial denaturation step, 45 cycles of amplification (95 °C for 30 s; 60 °C for 30 s; 72 °C for 1 min) prior to recording melting curves. qRT–PCR data was analysed using CFX Manager™ software (Version 3.1) (BioRAD) and GraphPad Prism (Version 9.0.2.161).

## Mitochondrial DNA quantification

HEK FlpIn cell lines (Sham, Control-RNAi, PGC-1α-RNAi, PGC-1α WT-res and PGC-1α ΔRS-res) were cultured in 6-well plates. Genomic DNA was extracted using the DNeasy Blood and Tissue DNA extraction kit (Qiagen) as per the manufacturer's protocol. Extracted DNA was quantified using a Nanodrop (NanoDropTechnologies). Mitochondrial DNA levels were assessed by qPCR using 100 ng genomic DNA and 300 nM primers (Supplementary Table 3) for the mitochondrial gene *MT-ND1* and the nuclear-encoded gene β-2 microglobulin *B2M*[67,68]. Reactions were performed in duplicate using the Brilliant III Ultra-Fast SYBR Green QPCR Master Mix (Agilent Technologies) on a C1000 Touch™ thermal cycler using the CFX96™ Real-Time System (BioRAD). Amplification conditions were: an initial denaturation step of 95 °C for 10 min followed by 40 cycles of amplification (95 °C for 15 s; 60 °C for 1 min; 72 °C for 1 min) prior to recording melting curves. qRT–PCR data was analysed using CFX Manager™ software (Version 3.1) (BioRAD) and GraphPad Prism (Version 9.0.2.161).

## Activity of mitochondrial respiratory chain complexes I, II and IV

HEK FlpIn cell (Sham, Control-RNAi, PGC-1α-RNAi, PGC-1α WT-res and PGC-1α ΔRS-res) were cultured in 10 cm plates. The activity of the mitochondrial complexes I, II and IV were measured using microplate activity assays (Abcam ab109721, ab109908, ab10990). Assays were performed according to the manufacturer's instructions using 20–35 µg protein per well. Activity reading were normalised to protein content. Sample protein concentrations were measured using Pierce BCA assay.

## Statistical analysis

One-way and two-way ANOVA (analysis of variance) with the recommended Sidak's or Tukey's corrections for multiple comparison tests (GraphPad Prism) were used for all normally-distributed data which involved more than 2 experimental conditions or groups. Quantification of PGC-1α and PGC-1α-ΔRS co-immunoprecipitation with NXF1 used unpair two-tailed Student $t$-test to compare 2 groups (assuming both groups have same standard deviation of the mean). Data were plotted using GraphPad Prism versions 8-9. Significance is indicated as follows; NS: non-significant, $p \geq 0.05$; *$p < 0.05$; **$p < 0.01$; ***$p < 0.001$; ****$p < 0.0001$.

## Reporting summary

Further information on research design is available in the Nature Portfolio Reporting Summary linked to this article.

## Data availability

The data supporting the findings of this study are available from the corresponding authors upon reasonable request. The ChIP-seq and RNA-seq data have been deposited in Gene Expression Omnibus under accession number GSE230429. The TMT mass spectrometry data was deposited in the ProteomeXchange Consortium via the PRIDE partner repository with the dataset identifier PXD031189. Source data for the figures and supplementary figures are provided as a Source data file. Source data are provided with this paper.

## Code availability

The scripts used in the DNA sequencing, RNA-seq and mass spectrometry investigations are provided in the Supplementary Information file (Supplementary Method 8).

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

## Acknowledgements

G.M.H. acknowledges support from a Faculty of Medicine, Health and Dentistry (FMDH) Research and Innovation Award (312366) from the University of Sheffield, the Royal Society research grant RG140690, the Medical Research Council (MRC) New Investigator research grant MR/R024162/1 and the Biotechnology and Biological Sciences Research Council (BBSRC) grant BB/S005277/1 over the course of this study. S.R.M. was supported by a FMDH PhD studentship funded by the University of Sheffield. A.G. was supported by a Governmental Turkish Doctoral Scholarship. N.S. is currently supported by a White Rose BBSRC iCASE Doctoral Training Programme in Mechanistic Biology (2594369) in collaboration with the pharmaceutical company *Nanna Therapeutics* (G.M.H., H.M.). H.M. further acknowledges support from a Parkinson's UK Senior Fellowship (F1301). T.A.S. acknowledges UKRI Future Leaders Fellowship (MR/W004615/1). S.K.U. acknowledges funding from the Francis Crick Institute (grant FC001201), which receives its core funding from Cancer Research UK, the Medical Research Council and the Wellcome Trust. We acknowledge Prof. Jernej Ule and Dr. Llywelyn Griffith from the Francis Crick Institute for stimulating discussions and advice.

## Author contributions

G.M.H. designed the overall study. G.M.H. oversaw the ChIP-seq, RNA-seq, biochemical, molecular and cellular biology experiments while the TMT spectrometry and mitochondrial respiratory chain investigations were, respectively, supervised by S.K.U. and H.M. S.R.M. generated the stable inducible cell models. S.R.M., L.M.C., Y.H.L., A.G., N.S. and G.M.H. performed the biochemical, molecular and cellular biology assays. T.A.S. performed the PLA and RNA-FISH experiments, while L.M.C. and C.H. did the respiratory chain activity assays. S.R.M., L.M.C. and Y.H.L. prepared the DNA and RNA samples for next-generation sequencing with assistance from O.P. R.G. aligned RNA/DNA-seq reads and performed the differential expression analysis. S.R.M., H.R.F., A.P.S. and M.J.D. performed mass spectrometry experiments. S.R.M. performed the GO analysis. S.R.M. undertook his PhD in GMH's group and is currently a postdoctoral researcher in SKU's group. G.M.H. and O.B. supervised the PhD thesis of S.R.M. G.M.H. wrote the manuscript. S.R.M., L.M.C., Y.H.L., T.A.S., H.M. and S.K.U. edited the manuscript. All authors provided comments and approved the manuscript.

## Competing interests

The authors declare no competing interests.
