## [Peer Review File · Nature Communications]

The master energy homeostasis regulator PGC-1 α exhibits a novel mRNA nuclear export functionREVIEWER COMMENTS

Reviewer #1 (Remarks to the Author):

Mihaylov et al. show that PGC-1 α has two distinct roles in regulation of gene expression: i) transcriptional activation and ii) nuclear export of mRNAs. Interestingly, these roles can be uncoupled. Previous studies had shown that PGC-1 α can bind RNA molecules. This study utilizes the PGC-1 α protein that lacks RS domain for these studies. The authors can easily build on this solid foundation to make the findings more novel, rather than a confirmation of previous hypotheses. Here are specific comments:

* It is important to indicate the caveats of the RS mutant, and keep in mind that this domain also binds to the Mediator Complex and p300 that are required for transcription. This should be indicated in the manuscript and explain/interpret the results. In addition, this domain precludes localization to speckles, how the authors interpret their results with functional deposition at these domains? Do speckles localize with the RNA export machinery?

*Figure 1b-c: (GGGGCC)₅ experiments in 1b (right panel) and 1c seem to be redundant. While it is great to see reproducibility between experiments, 1b-right panel can be moved to the supplement.

*Fig 1e – The increase in polyA is hard to detect by eye, and some pictures look overexposed. It would be great to see a quantitation of this normalized to a staining control.

*Fig 2,3- Some of the figures are disrupted by the pdf processing. Some “ α ” letters can’t be displayed and some antibody labels for Western Blots have empty boxes.

*Fig 2d- All this data suggests is that NXF1 and PGC-1 α are both nuclear. Co-localization is too strong of a description for these images. Proximity Ligation Assay (PLA) or a similar technique needs to be used to say they interact. Or if there was a punctate staining pattern and the puncta co-localized, that would be co-localization. PGC-1 α seems to be absent from the nucleolus, but that isn’t obvious for NXF1. Maybe a better representative image can be chosen if this is the point that authors are trying to make.

*Fig 5a – It seems that PGC-1 α that lacks the RS domain can still interact with Cox5a, even if it’s significantly less compared to WT. How do the authors explain the difference of Cox5a from other transcripts (Nrf1, Cox10, Tfam) that go back down to Sham levels in the RS mutant?

*Fig 5d- It would be nice to show single molecule FISH for one transcript, accumulating in the nucleus. One can even gather spatial information about the mechanism of this accumulation, are these transcripts buried in the nucleolus or are they interacting with the Nuclear Pore Complexes? One can also combine smFISH with antibody staining to show colocalization with PGC-1 α in WT vs the RS mutant. It is quite interesting that intrinsically disordered proteins can form liquid droplets in the nucleus, and it would be nice to see if this is the case for PGC-1 α and its regulated transcripts. smFISH would reveal if there are clusters of RNA and PGC-1 α in the nucleus. The overexpressed PGC-1 α is a haze over the nucleus, and my guess is that if the authors can decrease the amount of overexpressed PGC-1 α or stain for endogenous PGC-1 α , they may be able to capture a more punctate staining pattern.

*Fig7 – Authors could make use of previously published PGC-1 α ChIP-Seq data to identify PGC-1 α peaks. How much overlap is there with the proteins whose levels are changing? Can the RS mutant still bind known PGC-1 α peaks to transcriptionally activate?

*Fig 8 - Are the novel genes that weren’t known to be regulated by PGC-1 α bound by it (ChIP-qPCR or ChIP-Seq data)?

Reviewer #2 (Remarks to the Author):

In the manuscript entitled “The master energy homeostasis regulator PGC-1 α couples transcriptional co-activation and mRNA nuclear export”, Simeon et al. attempt to disentangle the biological function of PGC-1 α as a mRNA binding protein. As the authors point out in the introduction, in 2020 a paper was published characterising the PGC-1 α RNA targets. While recognising the existence of this publication, surprisingly, the authors still consider the RNA binding of PGC-1 as not characterised. As expected, they confirm that the protein can bind RNA in vivo and in vitro, and further determine that the RNA binding of this protein is mediated through the D3 domain. Importantly, the authors show that RNA binding is mediated by a RS region in the protein, and not by the putative RNA recognition motif. They also find that the same RS sub-domain also interacts with nuclear export receptor NXF1. Finally, the authors interrogate the functional relevance of PGC-1 α as an RNA binding protein, show how knocking down PGC-1 has implications in the nuclear accumulation of specific transcripts, observe that this effect is specifically mediated by the D3 domain of the protein and analyse its implications in cell viability.

The most relevant and repeated interpretation of their findings is that PGC-1 α protein drives nuclear export of specific transcripts. However, the authors’ data show that PGC-1 α is a nuclear protein. It is unclear how a protein that is not shuttling between nucleus and cytosol could act as a nuclear exporter. Considering that the protein has been found to bind mainly intronic RNA close to splicing sites, an alternative explanation would be that the protein is affecting RNA splicing and that its dysregulation affects the splicing of specific transcripts, and therefore their nuclear retention/export. While finding that this protein could selectively affect transcript retention is interesting, its role as an RNA binder is already described, and the authors do not provide mechanistic insights about how this protein can act specifically on some transcripts (while its over-expression seems to generate broad effects, figure 1), or the specific role of the protein in RNA export/splicing/nuclear-retention (nuclear retention can be the result of many different processes not interrogated in this study). As the authors discuss: “More research is now required to investigate whether PGC-1 α plays a role in the splicing of cellular transcripts in order to dissect the molecular mechanisms which couple the transcriptional co-activation and mRNA nuclear export functions of PGC-1 α .” I agree, without it the novelty of the manuscript relies on the nuclear retention of some RNAs upon PGC-1 α knock down. A more mechanistic answer is needed before saying, so insistently, that the role of the protein is to mediate RNA export.

Again, considering the binding of PGC-1 α to introns, its nuclear localisation, and the nuclear accumulation of transcripts upon its knock down, before considering that the protein has a direct role in active export of these transcripts, the authors should discard potential effects on splicing. Do the nuclear-retained transcripts retain introns upon PGC-1 α knock down?

“Taken together, our data demonstrate that the RNA-binding domain of PGC-1 α exhibits a novel NXF1-dependent mRNA nuclear export function which is linked to the transcripts it co-activates.”, how do

they know that it is NXF1-dependent? They should KD NXF1 and check if the nuclear retention of transcripts upon PGC-1a KD can be recovered or not with a knock down resistant PGC-1a.

Along the manuscript, the authors insist on the fact that PGC-1 post-transcriptionally regulated transcripts are also transcriptionally co-activated while, upon PGC-1a KD, the expression of these transcripts remains unchanged. A proof showing that these transcripts are transcriptionally regulated by PGC-1a in this model is therefore required to claim this.

In figure 2c there are many bands of lower molecular weight apart from the tagged PGC1-a protein. What are they? Likewise, in the figure 3d, there is indication of two molecular weights, one referring to the complete protein, and the other one to the RS mutated one. Interestingly, while in the RS mutated one there is a single band, in the complete PGC-1a there are several lower molecular weight bands. Can the authors speculate on what the other bands are?

In the lines 224-219, the authors say “Interestingly, D3 interacts with NXF1 when synthesised in a mammalian transcription/translation-coupled system which allow for post-translational modifications (PTMs) (Fig. 3b) whereas it does not when expressed in bacteria which do not support mammalian PTMs (Fig. 3c), suggesting that PTMs play a role in the NXF1-dependent regulation binding of PGC-1 α . However, preliminary experiments using phosphatase or methyl transferase inhibitors did not allow elucidating this further.” While potentially interesting, this is not more than a speculation. The authors should analyse the proteins produced in the different models by mass spectrometry and check if the proteins have PTMs on them or remove these comments from the manuscript.

There is no indication of the “n” used in any experiment (eg figure 4 or 5). Please, indicate the number of experiments in all figure legends.

After characterising the role of PGC-1a as an RBP using UV-C (a specific protein-RNA crosslinking method) the authors switch to formaldehyde (FA), a reagent that can be used to crosslink nucleic acids to proteins, but also proteins to proteins (and DNA to proteins). Importantly, when trying to provide extra specificity to FA crosslinking, favouring RNA-protein interactions, 0.1% is normally used. The authors use 1%, that is a common dosage when fixing protein-protein interactions. Why do the authors change the method of crosslinking? This has important implications when interpreting the results. If the authors are stabilising full protein complexes to RNAs, the RNA pulled down with PGC-1a can be, or not, interacting directly with PGC-1a. This is even more relevant, considering that according to the 1st figure, the RS motif seems to have a dual function in RNA binding and protein-protein interaction, so any mutation in RS and FA crosslinking could be interpreted as not forming RNA-protein OR protein-protein interactions, making any biological interpretation difficult, if possible. Moreover, the change in the crosslinking method has another relevant implication: In a recent article where the transcripts bound by PGC-1a are characterised (Ref44), they use the RNA-protein-specific UV-C crosslinking method. Importantly, the authors of that publication say that the transcripts bound by PGC-1a are not the same ones the protein regulates transcriptionally. However, in this manuscript, the authors say that the PGC-1a binds, as an RBP, to the same transcripts that its transcriptionally regulates. The authors must show if

the transcripts that they select as novel PGC-1 α targets are found in the CLIP experiment in Ref44. Importantly, if not, the disagreement could arise from the unspecific FA crosslink. The authors should repeat the protein-RNA interaction experiments in figure 5, 6 and 8 using UV-C, or prove the specificity (only PGC-1 α -RNA crosslink and not full protein complexes-RNA) of their crosslink method.

Authors suggest on several occasions that “PGC-1 α binds the cellular mRNAs it transcriptionally co-activates”. Nevertheless, when knocking down the protein, the expression of the transcripts (e.g. TFAM, NRF1, COX10 and COX5A) that are supposed to be co-transcriptionally activated does not change. The authors should show any proof that these transcripts are co-transcriptionally co-activated in their system if these conclusions remain in the manuscript.

The authors should consider PGC-1 α CLIP experiments in ref 44 when interpreting the impact of their results. Are transcripts like GLS, XRCC5, XRCC6 and NHEJ1 already been found to be bound by PGC-1 α ?

Surprisingly, some of the headers in the results section in this manuscript have little in common with the results the section contains. An extreme example is the section “Investigating the physiological relevance of the potential mRNA nuclear export activity of PGC-1 α ”. A section where the authors show that they can knock down the protein, and generate knock down resistant constructs, without further biological interpretation or physiological role investigation. Or “The RS domain of PGC-1 α interacts with transcriptionally co-activated mRNAs.” Where the authors show that PGC-1 α interacts with RNA in vitro and in vivo through the RS domain, but there is no evidence of its relationship with any transcriptional activity or co-activation.

Figures are of very poor quality (eg Fig 6 h Y axis, or fig 9 “model” where many names are not complete, like DN (A?)).

Minor comments,

Figure 1e shows a single cell. Do the authors have more images?

Figure 2c, if everything is treated with RNase, is it necessary to indicate it in the panel?

There are numerous question marks in figures. While I assume the authors try to indicate that those are the hypotheses they are investigating, it makes the figure difficult to interpret.

Authors put too many methodological comments in the results section (e.g. The depletion of PGC-1 α was validated using 2 pairs of primers annealing in either the coding sequences (CDS) or the 3'-untranslated region (3'-UTR) of the endogenous PGC-1 α mRNA, or using 10plex Tandem Mass Tag (TMT) spectrometry, a methodology for multiplexed quantification of proteins in 10 sample) which make reading of this section difficult.

In line 279, the authors say that after knocking down PGC-1 α , a mutant protein without “RNA/NXF1-binding domain inhibits the nuclear export of PGC-1 α target”. ‘Is not able to rescue the export of the target transcripts’ would be more appropriate.

The mass spec experiment is very oddly designed. The authors divide and highlight in figure 7 cells treated in Glucose and Galactose media. Nevertheless, they pool both conditions to compare between WT and RS mutated cells. I assume that considering that in some of the conditions (galactose treated) the n = 2, it is unlikely that they had any significant hit. Despite not being fundamentally wrong, the experimental design is not ideal. Moreover, they analyse the proteome-wide affects of the RS mutation by GO and KEGG pathways analysis of the proteins they found dysregulated, but consider both up and down regulated proteins together. Despite not erroneous, this complicates the biological interpretation. They should consider up and down regulated proteins independently and repeat the analysis to clarify the interpretation of the results.

It is not accurate to interpret the results of the proteomic analysis as “this investigation also highlighted reduced expression levels of proteins involved in processes not previously known to be regulated by PGC-1 α ”. After the length of the treatment, the effects could easily be indirect or secondary. ‘Regulated’ needs to be toned down.

Reviewer #3 (Remarks to the Author):

The manuscript by Mihaylov et al. investigates the RNA-related functions of PGC-1alpha, a transcriptional coactivator involved in the expression of genes required for energy metabolism homeostasis in mammals. The authors convincingly demonstrate that PGC-1alpha interacts with RNAs through its RS domain, both in vitro and in vivo (Fig. 1), in line with a published study identifying PGC-1alpha-bound mRNA targets based on CLIP (authors’ reference #44). They further establish that this same domain interacts in vitro and in vivo with NXF1, the main mRNA export receptor (Fig. 2-3). By using human cell lines engineered to down-regulate endogenous PGC-1alpha levels and express in an inducible manner either the wt or a truncated version of PGC-1alpha lacking the RS sequence (Δ RS, Fig. 4), they demonstrate that this protein domain is required for the nuclear export of a handful of mRNAs encoded by PGC-1alpha target genes sharing mitochondrial functions (Fig. 5). Phenotypically, the nuclear retention of these transcripts is further correlated with defects in mitochondrial biogenesis/activity and cell proliferation in Δ RS cell lines (Fig. 6). Furthermore, whole proteome analysis reveals changes in the cellular levels of several proteins in the absence of the RS domain (Fig. 7), some of them involved in mitochondrial biology. Novel PGC-1alpha-dependent proteins identified in this screen are further validated, showing that the nuclear export of the corresponding transcripts also depends on the presence of PGC-1alpha RS domain (Fig. 8).

Overall, while this study clearly establishes that the RS domain of PGC-1alpha is required for RNA/NXF1 binding, mRNA nuclear export and proper expression of dozens of proteins, the underlying mechanism remains elusive, especially in light of PGC-1alpha previously reported functions. Additional experiments

are thereby required to substantiate the authors' findings, as described below.

1. As pointed in the introduction and discussed, PGC-1alpha reportedly contributes to transcriptional activation and splicing. Importantly, optimal mRNA synthesis from PGC-1alpha-target genes and splicing regulation by PGC-1alpha require its C-ter domain, which encompasses the RS sequence (authors' reference 17). In view of the critical importance of splicing for mRNA export, it is thereby possible that PGC-1alpha primarily controls mRNA processing, indirectly impacting on mRNA export.

- Since the role of PGC-1alpha in splicing was solely described using a reporter gene (reference #17), can the authors check whether splicing is globally affected in Δ RS cell lines? Is it the case for the transcripts for which nuclear export is compromised? RNA-seq-based comparisons of wt and Δ RS cell lines should address this important issue (see also below, 3.).

- Are there intronless genes among PGC-1alpha transcriptional targets? If this is the case, these genes should be analyzed for RNA localization in the fractionation assay (Fig. 5) to explore a possible splicing-independent function of PGC-1alpha in mRNA export.

- PGC-1alpha interaction with TREX (Fig. 2C) suggests that PGC-1alpha could enhance export of spliced transcripts by contributing to TREX recruitment. Could the authors perform TREX RIP on PGC-1alpha target transcripts in wt and Δ RS cell lines (see also below, 2.)?

2. The authors fail to provide a mechanism reconciling in vitro findings (PGC-1alpha interaction with RNAs and NXF1) with in vivo observations (the requirement of PGC-1alpha for mRNA nuclear export).

- Does PGC-1alpha function as a (novel) NXF1 adaptor? Does it impact NXF1 RNA binding activity as previously reported for TREX (authors' reference #41)? To address these questions, the authors could ask (i) whether PGC-1alpha can simultaneously interact with NXF1 and RNA (using in vitro binding assays as featured in Figs. 1-3), and (ii) whether PGC-1alpha is required for proper loading of NXF1 onto its target transcripts (by performing NXF1 RIP in control and PGC-1alpha-depleted cells).

- The authors claim that the PGC-1alpha-TREX interaction is stabilized in the Δ RS mutant (lines 200-203, related to Fig. 2C), possibly reflecting a stalling in mRNP remodeling when PGC-1alpha does not interact with RNA. However, this conclusion is not supported by the data: the Δ RS mutant protein seems much highly expressed in transient transfections (see the input panel in Fig. 2C), and likely more abundant in immunoprecipitates. The increased interaction with TREX must thereby be quantified by normalizing the amount of co-immunoprecipitated TREX proteins to the amounts of PGC-1alpha detected in the respective pull-downs.

- Since PGC-1alpha-dependent mRNA export occurs for PGC-1alpha transcriptional targets (Fig. 5), could the authors discuss how they envision the transfer of PGC-1alpha from DNA to RNA?

3. The proteome analysis of wt and Δ RS cell lines reveals that the cellular levels of several dozens of proteins change depending on PGC-1alpha activity (Fig. 7). While these results are interesting as a basis for understanding the relationships between PGC-1alpha and multiple cellular or pathology-related processes, they do not document as such the proteome-wide impact of PGC-1alpha mRNA nuclear export activity. Indeed, changes in transcriptional activation or splicing could also ultimately impact protein levels. Whole proteome analysis has thereby to be coupled with RNA-seq (in wt and Δ RS cell lines, see also above 1.), to identify the proteome changes that are primarily caused by alterations in

mRNA nuclear export.

Minor points.

1. A few sentences need to be rephrased: « It further interacts and co-activates them » (line 77); « PGC-1alpha is able to splice a reporter.... It may co-transcriptionally splice transcripts it coactivates. » (lines 94-95).
2. The authors should tone down novelty claims related to their data in the abstract (lines 39-40) or in the Introduction (lines 122-123).
3. Figure titles have to be corrected to reflect the featured data. For instance, Fig. 1 is entitled «The RS domain of PGC-1alpha interacts with transcriptionally co-activated mRNAs » while it does not demonstrate that this interaction is restricted to transcriptional targets. Fig. 7 indicates « Proteome-dependence of the mRNA nuclear export function of PGC-1alpha » while the authors have investigated the impact of the nuclear export function of PGC-1alpha on the proteome.
4. Fig. 1B: please provide lower exposures of the images to see the radiolabeled bands.
5. The colocalization data (Fig. 2D, S4) are not convincing. Both NXF1 and PGC-1alpha appear to be nucleoplasmic with nucleolar exclusion, which could happen in the absence of any interaction. I would recommend to remove these data.
6. Could the authors discuss the fact that they do not detect in their proteomic screen the targets of PGC-1alpha mRNA nuclear export activity as identified in Figs. 5-6 (i.e. TFAM, NRF1, COX10, COX5A, COX2)?

POINT-BY-POINT RESPONSE TO THE REVIEWERS' COMMENTS: MANUSCRIPT NCOMMS-22-04129

We acknowledge the Reviewers and the Editor for their time and for very positive and constructive comments which have clearly improved our manuscript. Below, we are providing a point-by-point response in which we feel we addressed all potential concerns. Our response is labelled in blue in line with the revised text in the main article file.

The manuscript has been substantially revised including 3 new Figures (Figs. 3, 7, 8), some revised panels (Figs 1, 2, 5, 9), a more detailed model (Fig. 10), 3 new Supplementary Figures (3, 8, 9) and 5 new Supplementary Tables. Briefly, new immunofluorescence microscopy and proximity ligation assays were performed to show that PGC-1 α interacts with NXF1 in the nucleus outside of splicing speckles and with NUP107 at the entrance of the channel of the nuclear pore complex. We repeated the RNA immunoprecipitation assays (RIPs) using UV-C instead of formaldehyde cross-linking to reassuringly obtain the same findings. We combined our proteome investigation with RNA-seq and Chromatin IP following sequencing (ChIP-seq) to gain mechanistic details on the contribution of the different functions of PGC-1 α at genome-wide levels, identifying PGC-1 α -dependent RNA expression, splicing and nuclear export targets as well as binding to promoters. This led to the discovery of predicted novel non-canonical promoter-independent gene targets in the nucleocytoplasmic transport pathway which were further validated to be genuine mRNA nuclear export targets of PGC-1 α at experimental level.

REVIEWER COMMENTS

Reviewer #1 (Remarks to the Author):

Mihaylov et al. show that PGC-1 α has two distinct roles in regulation of gene expression: i) transcriptional activation and ii) nuclear export of mRNAs. Interestingly, these roles can be uncoupled. Previous studies had shown that PGC-1 α can bind RNA molecules. This study utilizes the PGC-1 α protein that lacks RS domain for these studies. The authors can easily build on this solid foundation to make the findings more novel, rather than a confirmation of previous hypotheses. Here are specific comments:

* It is important to indicate the caveats of the RS mutant, and keep in mind that this domain also binds to the Mediator Complex and p300 that are required for transcription. This should be indicated in the manuscript and explain/interpret the results. In addition, this domain precludes localization to speckles, how the authors interpret their results with functional deposition at these domains? Do speckles localize with the RNA export machinery?

We fully agree that we could have made more references to the interactions between the RS domain of PGC-1 α and the MED1 subunit of the Mediator/p300 complex. We have now detailed this further in the introduction and added a special point in the discussion. Moreover, we have modified the text throughout the manuscript by replacing alterations of the RNA/NXF1-binding functions in the PGC-1 α Δ RS-res cell line with loss-of-function of the RS domain.

It is noteworthy that the interactions of PGC-1 α with MED1 may not be required for co-activation at all promoters since PGC-1 α co-activates transcription with a multitude of other transcriptional co-activators depending on the cell type and metabolic/ diseased conditions. The new transcriptome studies we generated during revision of the manuscript together with qRT-PCR analysis also revealed that the steady state levels of most transcripts known to be transcriptionally co-activated by PGC-1 α (*NRF1*, *TFAM*, *COX10*, *COX5A* or *CoA6*) are not affected upon expression of the Δ RS mutant and/or depletion of endogenous PGC-1 α (new **Supplementary Table 4, Fig. 7**) in light with the fact that PGC-1 α is a transcriptional co-activator rather than a positive transcription factor on its own.

On the other hand, we did not expect the interaction of PGC-1 α to the competent nuclear export machinery to be occurring within nuclear speckles but rather to co-localise with the nucleopore complex. We have performed new experiments to specifically address this point. As shown by immunofluorescence microscopy in new **Fig. 3a**, PGC-1 α exhibits a nuclear speckled pattern however, we did not find that PGC-1 α localises to speckles or paraspeckles in agreement with a recent study highlighting that a large proportion of PGC-1 α foci corresponds to chromatin condensates rather than nuclear speckles (Perez-Schindler J et al. PNAS 2021; 118(36):e2105951118). This is in contrast to an older report indicating that overexpressed PGC-1 α -GFP co-localises with U1-snRNP70K/snRNP70 (Monsalve M et al. Mol. Cell 2000; 6:307-16) however, snRNP70 was later shown not to be a strict nuclear speckles marker as it rapidly exchanged between the nucleoplasm and nuclear speckles (Ali GS et al. PLoS One 2008; 3(4):e1953).

Moreover, proximity ligation assays (PLA), suggested below by the reviewer, showed that PGC-1 α and NXF1 interacts in the nucleus outside of the nuclear splicing speckles (new **Fig. 3b**) consistently with previous observations in the literature and Fig. 3a. In addition, we also show by PLA that PGC-1 α co-localizes with nucleoporin NUP107 in the nuclear outer ring at the channel entrance of the nuclear pore complex (new **Fig. 3c**) in full agreement with its role as a nuclear mRNA export adaptor licensing the nuclear export of mRNA.

*Figure 1b-c: (GGGGCC)₅ experiments in 1b (right panel) and 1c seem to be redundant. While it is great to see reproducibility between experiments, 1b-right panel can be moved to the supplement.

We acknowledge this suggestion however, we would prefer to show non-sequence specific binding to both AU and GC-rich RNAs in the same panel of a figure as these gels were exposed for autoradiography at the same time. We have therefore kept the right panel of figure 1b in the main figure and, as suggested by Reviewer 3, provided lower exposures for the phosphoimages.

*Fig 1e – The increase in polyA is hard to detect by eye, and some pictures look overexposed. It would be great to see a quantitation of this normalized to a staining control.

We acknowledge the Reviewer for this comment which helped clarifying the text. This is a qualitative assay and the Poly-A+ RNA signal is indeed saturated in the nucleus of the cells overexpressing Aly/REF and PGC-1 α due to the disruption of the bulk mRNA nuclear export. On the other hand, cells with physiological level of mRNA nuclear export in untransfected or the e1BAP5-control condition show both nuclear and cytoplasmic signal. To guide the reader, we added some lines to delineate the nucleus and cytoplasm. We also included an extra

sentence in the figure legend: “The solid white line delineates the cytoplasm while the dotted lines label nuclei. Cells with overexpression of FLAG-tagged Aly/REF and PGC-1 α show a block in the bulk nuclear export of mRNAs with saturated accumulation of Poly-A⁺ RNA nuclear staining and absence of detectable cytoplasmic signal”.

*Fig 2,3- Some of the figures are disrupted by the pdf processing. Some “ α ” letters can’t be displayed and some antibody labels for Western Blots have empty boxes.

Apologies for these formatting issues appearing during the submission process in the merged pdf. These have now been rectified as checked pdf files were up-loaded.

*Fig 2d- All this data suggests is that NXF1 and PGC-1 α are both nuclear. Co-localization is too strong of a description for these images. Proximity Ligation Assay (PLA) or a similar technique needs to be used to say they interact. Or if there was a punctate staining pattern and the puncta co-localized, that would be co-localization. PGC-1 α seems to be absent from the nucleolus, but that isn’t obvious for NXF1. Maybe a better representative image can be chosen if this is the point that authors are trying to make.

We acknowledge the Reviewer for this constructive comment and an excellent suggestion. As aforementioned, we have removed Fig. 2d/ Supplementary Fig. 4 and replaced them with various proximity ligation assays showing that the interaction of PGC-1 α with NXF1 occur in the nucleus outside of the nuclear splicing speckles (new **Fig. 3b**). Moreover, we show that PGC-1 α co-localises at an expected site of interaction with nucleoporin NUP107, a component of the nuclear outer ring of the nuclear pore complex (new **Fig. 3c**).

*Fig 5a – It seems that PGC-1 α that lacks the RS domain can still interact with Cox5a, even if it’s significantly less compared to WT. How do the authors explain the difference of Cox5a from other transcripts (Nrf1, Cox10, Tfam) that go back down to Sham levels in the RS mutant?

The formaldehyde RIP have been repeated using UV-C for crosslink as suggested below and Fig. 5a has now been moved to Fig. Supplementary 6a.

The conclusion which can be drawn is that PGC-1 α lacking the RS domain binds significantly less *COX5A* mRNA compared to wild-type PGC-1 α ($p < 0.0001$), however it is challenging to tell whether the residual binding of *COX5A* mRNA to PGC-1 α Δ RS is biologically significant or not. The key point is that the reduction in the binding of *COX5A* mRNA to PGC-1 α Δ RS is sufficient to inhibit the nuclear export of *COX5A* mRNA as shown in Fig. 5e, likely by inhibiting recruitment to the processing transcript. In addition, the potential remaining fraction of PGC-1 α Δ RS-bound *COX5A* transcripts would not interact with NXF1, due to the lack of the RS domain which is required for interactions with NXF1, thereby explaining further the nuclear export inhibition of *COX5A* mRNA. We have updated the text as: “transcripts of the known above co-activated genes specifically interact with FLAG-tagged PGC-1 α while the Δ RS mutant exhibits significant less binding to RNA”.

*Fig 5d- It would be nice to show single molecule FISH for one transcript, accumulating in the nucleus. One can even gather spatial information about the mechanism of this accumulation, are these transcripts buried in the nucleolus or are they interacting with the

Nuclear Pore Complexes? One can also combine smFISH with antibody staining to show colocalization with PGC-1 α in WT vs the RS mutant. It is quite interesting that intrinsically disordered proteins can form liquid droplets in the nucleus, and it would be nice to see if this is the case for PGC-1 α and its regulated transcripts. smFISH would reveal if there are clusters of RNA and PGC-1 α in the nucleus. The overexpressed PGC-1 α is a haze over the nucleus, and my guess is that if the authors can decrease the amount of overexpressed PGC-1 α or stain for endogenous PGC-1 α , they may be able to capture a more punctate staining pattern.

This is another very nice suggestion. We attempted to perform smFISH using *Stellaris* custom probes against *TFAM* and *NRF1* transcripts however we did not detect any specific localised signal. On the other hand, the new proximity ligation assays provided during the revision shall address this comment by showing specific puncta staining and co-localisation with both NXF1 and NUP107, a component of the nuclear pore complex, rather than the haze of overexpressed PGC-1 α over the nucleus.

*Fig7 – Authors could make use of previously published PGC-1 α ChIP-Seq data to identify PGC-1 α peaks. How much overlap is there with the proteins whose levels are changing? Can the RS mutant still bind known PGC-1 α peaks to transcriptionally activate?

During the revision of this manuscript, we have performed ChIP-seq and whole-cell/cytoplasmic RNA-seq on the PGC-1 α WT-res and Δ RS-res cell lines to investigate promoter-binding and RNA expression, splicing and nuclear export at genome-wide levels. Briefly, we identified binding of PGC-1 α to 2,241 promoters (new **Fig. 7a**), including 2,232 protein-coding promoters, with poor overlap with other published Chip-seq data in mouse skeletal muscle or hepatic cells (new **Supplementary Fig. 9a-c, new Supplementary Table 1**). We further determined that PGC-1 α binds 97% of its target promoters independently of the RS domain (new **Fig. 7b**). This is consistent with our findings that total steady state levels of most known PGC-1 α co-activated transcripts are not affected in our transcriptomics and qRT-PCR investigations. As suggested in the discussion, this might reflect the fact that the depletion of endogenous PGC-1 α is partial (approximately 70% at mRNA and protein levels, Fig. 5e, h) and that HEK293T cells do not necessarily rely on mitochondrial respiration for growth in contrast to neuronal or hepatic cells for examples. They provide however an ideal model here to identify mRNA nuclear export targets, which are not differentially expressed at steady-state levels.

465 mRNAs and 128 proteins are down-regulated in the Δ RS-res cell line with a 15% overlap only, in line with genome-wide studies indicating that the abundance of mammalian proteins is mostly not attributed to total mRNA concentrations (Vogel C et al. *Molecular Systems Biology* 2010; 6:1-9; Schwanhäusser B et al. *Nature* 2011; 473:337). On the other hand, 56% of the downregulated proteome changes depend on reduced cytoplasmic mRNA levels (new **Fig. 7j**). Importantly, 52% of the down-regulated proteome changes are linked to the mRNA nuclear export function of PGC-1 α while 18% correlate with PGC-1 α -dependent binding to promoters indicating that both canonical and non-canonical transcripts are regulated by the nuclear export function of PGC-1 α (new **Figs. 8f-g**), in agreement with its recently identified RNA-binding targets (Tavares CDJ et al. *PNAS* 2020; 117:2220; Perez-Schindler J et al. *PNAS* 2021; 118:e2105951118).

*Fig 8 - Are the novel genes that weren't known to be regulated by PGC-1 α bound by it (ChIP-qPCR or ChIP-Seq data)?

This is indeed a very interesting point which has now been answered with the genome-wide investigation which identified mRNAs nuclear export targets which do and do not involve PGC-1 α binding to promoters. The novel genes not previously known to be regulated by PGC-1 α therefore fall into these two categories: PGC-1 α is involved in binding the promoters of *XRCC5* and *XPO1* but not of *GLS*, *NHEJ1*, *XRCC6*, *KPNA2*, *DDX19B*, *NUP58*, *62*, *98*. Interestingly, gene ontology investigation of PGC-1 α bound promoters and nuclear export targets allowed defining pathways involving either both or one of the functions of PGC-1 α . Thermogenesis and oxidative respiration appear to be specific of the promoter-binding function of PGC-1 α while other canonical processes such as mitochondrial/ metabolic pathways, AMPK/ insulin signalling and regulation of RNA polymerase II transcription and longevity involves both the binding to promoters and mRNA nuclear export functions of PGC-1 α . On the other hand, non-canonical pathways including glucagon signalling (also identified in the eCLIP study by Tavares CDJ *et al. PNAS* 2020; 117:2220) and newly identified processes such as translation initiation and age-related maintenance of telomere and nucleocytoplasmic transport are specifically regulated by the nuclear export function of PGC-1 α . A revised model is presented in **Fig. 10**.

Reviewer #2 (Remarks to the Author):

In the manuscript entitled “The master energy homeostasis regulator PGC-1 α couples transcriptional co-activation and mRNA nuclear export”, Simeon *et al.* attempt to disentangle the biological function of PGC-1 α as a mRNA binding protein. As the authors point out in the introduction, in 2020 a paper was published characterising the PGC-1 α RNA targets. While recognising the existence of this publication, surprisingly, the authors still consider the RNA binding of PGC-1 as not characterised. As expected, they confirm that the protein can bind RNA *in vivo* and *in vitro*, and further determine that the RNA binding of this protein is mediated through the D3 domain. Importantly, the authors show that RNA binding is mediated by a RS region in the protein, and not by the putative RNA recognition motif. They also find that the same RS sub-domain also interacts with nuclear export receptor NXF1. Finally, the authors interrogate the functional relevance of PGC-1 α as an RNA binding protein, show how knocking down PGC-1 has implications in the nuclear accumulation of specific transcripts, observe that this effect is specifically mediated by the D3 domain of the protein and analyse its implications in cell viability.

The most relevant and repeated interpretation of their findings is that PGC-1 α protein drives nuclear export of specific transcripts. However, the authors’ data show that PGC-1 α is a nuclear protein. It is unclear how a protein that is not shuttling between nucleus and cytosol could act as a nuclear exporter.

We did not imply that PGC-1 α is exclusively localised to the nucleus or that it does not shuttle and apologise for any potential confusion. Canonical mRNA nuclear export adaptors such as Aly/REF or SRSF1,3,7 show also a nuclear pattern of expression while they are known shuttling proteins, the shuttling cytoplasmic fraction being of too low abundance to be detected in microscopy experiments. Here, our data also show that PGC-1 α is a predominantly nuclear protein as reported by other groups in the field. We revised the manuscript by adding “predominantly nuclear” in the text relating to **new Fig. 3a-b**, which report that PGC-1 α interacts with NXF1 in the nucleus outside splicing speckles and paraspeckles. Moreover, proximity ligation assays showed co-localisation of PGC-1 α with

NUP107 in the nuclear outer ring at the channel entrance of the nuclear pore complex (**new Fig. 3c**) in full agreement with a role as a nuclear mRNA export adaptor licensing the nuclear export of mRNA.

Considering that the protein has been found to bind mainly intronic RNA close to splicing sites, an alternative explanation would be that the protein is affecting RNA splicing and that its dysregulation affects the splicing of specific transcripts, and therefore their nuclear retention/export. While finding that this protein could selectively affect transcript retention is interesting, its role as an RNA binder is already described, and the authors do not provide mechanistic insights about how this protein can act specifically on some transcripts (while its over-expression seems to generate broad effects, figure 1), or the specific role of the protein in RNA export/splicing/nuclear-retention (nuclear retention can be the result of many different processes not interrogated in this study). As the authors discuss: “More research is now required to investigate whether PGC-1 α plays a role in the splicing of cellular transcripts in order to dissect the molecular mechanisms which couple the transcriptional co-activation and mRNA nuclear export functions of PGC-1 α .” I agree, without the novelty of the manuscript relies on the nuclear retention of some RNAs upon PGC-1 α knock down. A more mechanistic answer is needed before saying, so insistently, that the role of the protein is to mediate RNA export. Again, considering the binding of PGC-1 α to introns, its nuclear localisation, and the nuclear accumulation of transcripts upon its knock down, before considering that the protein has a direct role in active export of these transcripts, the authors should discard potential effects on splicing. Do the nuclear-retained transcripts retain introns upon PGC-1 α knock down? “Taken together, our data demonstrate that the RNA-binding domain of PGC-1 α exhibits a novel NXF1-dependent mRNA nuclear export function which is linked to the transcripts it co-activates.”, how do they know that it is NXF1-dependent? They should KD NXF1 and check if the nuclear retention of transcripts upon PGC-1 α KD can be recovered or not with a knock down resistant PGC-1 α .

PGC-1 α was indeed reported to bind intronic RNA sequences (Tavares CDJ *et al. PNAS* 2020; 117:2220) but also to exonic sequences and UTRs with binding to exons as one of the most enriched category (Perez-Schindler J *et al. PNAS* 2021; 118:e2105951118), placing PGC-1 α in an ideal environment for recruitment to the nascent transcripts during pre-mRNA processing/splicing and/or co-transcriptional co-activation. On the other hand, we did not dismiss a role of PGC-1 α in splicing and cited a study suggesting its role as a splicing factor in a reporter system regulated by a PGC-1 α co-activated promoter (Monsalve M *et al. Mol. Cell* 2000; 6:307-1). During the revision, we performed ChIP-seq and RNA-seq experiments in our cell models to provide additional mechanistic insights. We identified 897 alternative splicing changes in the Δ RS-res cell line which lead to expression of PGC-1 α lacking the RS domain and depletion of endogenous PGC-1 α (**new Fig. 8d**), however the total expression levels of mRNAs encoding 30 splicing factors are also down-regulated in the Δ RS-res cells (**new Supplementary Fig. 9e**). It remains thus unclear whether the RNA-binding function of PGC-1 α plays a direct role in pre-mRNA splicing.

On the other hand, we found that 56% of downregulated proteome changes depend on reduced cytoplasmic mRNA levels (**new Fig. 7j**) while 52% of the down-regulated proteome changes are linked to the mRNA nuclear export function of PGC-1 α (**new Fig. 8g**) and 1,274 mRNAs are regulated by the nuclear export function of PGC-1 α (**new Supplementary Table 4 tab 4**). On the other hand, 18% of mRNA nuclear export targets also involve PGC-1 α -dependent binding to promoters indicating that both canonical and non-canonical transcripts

are regulated by the nuclear export function of PGC-1 α (**new Fig. 8f**), in agreement with its recently identified RNA-binding targets (Tavares CDJ *et al. PNAS* 2020; 117:2220; Perez-Schindler J *et al. PNAS* 2021; 118:e2105951118).

Regarding a justification for the involvement of NXF1 in the PGC-1 α -dependent nuclear export of RNA, we showed in this study that the RS domain interacts directly with both RNA and NXF1 in *in vitro* pulldown assays using purified recombinant proteins (**Fig. 1 and 2**). Moreover we show that PGC-1 α interacts with NXF1 in an RS domain dependent manner in co-immunoprecipitation assays (**Fig. 2d**). Knocking down NXF1 might appear as an interesting experiment however this is not feasible as depletion of the essential NXF1 protein would rapidly inhibit the bulk of the mRNA nuclear export and kill cells. Moreover, knocking down endogenous PGC-1 α while expressing a resistant form of PGC-1 α upon loss-of-function of NXF1 would not rescue the nuclear retention/ altered nuclear export of transcripts since the knock down resistant PGC-1 α protein would require interaction with NXF1 to license the nuclear export of mRNAs.

In addition, the manuscript does not “only relies on the nuclear retention of some RNAs upon PGC-1 α knock down”. Multiple biochemical and cellular assays are used throughout, including new proteomics and new Chip-seq/RNA-seq data sets, in an isogenic inducible complementation system allowing simultaneous depletion of endogenous PGC-1 α and expression of the Δ RS mutant lacking the RNA/NXF1 binding activity.

Along the manuscript, the authors insist on the fact that PGC-1 post-transcriptionally regulated transcripts are also transcriptionally co-activated while, upon PGC-1 α KD, the expression of these transcripts remains unchanged. A proof showing that these transcripts are transcriptionally regulated by PGC-1 α in this model is therefore required to claim this.

We made this statement because the 5 transcripts known to be co-activated by PGC-1 α that were tested in the study were also nuclear export targets of PGC-1 α , however we completely agree with the Reviewer that this cannot be extended at genome-wide level. Using ChIP-seq in our cell model, we identified binding of PGC-1 α to 2,241 promoters (**new Fig. 7a**), including 2,232 protein-coding promoters, with poor overlap with other published ChIP-seq data in mouse skeletal muscle or hepatic cells (**new Supplementary Fig. 6a-b, new Supplementary Table 1**). We further determined that PGC-1 α binds 97% of its target promoters independently of the RS domain (**new Fig. 7b**). Moreover this investigation allowed us to identify that 18% of mRNA nuclear export targets involves PGC-1 α -dependent binding to promoters indicating that both canonical and non-canonical transcripts are regulated by the nuclear export function of PGC-1 α (**new Fig. 8f**), in agreement with its recently identified RNA-binding targets (Tavares CDJ *et al. PNAS* 2020; 117:2220; Perez-Schindler J *et al. PNAS* 2021; 118:e2105951118). We have now revised the text throughout the manuscript highlighting “promoter-binding” rather than “transcriptional co-activation” and by replacing “The RS domain of PGC-1 α drives the nuclear export of transcriptionally co-activated mRNAs” by “The RS domain of PGC-1 α drives the nuclear export of some transcriptionally co-activated mRNAs” or “The RS domain of PGC-1 α plays a role in the mRNA nuclear export pathway”. To take into account that transcriptional co-activation was not directly investigated in the manuscript, we removed it from the title focussing on the findings in the manuscript i.e. “The master energy homeostasis regulator PGC-1 α exhibits a novel mRNA nuclear export function”.

A revised model is presented in **Fig. 10** to account for the genome-wide contribution of the PGC-1 α functions, with mRNA nuclear export targets which do and do not involve PGC-1 α binding to promoters. Thermogenesis and oxidative respiration are specific of the promoter-binding function of PGC-1 α while other canonical processes such as mitochondrial/metabolic pathways, AMPK/ insulin signalling and regulation of RNA polymerase II transcription and longevity involves both the binding to promoters and mRNA nuclear export functions of PGC-1 α . On the other hand, non-canonical pathways including glucagon signalling (also identified in the eCLIP study by Tavares CDJ *et al. PNAS* 2020; 117:2220) and newly identified processes such as translation initiation and age-related maintenance of telomere and nucleocytoplasmic transport are specifically regulated by the nuclear export function of PGC-1 α .

In figure 2c there are many bands of lower molecular weight apart from the tagged PGC1-a protein. What are they? Likewise, in the figure 3d, there is indication of two molecular weights, one referring to the complete protein, and the other one to the RS mutated one. Interestingly, while in the RS mutated one there is a single band, in the complete PGC-1a there are several lower molecular weight bands. Can the authors speculate on what the other bands are?

PGC-1 α was shown to be targeted quickly by the proteasome pathway including FLAG-tagged PGC-1 α (Sano M *et al. JBC* 2007; 282:25970-80). The lower molecular weight bands are thus likely to be degradation products. We have added a comment in the introduction to indicate that PGC-1 α is substrate of the ubiquitin/proteasome pathway.

In the lines 224-219, the authors say “Interestingly, D3 interacts with NXF1 when synthesised in a mammalian transcription/translation-coupled system which allow for post-translational modifications (PTMs) (Fig. 3b) whereas it does not when expressed in bacteria which do not support mammalian PTMs (Fig. 3c), suggesting that PTMs play a role in the NXF1-dependent regulation binding of PGC-1 α . However, preliminary experiments using phosphatase or methyl transferase inhibitors did not allow elucidating this further.” While potentially interesting, this is not more than a speculation. The authors should analyse the proteins produced in the different models by mass spectrometry and check if the proteins have PTMs on them or remove these comments from the manuscript.

We agree that the PTM comments do not add to the story as these were indeed speculative. They were thus removed from the manuscript.

There is no indication of the “n” used in any experiment (eg figure 4 or 5). Please, indicate the number of experiments in all figure legends.

This is a surprising comment. It was indicated in these figure legends that biological triplicate experiments were used. To emphasize this, “N=3” has also been added in the legends.

After characterising the role of PGC-1a as an RBP using UV-C (a specific protein-RNA crosslinking method) the authors switch to formaldehyde (FA), a reagent that can be used to crosslink nucleic acids to proteins, but also proteins to proteins (and DNA to proteins). Importantly, when trying to provide extra specificity to FA crosslinking, favouring RNA-protein interactions, 0.1% is normally used. The authors use 1%, that is a common dosage

when fixing protein-protein interactions. Why do the authors change the method of crosslinking? This has important implications when interpreting the results. If the authors are stabilising full protein complexes to RNAs, the RNA pulled down with PGC-1 α can be, or not, interacting directly with PGC-1 α . This is even more relevant, considering that according to the 1st figure, the RS motif seems to have a dual function in RNA binding and protein-protein interaction, so any mutation in RS and FA crosslinking could be interpreted as not forming

RNA-protein OR protein-protein interactions, making any biological interpretation difficult, if possible. Moreover, the change in the crosslinking method has another relevant implication: In a recent article where the transcripts bound by PGC-1 α are characterised (Ref44), they use the RNA-protein-specific UV-C crosslinking method. Importantly, the authors of that publication say that the transcripts bound by PGC-1 α are not the same ones the protein regulates transcriptionally. However, in this manuscript, the authors say that the PGC-1 α binds, as an RBP, to the same transcripts that its transcriptionally regulates. The authors must show if the transcripts that they select as novel PGC-1 α targets are found in the CLIP experiment in Ref44. Importantly, if not, the disagreement could arise from the unspecific FA crosslink. The authors should repeat the protein-RNA interaction experiments in figure 5, 6 and 8 using UV-C, or prove the specificity (only PGC-1 α -RNA crosslink and not full protein complexes-RNA) of their crosslink method.

We have repeated the formaldehyde-RIP using UV-C-RIP and obtained the same results. The UV-RIP have been inserted in the main figures (**new Fig. 5a, 9b, 9c**) while the formaldehyde-RIP was moved into **Supplementary Fig. 6a**. Of note, the transcript *COX5A* could not be quantified in both the total input and RIP conditions in the UV-RIP. Potential UV-induced dimers of thymidine or remaining covalent peptide:RNA bounds, which cannot be reversed unlike upon formaldehyde treatment, were likely to affect the qPCR amplification.

On the other hand, to answer “The authors must show if the transcripts that they select as novel PGC-1 α targets are found in the CLIP experiment in Ref44”, Ref 44 does not provide any lists of transcripts bound by PGC-1 α and the GEO submission did not provide these lists or simple mean to re-make them either. However, the glucagon-signaling pathway, identified in reference 44 (Tavares CDJ et al. *PNAS* 2020; 117:22204) as non-canonical RNA-binding targets of PGC-1 α , was also enriched in our list of PGC-1 α mRNA nuclear export targets and not found in the promoter-binding list. On the other hand, the novel mRNA nuclear export targets that we identified in our study were not reported in the Tavares et al manuscript.

Authors suggest on several occasions that “PGC-1 α binds the cellular mRNAs it transcriptionally co-activates”. Nevertheless, when knocking down the protein, the expression of the transcripts (e.g. TFAM, NRF1, COX10 and COX5A) that are supposed to be co-transcriptionally activated does not change. The authors should show any proof that these transcripts are co-transcriptionally co-activated in their system if these conclusions remain in the manuscript.

The results obtained in our initial study for 5 known promoters transcriptionally co-activated by PGC-1 α indicated that the transcripts produced from these promoters were also nuclear export targets of PGC-1 α . However, we completely agree that this cannot be generalised to genome-wide level. As aforementioned, we have now produced ChIP-seq and RNA-seq datasets to gain mechanistic insights and investigate the genome-wide contribution of PGC-

1 α functions including promoter binding, RNA expression levels, alternative splicing and nuclear RNA export targets.

Based on the new genome-wide investigation which highlighted that the mRNA nuclear export function of PGC-1 α regulates the cytoplasmic levels of canonical transcripts produced from promoters bound by PGC-1 α or independently of promoters bound by PGC-1 α , we have therefore revised the manuscript by removing that “PGC-1 α binds the cellular mRNAs it transcriptionally co-activates” and replacing this statement with “PGC-1 α drives the mRNA nuclear export of some canonical gene targets” or by linking the nuclear export function of PGC-1 α is to its binding to promoters rather than transcriptional co-activation.

The authors should consider PGC-1a CLIP experiments in ref 44 when interpreting the impact of their results. Are transcripts like GLS, XRCC5, XRCC6 and NHEJ1 already been found to be bound by PGC-1a?

Ref 44 does not provide any lists of transcripts bound by PGC-1 α and the GEO submission did not provide these lists or simple mean to re-make them either. However, the glucagon-signaling pathway, identified in reference 44 (Tavares CDJ et al. *PNAS* 2020; 117:22204) as non-canonical RNA-binding target of PGC-1 α , was also enriched in our list of PGC-1 α mRNA nuclear export targets and not found in the promoter-binding list. On the other hand, the GLS, XRCC5, XRCC6 and NHEJ1 transcripts were not reported to be bound by PGC-1 α in the Tavares et al manuscript.

Surprisingly, some of the headers in the results section in this manuscript have little in common with the results the section contains. An extreme example is the section “Investigating the physiological relevance of the potential mRNA nuclear export activity of PGC-1 α ”. A section where the authors show that they can knock down the protein, and generate knock down resistant constructs, without further biological interpretation or physiological role investigation. Or “The RS domain of PGC-a interacts with transcriptionally co-activated mRNAs.” Where the authors show that PGC-a interacts with RNA in vitro and in vivo through the RS domain, but there is no evidence of its relationship with any transcriptional activity or co-activation.

In the main text, the header “Investigating the physiological relevance of the potential mRNA nuclear export activity of PGC-1 α ” was replaced by “Engineering of isogenic inducible cell lines to investigate the potential mRNA nuclear export activity of PGC-1 α ”

In Fig. 1 legend, the header “The RS domain of PGC-1 α interacts with transcriptionally co-activated mRNAs” was replaced by “The RS domain of PGC-1 α interacts with RNA”.

Figures are of very poor quality (eg Fig 6 h Y axis, or fig 9 “model” where many names are not complete, like DN (A?)).

We apologise for several formatting issues which appeared during submission in the merged pdf. These have been checked and pdf copies were uploaded onto the web portal.

Minor comments,

Figure 1e shows a single cell. Do the authors have more images?

Additional images are provided in new **Supplementary Fig. 4**.

Figure 2c, if everything is treated with RNase, is it necessary to indicate it in the panel?

RNase was removed from Fig. 2c (revised Fig. 2d) and some text added in the legend.

There are numerous question marks in figures. While I assume the authors try to indicate that those are the hypotheses they are investigating, it makes the figure difficult to interpret.

This was indeed the intention, however we are happy to follow the Reviewer's guidance and have removed the question marks.

Authors put too many methodological comments in the results section (e.g. The depletion of PGC-1 α was validated using 2 pairs of primers annealing in either the coding sequences (CDS) or the 3'-untranslated region (3'-UTR) of the endogenous PGC-1 α mRNA, or using 10plex Tandem Mass Tag (TMT) spectrometry, a methodology for multiplexed quantification of proteins in 10 sample) which make reading of this section difficult.

We apologise for making these sections difficult to read. To try avoid making the first section about the primer pairs to read as materials and methods, the text was modified to: "The depletion of PGC-1 α was quantified using primers annealing in the coding sequence (CDS) or in the 3'-untranslated region (3'-UTR)". The justification for using the 2 primer pairs was explained in the following experiment with the Δ RS mutant Fig. 5e as follow: "Moreover, doxycycline-induced depletion of endogenous PGC-1 α quantified with the 3'UTR primers and expression of RNAi-resistant FLAG-tagged PGC-1 α WT/ Δ RS measured by the CDS primers led to...".

"10plex Tandem Mass Tag (TMT) spectrometry, a methodology for multiplexed quantification of proteins in 10 samples" was written to justify that we could only multiplex at the same time n=3 (glucose) and n=2 (galactose) for WT and DeltaRS strain i.e. 10 conditions. However, in light of the comments below, we have decided to completely removed the galactose dataset which did not provide much more additional data.

In line 279, the authors say that after knocking down PGC-1 α , a mutant protein without "RNA/NXF1-binding domain inhibits the nuclear export of PGC-1 α target". 'Is not able to rescue the export of the target transcripts' would be more appropriate.

The text has now been revised accordingly.

The mass spec experiment is very oddly designed. The authors divide and highlight in figure 7 cells treated in Glucose and Galactose media. Nevertheless, they pool both conditions to compare between WT and RS mutated cells. I assume that considering that in some of the

conditions (galactose treated) the $n = 2$, it is unlikely that they had any significant hit. Despite not being fundamentally wrong, the experimental design is not ideal. Moreover, they analyse the proteome-wide affects of the RS mutation by GO and KEGG pathways analysis of the proteins they found dysregulated, but consider both up and down regulated proteins together. Despite not erroneous, this complicates the biological interpretation. They should consider up and down regulated proteins independently and repeat the analysis to clarify the interpretation of the results.

We used this design as we could only multiplex 10 samples at the same time: $n=3$ (glucose) and $n=2$ (galactose) for WT and DeltaRS strain i.e. 10 conditions. However, we have now completely removed the galactose dataset which did not provide much more additional data. On the other hand, all of the gene ontology and KEGG pathways analysis has now been performed on the lists of down-regulated proteins, mRNAs or cytoplasmic mRNA in order to align the interpretation of data to the loss-of-functions of the RS domain of PGC-1 α .

It is not accurate to interpret the results of the proteomic analysis as “this investigation also highlighted reduced expression levels of proteins involved in processes not previously known to be regulated by PGC-1 α ”. After the length of the treatment, the effects could easily be indirect or secondary. ‘Regulated’ needs to be toned down.

The nuclear export of the transcripts encoding these proteins which are involved in new biological processes is specifically affected in the cell lines with expression of PGC-1 α lacking the RS domain and/or depletion. The reduction in the corresponding protein levels is thus functionally correlated to the alteration of the mRNA nuclear export function. “potentially” was added to tone down the sentence: “involved in processes not previously known to be potentially regulated by PGC-1 α ”.

Reviewer #3 (Remarks to the Author):

The manuscript by Mihaylov et al. investigates the RNA-related functions of PGC-1 α , a transcriptional coactivator involved in the expression of genes required for energy metabolism homeostasis in mammals. The authors convincingly demonstrate that PGC-1 α interacts with RNAs through its RS domain, both in vitro and in vivo (Fig. 1), in line with a published study identifying PGC-1 α -bound mRNA targets based on CLIP (authors' reference #44). They further establish that this same domain interacts in vitro and in vivo with NXF1, the main mRNA export receptor (Fig. 2-3). By using human cell lines engineered to down-regulate endogenous PGC-1 α levels and express in an inducible manner either the wt or a truncated version of PGC-1 α lacking the RS sequence (Δ RS, Fig. 4), they demonstrate that this protein domain is required for the nuclear export of a handful of mRNAs encoded by PGC-1 α target genes sharing mitochondrial functions (Fig. 5). Phenotypically, the nuclear retention of these transcripts is further correlated with defects in mitochondrial biogenesis/activity and cell proliferation in Δ RS cell lines (Fig. 6). Furthermore, whole proteome analysis reveals changes in the cellular levels of several proteins in the absence of the RS domain (Fig. 7), some of them involved in mitochondrial biology. Novel PGC-1 α -dependent proteins identified in this screen are further validated,

showing that the nuclear export of the corresponding transcripts also depends on the presence of PGC-1alpha RS domain (Fig. 8). Overall, while this study clearly establishes that the RS domain of PGC-1alpha is required for RNA/NXF1 binding, mRNA nuclear export and proper expression of dozens of proteins, the underlying mechanism remains elusive, especially in light of PGC-1alpha previously reported functions. Additional experiments are thereby required to substantiate the authors' findings, as described below.

1. As pointed in the introduction and discussed, PGC-1alpha reportedly contributes to transcriptional activation and splicing. Importantly, optimal mRNA synthesis from PGC-1alpha-target genes and splicing regulation by PGC-1alpha require its C-ter domain, which encompasses the RS sequence (authors' reference 17). In view of the critical importance of splicing for mRNA export, it is thereby possible that PGC-1alpha primarily controls mRNA processing, indirectly impacting on mRNA export.

- Since the role of PGC-1alpha in splicing was solely described using a reporter gene (reference #17), can the authors check whether splicing is globally affected in Δ RS cell lines? Is it the case for the transcripts for which nuclear export is compromised? RNA-seq-based comparisons of wt and Δ RS cell lines should address this important issue (see also below, 3.).

Since mRNA nuclear export is linked to co-transcriptional processing, we agree that impact on splicing may affect the nuclear export of mature mRNAs. However, PGC-1 α -dependent mRNA nuclear export targets were identified based on (i) down-regulated cytoplasmic expression levels and (ii) unchanged whole-cell RNA expression levels to avoid transcripts with potential altered transcription rate, processing and/or decay in the Δ RS-res or PGC-1 α -RNAi cells (**Fig. 5 d-e; new Figs. 8a-b, 9c**).

On the other hand, we agree with the Reviewer (and the other Reviewers) that additional genome-wide studies will substantiate mechanistically our findings and allow evaluating the global contribution of PGC-1 α functions in the regulation of gene expression. During the revision of this manuscript, we performed ChIP-seq and whole-cell/ cytoplasmic RNA-seq on the PGC-1 α WT-res and Δ RS-res cell lines to investigate promoter-binding and RNA expression, splicing and nuclear export at genome-wide levels (**new Figs 7 and 8**).

In relation to a potential role of PGC-1 α in splicing, we identified 897 alternative splicing changes in the Δ RS-res cell line (**new Fig. 8d**), however the total expression levels of mRNAs encoding 30 splicing factors are also down-regulated in the Δ RS-res cells (**new Supplementary Fig. 9e**). It is therefore not possible to conclude through this analysis whether the RNA-binding function of PGC-1 α plays a direct role in pre-mRNA splicing. On the other hand, we also characterised cytoplasmic transcriptomes to identify 1,274 mRNA nuclear export targets which are down-regulated in the cytoplasm but unchanged at total expression level (**new Fig. 8a-b**). There is a poor overlap between the alternative splicing and mRNA nuclear export targets (**new Fig. 8f**), suggesting that the nuclear export function of PGC-1 α is not linked to the recruitment of PGC-1 α during splicing in contrast to other mammalian mRNA nuclear export adaptors. Gene ontology analysis of the splicing list corroborated this, lacking enrichment in non-canonical mRNA-binding/nuclear export targets involved in glucagon-signaling (reported in the eCLIP data from Tavares CDJ *et al. PNAS* 2020; 117:2220) and age-related NHEJ maintenance of telomere and nucleocytoplasmic transport. This is also supported by the observation that the altered expression of alternative splicing in the Δ RS-res cell line may be caused by the down-regulation of mRNAs encoding 30 splicing factors.

- Are there intronless genes among PGC-1alpha transcriptional targets? If this is the case, these genes should be analyzed for RNA localization in the fractionation assay (Fig. 5) to explore a possible splicing-independent function of PGC-1alpha in mRNA export.

This is an excellent suggestion which can now be extended at genome-wide level in our cell model. However, we identified that only 1.1% of promoters bound by PGC-1 α and 0.1-0.2% of differentially-expressed mRNAs (*FOXD4*, *CCDC8*, *FZD8*) or mRNA nuclear export targets (*CEBPB*, *IR54*) involve intronless genes (*Intronless Genes Database*), indicating that PGC-1 α is not significantly involved in the expression of intronless genes. Therefore, we have not conducted any experiments around this point which is not included in the manuscript, however *CEBPB* and *IR54* mRNAs could indeed be quantified in the fractionation assay if deemed necessary.

- PGC-1alpha interaction with TREX (Fig. 2C) suggests that PGC-1alpha could enhance export of spliced transcripts by contributing to TREX recruitment. Could the authors perform TREX RIP on PGC-1alpha target transcripts in wt and Δ RS cell lines (see also below, 2.)?

This is a very good suggestion however, in light with other comments, we preferred to remove the data indicating co-immunoprecipitation of PGC-1 α with THOC1, Aly/REF and UAP56, as implicating that PGC-1 α associate with the TREX complex and plays a role as part of TREX would indeed require a more thorough investigation including *in vitro* pulldown assays with purified recombinant proteins and RIP assays (please also see complementary answer to point 2). In addition, the genome-wide investigation largely indicated that the nuclear export function of PGC-1 α is not linked to splicing and is either dependent or independent binding of PGC-1 α to promoters. This indicates that the recruitment of PGC-1 α to mRNAs are multiple and much more complex than initially thought from the data obtained in Figure 5, in which the transcripts expressed from known promoters and co-activated by PGC-1 α were also nuclear export targets. The text has now been revised throughout the manuscript to include the genome-wide data findings.

2. The authors fail to provide a mechanism reconciling *in vitro* findings (PGC-1alpha interaction with RNAs and NXF1) with *in vivo* observations (the requirement of PGC-1alpha for mRNA nuclear export).

- Does PGC-1alpha function as a (novel) NXF1 adaptor? Does it impact NXF1 RNA binding activity as previously reported for TREX (authors' reference #41)? To address these questions, the authors could ask (i) whether PGC-1alpha can simultaneously interact with NXF1 and RNA (using *in vitro* binding assays as featured in Figs. 1-3), and (ii) whether PGC-1alpha is required for proper loading of NXF1 onto its target transcripts (by performing NXF1 RIP in control and PGC-1alpha-depleted cells).

- The authors claim that the PGC-1alpha-TREX interaction is stabilized in the Δ RS mutant (lines 200-203, related to Fig. 2C), possibly reflecting a stalling in mRNP remodeling when PGC-1alpha does not interact with RNA. However, this conclusion is not supported by the data: the Δ RS mutant protein seems much highly expressed in transient transfections (see the input panel in Fig. 2C), and likely more abundant in immunoprecipitates. The increased interaction with TREX must thereby be quantified by normalizing the amount of co-immunoprecipitated TREX proteins to the amounts of PGC-1alpha detected in the respective pull-downs.

Our study indeed shows that the RS domain of PGC-1 α exhibits the hallmarks of mRNA nuclear export adaptors by directly interacting with RNA and the nuclear export receptor NXF1. This has been stated in the manuscript. However, we have not characterised the detailed molecular mechanisms of PGC-1 α recruitment to pre/mRNAs which in addition to the suggested binding assays using purified proteins and RIPs would also require structural studies, such as the determination of NMR solution structures, as reported in our previous published studies. Another challenge comes from the large protein size of PGC-1 α in contrast to the previous mRNA nuclear export adaptors that we have previously investigated (Aly/REF, SRSF1, 3, 7). As reported in this study, we cannot express full length PGC-1 α in *E. coli* and have not investigated its potential NXF1-remodelling properties or performed *in vitro* pull-down assays to test whether PGC-1 α directly interacts with the TREX complex and if it is the case, with which subunits. It is noteworthy that protein-binding by PGC-1 α was reported to involve both the N-terminal zinc fingers and the C-terminal RS domain. In addition, the co-transcriptional processing mechanisms of PGC-1 α recruitment to pre/mRNAs are likely to be multiple given that the new ChIP-seq and RNA-seq investigation identified dual mRNA nuclear export targets which do or don't involve PGC-1 α binding to promoters. Understanding the precise nature of these molecular mechanisms will require a few years of work and will be a completely novel study. We hope that the Reviewer and Editor can appreciate that this work is out of the scope of this manuscript which has been substantially revised with the inclusion of (i) ChIP-seq/ RNA-seq and (ii) proximity ligation assays showing interactions of PGC-1 α with NXF1 outside of splicing speckles/ paraspeckles and of PGC-1 α with NUP107, a nucleoporin in the nuclear outer ring at the entrance of the channel of the nuclear pore complex. This study already provides new comprehensive findings identifying for the first time a function for the RNA-binding activity of PGC-1 α using cellular, molecular, biochemical and functional assays as well as integrated ChIP-seq, RNA-seq and proteome datasets.

We agree that the statement “the PGC-1 α -TREX interaction is stabilized in the Δ RS mutant possibly reflecting a stalling in mRNP remodeling when PGC-1 α does not interact with RNA” is speculative and acknowledge the reviewer for improving the quality of our manuscript. Much more work would indeed be required to validate this statement and we therefore removed the couple of sentence about the co-immunoprecipitation of PGC-1 α with the TREX complex (point 1 above).

On the other hand, as rightly highlighted by the Reviewer, the Δ RS mutant protein is highly expressed in transient transfections and much more abundant in the immunoprecipitates. As suggested, we repeated the co-immunoprecipitation experiments in biological triplicates and normalised Δ RS to wild type immunoprecipitates to avoid saturating the western blots signals. In these conditions, the new experiment and quantification clearly show that deleting the RS domain of PGC-1 α leads to a significant reduction (~85%) in the co-immunoprecipitation of NXF1 (**new Fig. 2d**), reconciling *in vivo* findings with the *in vitro* data showing that the RS domain is required and sufficient for the interaction of PGC-1 α with NXF1 (**Fig. 2f**) and that deleting the RS domain of PGC-1 α also inhibits interactions with NXF1 in an *in vitro* pull-down assays using reticulocytes (**Supplementary Fig. 5**). Accordingly, the deletion of the RS domain of PGC-1 α in the Δ RS-res cell line and reduced interactions with RNA/NXF1 affects nuclear export of mRNAs targeted by PGC-1 α .

- Since PGC-1 α -dependent mRNA export occurs for PGC-1 α transcriptional targets

(Fig. 5), could the authors discuss how they envision the transfer of PGC-1 α from DNA to RNA?

The results obtained for 5 known promoters transcriptionally co-activated by PGC-1 α indeed indicated that the transcripts produced from these promoters were also all nuclear export targets of PGC-1 α . However, the genome-wide investigation identified mRNAs nuclear export targets which are mostly independent of PGC-1 α binding to promoters while only 18% correlate with PGC-1 α -dependent binding to promoters (**new Fig. 8f**), in agreement with studies recently identifying RNA-binding targets which are not transcribed from PGC-1 α co-activated promoters (Tavares CDJ *et al. PNAS* 2020; 117:2220; Perez-Schindler J *et al. PNAS* 2021; 118:e2105951118). Overall, the genome-wide study indicates that PGC-1 α is recruited to RNA through at least 2 different mechanisms involving (i) potential transfer from direct binding to promoters and/or from the RNA polymerase II to which it interacts via the Mediator complex and (ii) transcription-independent recruitment during pre-mRNA processing which is not linked to splicing. Nuclear mRNA export is also known to be coupled to cleavage/poly-adenylation, particularly in lower eukaryotes, and additional work in future studies will be required to investigate this point.

We dissected further the contribution of the different functions of PGC-1 α to gain mechanistic insights. Thermogenesis and oxidative respiration appear to be specific of the promoter-binding function of PGC-1 α while other canonical processes such as mitochondrial/ metabolic pathways, AMPK/ insulin signalling and regulation of RNA polymerase II transcription and longevity involve both the binding to promoters and mRNA nuclear export functions of PGC-1 α . On the other hand, non-canonical pathways including glucagon signalling (also identified in the eCLIP study by Tavares CDJ *et al. PNAS* 2020; 117:2220) and the newly identified processes such as translation initiation and age-related maintenance of telomere and nucleocytoplasmic transport are specifically regulated by the nuclear export function of PGC-1 α . A revised model is presented in **Fig. 10**.

3. The proteome analysis of wt and Δ RS cell lines reveals that the cellular levels of several dozens of proteins change depending on PGC-1 α activity (Fig. 7). While these results are interesting as a basis for understanding the relationships between PGC-1 α and multiple cellular or pathology-related processes, they do not document as such the proteome-wide impact of PGC-1 α mRNA nuclear export activity. Indeed, changes in transcriptional activation or splicing could also ultimately impact protein levels. Whole proteome analysis has thereby to be coupled with RNA-seq (in wt and Δ RS cell lines, see also above 1.), to identify the proteome changes that are primarily caused by alterations in mRNA nuclear export.

We acknowledge this comment and performed RNA-seq on whole-cell and cytoplasmic fractions from WT-res and Δ RS-res cell lines to investigate PGC-1 α -dependent RNA expression, splicing and nuclear export (**new Fig. 7 and 8**). Alternative splicing was summarised above. We identified that 1,274 mRNAs depend on the RS domain of PGC-1 α for their nuclear export (**Fig. 8a-b**). These involve transcripts encoding mRNAs involved in both canonical (mitochondrial/ metabolic pathways, AMPK/ insulin signalling and regulation of RNA polymerase II transcription and longevity) and non-canonical pathways such as the previously identified glucagon signalling (Tavares CDJ *et al. PNAS* 2020) or the novel predicted age-related maintenance of telomere and nucleocytoplasmic transport. We further validated experimentally that PGC-1 α binds these novel targets and is responsible for their nuclear export using RIP and qRT-PCR quantification of mRNAs in total, nuclear and

cytoplasmic fractions (**Fig. 9b-c**). Overall, 56% of the downregulated proteome changes depend on reduced cytoplasmic mRNA levels (**new Fig. 7j**). Importantly, 52% of the downregulated proteome changes are linked to the mRNA nuclear export function of PGC-1 α while 18% correlate PGC-1 α -dependent binding to promoters indicating that both canonical and non-canonical transcripts are regulated by the nuclear export function of PGC-1 α (**new Fig. 8f-g**). The genome-wide investigation also indicated that the nuclear export function of PGC-1 α controls the cytoplasmic abundance of 60% of its regulated mRNAs, while 94% of mRNAs with reduced cytoplasmic expression depends on the nuclear export function of PGC-1 α (**Fig. 8c**).

Minor points.

1. A few sentences need to be rephrased: « It further interacts and co-activates them » (line 77); « PGC-1alpha is able to splice a reporter.... It may co-transcriptionally splice transcripts it coactivates. » (lines 94-95).

We agree that these sentences do not read well and have rephrased them as follow: “It also directly interacts and co-activates these transcription factors” and “PGC-1 α co-transcriptionally splices reporter transcripts it co-activates”.

2. The authors should tone down novelty claims related to their data in the abstract (lines 39-40) or in the Introduction (lines 122-123).

“Putative” in a “putative RNA recognition motif” was removed and “the potential RNA-processing functions remained elusive” was replaced by “the RNA-processing function(s) were poorly investigated”. The second part of the abstract was rewritten to account for the incorporation of the new ChIP-seq and RNA-seq investigation.

3. Figure titles have to be corrected to reflect the featured data. For instance, Fig. 1 is entitled «The RS domain of PGC-1alpha interacts with transcriptionally co-activated mRNAs » while it does not demonstrate that this interaction is restricted to transcriptional targets. Fig. 7 indicates « Proteome-dependence of the mRNA nuclear export function of PGC-1alpha » while the authors have investigated the impact of the nuclear export function of PGC-1alpha on the proteome.

We acknowledge these comments and multiple titles have been revised. Figures 1 and 7 are now respectively entitled “The RS domain of PGC-1 α interacts with RNA” and “Genome-wide contribution of RS-mediated functions of PGC-1 α at promoter-binding, transcriptome and proteome levels”.

4. Fig. 1B: please provide lower exposures of the images to see the radiolabeled bands.

The radioactive signal is very strong as enough proteins need to be loaded for Coomassie-staining. However, we agree that the radiolabelled bands are saturated. This experiment was repeated several times and we now provide a different gel with a lower exposure (**Fig. 1B**).

5. The colocalization data (Fig. 2D, S4) are not convincing. Both NXF1 and PGC-1alpha appear to be nucleoplasmic with nucleolar exclusion, which could happen in the absence of any interaction. I would recommend to remove these data.

We agree with this comment and have removed the data. Instead, we have added a new figure showing by immunofluorescence staining and smFISH that PGC-1 α interacts with NXF1 in the nucleus outside of splicing speckles and paraspeckles (new **Fig. 3a**). Moreover, proximity ligation assays (PLA) confirmed that PGC-1 α and NXF1 interacts in the nucleus outside of the nuclear splicing speckles (new **Fig. 3b**) consistently with previous observations in the literature and Fig. 3a. In addition, we also show by PLA that PGC-1 α co-localizes with nucleoporin NUP107 in the nuclear outer ring at the channel entrance of the nuclear pore complex (new **Fig. 3c**) in full agreement with its role as a nuclear mRNA export adaptor licensing the nuclear export of mRNA.

6. Could the authors discuss the fact that they do not detect in their proteomic screen the targets of PGC-1alpha mRNA nuclear export activity as identified in Figs. 5-6 (i.e. TFAM, NRF1, COX10, COX5A, COX2)?

Proteome screen are known to be much less sensitive than RNA-seq. The data was validated by qRT-PCR for all (Fig. 5d-e) and by western blotting for TFAM (Fig. 5f-h). This is also clearly shown in Venn Diagrams comparing RNA-seq and proteome (new **Fig. 7j**). A comment was added in the text. Significantly, we found that 56% of downregulated proteome changes correlate with reduced cytoplasmic mRNA levels (new **Fig. 7j**) while 52% of the down-regulated proteome changes are linked to the mRNA nuclear export function of PGC-1 α (new **Fig. 8g**) and 1,274 mRNAs are regulated by the nuclear export function of PGC-1 α (new **Supplementary Table 4 tab 4**).

Additionally, the proteome screen was carried out using whole-cell lysate to look for global effect of the loss of the RS domain of PGC-1a. For more mitochondrial focus, mitochondrial enrichment prior to mass spectrometry analysis would be beneficial. Further, TMT labelling often suffers from ratio compression which may affect the quantification of peptides/proteins that are of low abundance in the sample.

REVIEWERS' COMMENTS

Reviewer #1 (Remarks to the Author):

The authors have adequately address my previous critiques with new experimental data and explanations. I also found the final model helps to integrate previous studies with the new findings with PGC-1 α and nuclear export.

Reviewer #2 (Remarks to the Author):

The authors have made a remarkable experimental effort, what has resulted in a much-improved manuscript. Nevertheless, there are some minor comments that I think they need to be considered:

In line 572 the authors include a reference to data not shown. This is far from optimal. The sentence must be removed, or the data shown in the results section.

In line 230 authors interpret PLA as a direct interaction assay instead of a proximity assay. The text needs to be amended so the interpretation reflects that PGC-1 α is in close proximity to NUP107 (not necessarily in direct interaction with it).

Personally, I think that the section about “Engineering of isogenic inducible cell lines to investigate the potential mRNA nuclear export activity of PGC-1 α ”, would be better placed in methods and supplementary material. It is not a result per se, but a methodological intermediate step required for the next results section.

The number of promoters bound by PGC-1 α is greatly increased when the RS motif is removed. Considering that the RS motif is implicated in RNA binding, it may be worth to include in the discussion the speculation about the potential role of RNA mediating the recruitment of PGC-1 α to specific promoters.

Reviewer #3 (Remarks to the Author):

The authors have made a substantial effort to address the reviewers' comments and now provide a more comprehensive set of data, disentangling the multiple roles of PGC1 α at different stages of the gene expression process. In this respect, with the added-value of genome-wide analyses, this study will nurture the research field and now deserves publication. I only have an important issue with the interaction now reported by the authors between PGC1 α and the Nuclear Pore Complex (NPC), based on Proximity Ligation Assays using antibodies directed to PGC1 α and the NPC component Nup107 (new Fig. 3c): this experiment does not convincingly demonstrate an interaction since the PLA signal consists of

nucleoplasmic foci. Since NUP107 is predominantly localized at NPCs, a relevant PLA signal would thus be expected to mostly enrich at the nuclear periphery (as exemplified in reported PLA assays using NPC-directed antibodies, e.g. PMID: 33023979), or at the surface of nuclei if using confocal microscopy to distinguish the nuclear envelope from the nucleoplasm. The signal observed by the authors is thus unlikely to correspond to NPCs and these data, which are not essential for the study, should be simply removed from the manuscript.

Minor remarks

1. The presentation of the genome-wide analyses could be improved with the following suggestions:

- it would be useful to provide a few snapshots of CHIP-seq signals over typical PGC1a-bound promoters, complementing the CHIP-seq data analysis featured in Fig. 7a-c.

- could the authors indicate whether the down-regulated mRNAs (n=616, WCT panel, Fig. 7e) correspond to transcripts encoded by PGC1a-bound genes (from their CHIP-seq data)?

- the authors should describe how the colors relate to the enrichment scores in Fig. 8h.

2. The wording has to be revised in some instances:

- Abstract – “novel regulatory targets of PGC-1 α in age-related non-homologous end-joining and nucleocytoplasmic transport”

NHEJ and nucleocytoplasmic transport cannot be described as “age-related” in this context.

- Line 101 – “PGC-1 α co-transcriptionally splices reporter transcripts it coactivates”

Rephrase: “PGC-1 α is involved in the co-transcriptional splicing of a co-activated reporter transcript”.

- Line 120 - “PGC-1 α couples an additional function in the nuclear export”

Rephrase: “PGC-1 α displays an additional function in the nuclear export”.

- Line 574 “Other proteins involved in binding the RS domain of PGC-1 α may therefore also contribute to stimulate the recruitment of PGC-1 α to processing transcripts”

Rephrase: “...to transcripts undergoing processing”.

- Line 577 “...indicating that it is recruited to nascent transcripts via at least 2 different mechanisms which remain to be identified in future studies”

Rephrase: “...suggesting that it could be recruited...”.

NCOMMS-22-04129A: Point-by-point response to reviewers

We acknowledge again the Reviewers and the Editor for their time in reviewing our revised manuscript and for their final constructive comments. Below, we are providing a point-by-point response labelled in blue in line with the revised text in the main article file.

REVIEWERS' COMMENTS

Reviewer #1 (Remarks to the Author):

The authors have adequately addressed my previous critiques with new experimental data and explanations. I also found the final model helps to integrate previous studies with the new findings with PGC-1 α and nuclear export.

We acknowledge these positive comments.

Reviewer #2 (Remarks to the Author):

The authors have made a remarkable experimental effort, what has resulted in a much-improved manuscript. Nevertheless, there are some minor comments that I think they need to be considered:

We acknowledge these positive comments.

In line 572 the authors include a reference to data not shown. This is far from optimal. The sentence must be removed, or the data shown in the results section.

The “data not shown” referred to an experiment which failed to work optimally and there is no data to show. We have therefore deleted the text related to this point.

In line 230 authors interpret PLA as a direct interaction assay instead of a proximity assay. The text needs to be amended so the interpretation reflects that PGC-1 α is in close proximity to NUP107 (not necessarily in direct interaction with it).

We agree that direct interactions cannot be concluded from PLA experiments. We acknowledge the reviewer for spotting this mis-leading interpretation and have now used “in close proximity” in line 230.

Personally, I think that the section about “Engineering of isogenic inducible cell lines to investigate the potential mRNA nuclear export activity of PGC-1 α ”, would be better placed in methods and supplementary material. It is not a result per se, but a methodological intermediate step required for the next results section.

We appreciate this comment. However, this is a short paragraph linking the biochemical and cellular assays to the functional investigation of the previous findings at single molecule and genome-wide level. We feel that this is an important transition that should not be buried into the methods. Given that the 2 other reviewers did not report any issue, we therefore left this paragraph in the main text.

The number of promoters bound by PGC-1 α is greatly increased when the RS motif is removed. Considering that the RS motif is implicated in RNA binding, it may be worth to include in the discussion the speculation about the potential role of RNA mediating the

recruitment of PGC-1a to specific promoters.

We agree that this result suggest that the RS domain may play a regulatory role in the binding of promoters, potentially via its interactions with RNA and proteins. It is an excellent suggestion and we have now included this point in the discussion (line 556).

Reviewer #3 (Remarks to the Author):

The authors have made a substantial effort to address the reviewers' comments and now provide a more comprehensive set of data, disentangling the multiple roles of PGC1a at different stages of the gene expression process. In this respect, with the added-value of genome-wide analyses, this study will nurture the research field and now deserves publication.

We acknowledge these positive comments.

I only have an important issue with the interaction now reported by the authors between PGC1a and the Nuclear Pore Complex (NPC), based on Proximity Ligation Assays using antibodies directed to PGC1a and the NPC component Nup107 (new Fig. 3c): this experiment does not convincingly demonstrate an interaction since the PLA signal consists of nucleoplasmic foci. Since NUP107 is predominantly localized at NPCs, a relevant PLA signal would thus be expected to mostly enrich at the nuclear periphery (as exemplified in reported PLA assays using NPC-directed antibodies, e.g. PMID: 33023979), or at the surface of nuclei if using confocal microscopy to distinguish the nuclear envelope from the nucleoplasm. The signal observed by the authors is thus unlikely to correspond to NPCs and these data, which are not essential for the study, should be simply removed from the manuscript.

PLA figures were not generated from confocal images but from whole cell imaging.

Therefore, what appears to be nucleoplasmic foci are in fact likely localised at the nuclear periphery but observed from the top of the cells. The text has now been revised to tone down the phrasing of "interactions" with "close proximity". We think this data fits well with a role of PGC-1 α in the mRNA nuclear export and have kept the data, given that the 2 others reviewers were supportive of this experiment.

Minor remarks

1. The presentation of the genome-wide analyses could be improved with the following suggestions:

- it would be useful to provide a few snapshots of ChIP-seq signals over typical PGC1a-bound promoters, complementing the ChIP-seq data analysis featured in Fig. 7a-c.

We acknowledge an excellent idea and have now added this data in Supplementary Fig. 9a. We also added some text in the main manuscript (line 348).

- could the authors indicate whether the down-regulated mRNAs (n=616, WCT panel, Fig. 7e) correspond to transcripts encoded by PGC1a-bound genes (from their ChIP-seq data)?

We agree that this is a suggestion. Only 15% of down-regulated mRNAs are transcribed promoters bound by PGC-1 α , in agreement with findings in figures 5e,h and with our related comment in the discussion (line 562-566). We included the above suggested comparison in the main text in line 394-395.

- the authors should describe how the colors relate to the enrichment scores in Fig. 8h.

Apologies for this omission. The color scale is now described in the figure legend.

2. The wording has to be revised in some instances:

- Abstract – “novel regulatory targets of PGC-1 α in age-related non-homologous end-joining and nucleocytoplasmic transport”

NHEJ and nucleocytoplasmic transport cannot be described as “age-related” in this context.

We agree that “age-related” could be interpreted in different ways. The link connecting our findings with age-related pathways is described and referred to with citations in the main text. We have therefore deleted “age-related” from the abstract.

- Line 101 – “PGC-1 α co-transcriptionally splices reporter transcripts it coactivates”

Rephrase: “PGC1-1a is involved in the co-transcriptional splicing of a co-activated reporter transcript”.

This has been corrected as suggested.

- Line 120 - “PGC-1 α couples an additional function in the nuclear export”

Rephrase: “PGC-1 α displays an additional function in the nuclear export”.

This has been corrected as suggested.

- Line 574 “Other proteins involved in binding the RS domain of PGC-1 α may therefore also contribute to stimulate the recruitment of PGC-1 α to processing transcripts”

Rephrase: “...to transcripts undergoing processing”.

This has been corrected as suggested (line 563 of the revised text).

- Line 577 “...indicating that it is recruited to nascent transcripts via at least 2 different mechanisms which remain to be identified in future studies”

Rephrase: “...suggesting that it could be recruited...”.

This has been corrected as suggested (line 579 of the revised text).